# Crystal gravity

Jan Zaanen[1], Floris Balm[1] and Aron J. Beekman[2]

[1]*The Institute Lorentz for Theoretical Physics, Leiden University, Leiden, The Netherlands and*
[2]*Department of Physics and Research and Education Center for Natural Sciences,*
*Keio University, 3-14-1 Hiyoshi, Kohoku-ku, Yokohama 223-8522, Japan*

(Dated: April 22, 2022)

We address a subject that could have been analyzed century ago: how does the universe of general relativity look like when it would have been filled with solid matter? Solids break spontaneously the translations and rotations of space itself. Only rather recently it was realized in various context that the order parameter of the solid has a relation to Einsteins dynamical space time which is similar to the role of a Higgs field in a Yang-Mills gauge theory. Such a "crystal gravity" is therefore like the Higgs phase of gravity. The usual Higgs phases are characterized by a special phenomenology. A case in point is superconductivity exhibiting phenomena like the Type II phase, characterized by the emergence of an Abrikosov lattice of quantized magnetic fluxes absorbing the external magnetic field. What to expect in the gravitational setting? The theory of elasticity is the universal effective field theory associated with the breaking of space translations and rotations having a similar status as the phase action describing a neutral superfluid. A geometrical formulation appeared in its long history, similar in structure to general relativity, which greatly facilitates the marriage of both theories. With as main limitation that we focus entirely on stationary circumstances – the dynamical theory is greatly complicated by the lack of Lorentz invariance – we will present a first exploration of a remarkably rich and often simple physics of "Higgsed gravity".

## CONTENTS

# I. INTRODUCTION AND OVERVIEW.

## A. Crystal gravity: the Higgs phase of general relativity.

While working on this paper, we had the uneasy feeling that we were rediscovering a wheel. But apparently this is not quite the case. This paper could have been written surely in the 1950's and perhaps even in the 1920's. It departs from the simple question: *how would the universe have looked like when all the matter and energy would occur exclusively in the form of solid matter?*

Although not widely disseminated in the physics community at large, it has been realized for a while that such a universe comprises the *Higgs phase of gravity* [1]. The most obvious example of a Higgs phase is the superconducting state. First consider the electromagnetically neutral superfluid as realized e.g. in Helium, breaking spontaneously the internal $U(1)$ symmetry. Upon "gauging" by coupling it to the electromagnetic gauge fields this turns into the superconductor. The relativistic version is the Abelian Higgs mechanism that formed the template for the non-Abelian generalization that underpins the Higgs mechanism of the standard model of high energy physics. In this analogy, the way that the solid state is discussed in physics textbooks is like the theory of superfluids but now revolving around the spontaneous breaking of the symmetries of space-time. But the role of the gauge fields is now taken instead by the dynamical space-time of general relativity (GR).

This combination of solid matter with the dynamical space-time of general relativity – "crystal gravity" – is the subject of this paper.

We are however focussed on a particular aspect of this problem that is arguably alluding to the very essence of "Higgsed gravity" that appears to be hitherto completely overlooked. It is in essence orthogonal to aspects that have been at the focus of attention in the gravitational community – the way that solid matter influences the time evolution in a cosmological setting and the way that strong gravitational fields modify matter itself.

The aspect we will focus on is a general motive associated with the Higgs phase that has been at the focus of attention in the condensed matter tradition: the "Abrikosov vortices" also called magnetic "fluxoids". This theme became central in the study of the phenomenology of superconductors as they are realized in the laboratory. Given also the importance for applications a vast literature emerged, e.g. [2, 3], but it appears to be not part of the standard canon of the relativist

community. To a degree this is intended to be a tutorial communicating these wisdoms through the barrier between the disciplines. It is actually unclear whether it will have any ramifications for e.g. "solid cosmologies", be it for reasons that are actually by itself quite untrivial. This will be explained in Section (VII).

What are the magnetic fluxoids? This revolves around type II superconductivity. The superconducting state is of course the incarnation of the Higgs phase that is realized in earthly laboratories. Upon applying an external gauge curvature – a magnetic field – the Higgs field (the superconductor formed from electron pairs) is in control. Rooted in the principles of spontaneous symmetry breaking, circulation can only occur in the form of *quantized* vortices. These "merge" with the background gauge curvature in a lattice of lines carrying a *quantized* magnetic flux: the Abrikosov flux lattice. We will refer to this *the gauge curvature is "topologized" in a Higgs phase*. This "topologization principle" is completely general dealing with the internal symmetries of any Yang-Mills theory. For instance, an example of a non-Abelian generalization of the fluxoid is the 't Hooft-Polyakov monopole of electro-weak theory.

Despite the fundamental differences between GR and Yang-Mills we will show that the phenomenology of the gravitational Higgs phase in this sense is governed by a machinery that is remarkably similar to the usual Yang-Mills affair. Perhaps disorientating for the relativist, the *spatial* manifold is on centre stage. The issue is that crystals break exclusively the translations and rotations of space, the time axis is not involved. The role of gauge curvature – e.g. the magnetic field of superconductivity – is now taken by the *geometrical* "curvature" of the spatial manifold. We put here quotation marks because this refers not only to the Riemannian curvature but also to the geometrical torsion of Cartan-Einstein theory.

The order parameter theory capturing the solid is the theory of elasticity [4]. Given its ancient origin, it has disappeared to a degree from the radar of the physics community. But as a field theory it is remarkably rich, revolving around rank 2 symmetric tensor fields. This becomes particularly manifest focussing in on the topological excitations. The analogous of the vortices of the superfluid span up a rich topological universe, involving dislocations, disclinations, grain boundaries and a lot more, a subject that is still at the focus of attention of the soft matter community while it is at center stage in engineering oriented materials science. The mathematical language to handle this has been available since the 1980's [5], highlighting already the relations with Cartan-Riemann geometry. It appears to be a historical accident that nobody got the idea to "gauge" this affair with the dynamical background geometry of Einstein theory.

This is what we will accomplish. This is very long paper for the simple reason that a lot is going on. For reasons that are beforehand not obvious, there are reasons to be sceptical whether this will be of serious consequence for physics. However, it is conceptually quite entertaining and we suspect that it may bear consequence for pure mathematics, suggesting hitherto unrecognized relations between three dimensional geometry/topology and the art of *discrete* geometry.

Before we proceed this introductory Section with an executive summary of what will happen (Sections I C - I J), let us start with a short discussion of the established state of the art of the understanding of gravity-sourced-by-solids and how this may relate to our findings.

## B. Elasticity and gravity: the state of the art.

A first obvious problem is the way that solid matter behaves under the condition that the gravitational forces are very strong. One encounters such circumstances dealing with the solid crust of a neutron star. The study of this "relasticity" started in the 1970's by Carter and others [6, 7], evolving in the course of time into predictions how this solid crust can modify the gravitational wave signals associated with neutron star mergers, see Ref. [8]. This revolves around questions like how this crust reacts to the extreme tidal forces during the merger, whether mountains may form and so forth. This focusses on how the solid crust matter *reacts* dynamically to the usual gravitational forces. We have nothing to add in this regard: our focus is on the conditions where the nature of gravity itself is changed by the interplay with the solid medium. As will be discussed at length in Section (IV), neutron star crusts should be bigger by many order of magnitudes in order to enter this regime. Given the actual size of neutron stars the approach taken in this literature is just appropriate.

Yet another school of thought has developed in the cosmology tradition, asking a question which is at first sight the same as ours: what would happen to the dynamical evolution of the universe when e.g. dark matter or the inflationary field would be elastic? An early contribution is by Spargel and coworkers [1] who appear to be the first to realize that elastic matter is Higgsing gravity. This turned in recent times into a substantial subfield by itself. The close relations with massive gravity is acknowledged [9]. Possible ramifications for cosmological evolution are a flourishing affair presently, both in the context of elastic dark matter (e.g. [10]) and inflation driven by solid matter [11]. Yet again, the emphasis is here on the *dynamical* evolution and we do not have much in the offering in this regard except for an assortment of caveats that may be of interest to this community to further refine their models.

In fact, the original motivation to have a closer look at these matters came from yet another modern development: the study of "strongly coupled" states of matter in D space-time dimensions using the holographic duality that maps it onto a classical gravity problem in a D+1 dimensional asymptotic Anti-de-Sitter spacecite [12]. This was triggered by the discovery of holographic superconductivity [13, 14], followed by holographic crystallization:

the spontaneous breaking of space translations in the boundary [15–18]. This is dual to a gravitational bulk that is different from the matters we just discussed in the sense that it is in the regime where GR is modified itself into the "crystal gravity" affair. A highlight of holography is the "fluid-gravity duality" [17, 19] highlighting the deep and beautiful relations between hydro in the holographic boundary and the near black-horizon gravitational physics in the bulk. The construction of a similar "elasticity-gravity" duality is still work in progress. However, the most pressing ingredient in this context is in the form of the linearized theory that is the starting point of this whole affair, that we will explain in Section II. This is however in the mean time independently discovered by others [20]. Although much of what we have to stay is rather tangential to this development, holographers may find it useful to have a closer look at the views and techniques of condensed matter physics in this context.

### C. Solid matter: the spatial manifold is on center stage.

Let us now embark on a summary of the paper that may help the reader to keep track of the long line of arguments.

Let us first stress the limitation of our exposition: the role of time. It is perhaps disorientating for the relativist: dealing with solid matter the curvature of the *spatial* manifold is at centre stage. This is deeply rooted in the uncomfortable fact that Lorentz invariance is broken in a most devastating way. Spontaneous symmetry breaking is a mighty force and when it happens it takes over the physics. In a solid the translational symmetry of space is broken but as we will discuss in more detail in Section II B it is impossible to break time translations. At least in our approach, departing from solid matter, this is messing up the tensor structure associated with dynamical evolution, complicating the computations greatly. For this technical reason as discussed in Section IV G we will concentrate on statics as it appears in a co-moving frame, leaving a generalization to the dynamical realms for future work. It is actually not quite clear to us how this motive is dealt with in the various "solid cosmology" approaches.

Eventually, at the centre of it one may well find a gravitational incarnation of a core business of materials science and metallurgy. The non-equilibrium physics of solids is well understood to be associated with the topological excitations: dislocations, grain boundaries and so forth. This is an affair that has been studied by mankind in essence since the start of the bronze age, being still a very large field of research in the present era revealing much complexity. Especially in so far the evolution of *spatial* curvature is at stake in "solid" cosmological dynamical evolutions one has to cope with these complexities. Ironically, at the very end of the development we will explain that for quite surprising reasons this may be less of an issue – the "dislocation gas" response to spatial curvature discussed in Section VII C.

### D. Elasticity as a tensor field theory.

Given this caveat, how to understand the analogy with the Higgs physics of Yang-Mills theories? To recognize the Higgs mechanism one needs next to the "gauge theory" (GR) an effective theory describing the consequences of the spontaneous breaking of the symmetry by matter: the "Josephson" or ("Stueckelberg") action descending from the neutral (ungauged) system. As a triumph of 19-th century physics for solids such a theory is lying on the shelf in the form of the theory of elasticity. Einstein himself was surely aware of it as testified by his metaphorical reference to elasticity, referring to space time as a "fabric".One benefit of the story we have to tell is that the shortcomings of this metaphor will become crystal clear.

One better be aware of a bias rooted in human intuition dealing with elasticity. This is rooted in the fact that the solid state of matter is the only form of spontaneous symmetry breaking being manifest to biological lifeforms. That we can sit on a chair staring at a screen with a solid encasing that rests on a table – it is overly familiar to us. But to explain this to a "liquid state intelligence" would require them to be trained physicists who understand the rule book of fancy field theory, containing chapters explaining that the spontaneous breaking of symmetry goes hand in hand with the emergence of rigidity – shear rigidity in the case of solids being the condition making it possible to sit on a chair.

In fact, elasticity was constructed as the first fledged order parameter theory by 19-th century mathematicans [4] on mere phenomeological grounds, long before the notion of order parameter was realized. As such it is actually quite sophisticated – it revolves around rank 2 symmetric tensors and it is in this regard a close sibling of GR. That there is something "gravity like" to elasticity is obvious to engineering students having the stamina to attend a GR course. It should be familiar for GR teachers that these students struggle less with tensor calculus than the average physics students since they already encountered it in the elasticity course. It is also the birth place of the idea of topological excitations, in the form of dislocations with their Burgers vector topological charge. However, in a more recent era it got banned to mechanical engineering departments and it is typically not longer taught in physics programs. But in the course of time a powerful field-theoretical machinery emerged to deal with especially the topological aspects of solid matter that was completed in the 1980's.

Given that this will be a rather unfamiliar affair for most readers we will review this "Kleinert style" formalism at length. On the other hand, we take it for granted that the readership is well at home with GR.

### E. The mathematical machinery: Kleinert's multivalued fields.

The relationship between gravity and elasticity in the way that these theories are conventionally formulated is not at all that obvious. But we are here helped by another development. From the 1950's onward, mathematically inclined elasticians were intrigued by similarities with GR in their exploration of the topological structure of elasticity. This was collected and further perfected by Hagen Kleinert in the mid 1980's casting it in a powerful field theoretical machinery highlighting the strong-weak duality structures [5, 21]. GR-like geometrical structure is shimmering through all along – Kleinert himself got distracted by his "worldcrystal" idea [22, 23]. Resting on the similarity with gravity, he contemplated the possibility that some special Planck scale crystal is formed that coarse-grains in GR itself. He overlooked the opportunity to couple in GR itself.

In a recent era two of the authors were themselves involved in further extending this affair into the realms of the weak-strong dualities associated with the description of quantum liquid crystals [24–26], highlighting the powers of the formalism. This is the reason that we are rather at home with this methodology that is otherwise not widely disseminated. This "GR-like representation" of elasticity turns out to be greatly convenient in the combination with gravity itself. At least within the limitation of this study all of the machinery we need can be found in Kleinert's toolbox [5], together with our quantum liquid crystal papers for some secondary issues [25]. To keep the presentation self-contained, we will derive and explain the required ingredients at length in this paper.

### F. The solids on the rigid two dimensional manifolds of the soft matter community.

Another mature affair of relevance to crystal gravity is the exploration of the soft matter community of solid matter covering the curved surfaces of three dimensional rigid bodies [27, 28]. With regard to elasticity the soft matter tradition is special because it did not forget the profundity of the solid state. This has its historical reasons. In fact, Kosterlitz and Thouless set out to address thermal melting of solids in two dimensions identifying the simpler "global $U(1)$" topological melting of the superfluid culminating in the famous BKT theory. Shortly thereafter this was generalized to the Nelson-Halperin-Young-Kosterlitz-Thouless theory addressing the topological thermal melting of triangular crystals in two space dimensions [29]. This revealed the existence of an intermediate "hexatic" liquid crystalline phase, while our quantum liquid crystal work [25] can be viewed as a generalization to the zero temperature quantum realms.

In the 1980's the issue of solid matter on rigid curved 2D manifolds came into view and has been pursued since then using experimental-, theoretical- as well as computational approaches. In part this is motivated by applications such as the way that quasi-solid protein structures cover cell walls. But it is also of theoretical interest. The curvature has to be "absorbed" by the topological excitations (dislocations, disclinations, grain boundaries) – the central ingredient of crystal gravity – but given the rigid curvature this turns into an intricate frustration problem.

One limitation of this tradition is that it addresses rigid background geometry instead of the dynamical GR geometry – for reasons that may surprise this will turn out to be relevant also for the physical universe as will be explained in Section VII. The other limitation is that the focus is exclusively on 2D curved manifolds characterized by the simple scalar (Gaussian) curvature. However, dealing with the three dimensional spatial manifold of crystal gravity the richness of Riemannian geometry becomes manifest with its 6 curvature invariants being a subject of considerable contemporary mathematical interest. This will be the central theme of Sections XIII and IX.

### G. A simple essence: the "wedge fluxoid".

As a first encounter, let us start with an elementary motive that will play a remarkably important role in the second half of this paper. Perhaps the simplest example of a curved Riemannian manifold is the conical singularity of gravity in 2+1D. The lecturer will fold a piece of paper in a cone. He has drawn a circle on the flat piece of paper, to then demonstrate that the circumference of this circle shrinks from $2\pi r$ to $2\pi(1 - \alpha)r$ where $\alpha$ is the opening angle of the cone, demonstrating the meaning of a geodesic in a curved background.

But this cone is made from a solid – paper – and in what regard does it fall short in explaining the Riemannian geometry? In fact, we have been exploiting the embedding in three dimensions. To form the papercone we have to tilt it in the vertical direction which does not exists in the 2D spatial manifold. What happens when one tries to accommodate the cone a flat two dimensional universe, the surface of the desk? Push the tip of the cone to the desk and the paper will crumble. This illustrates the confinement of curvature that will be a highlight dealing with curvature (Section VI).

But now imagine that the desk is instead a dynamical space-time with a curvature that can adapt to the presence of the cone. A conical singularity may form in this "background" spatial geometry that can be precisely matched to the paper cone so that the latter does not need to crumble. The most ideal way for the solid to accommodate the cone is by having a disclination at the tip (see Fig. 7, appendix C). This imposes a topological quantization of the opening angle in units of the discrete pointgroup symmetry of the crystal. The outcome is a quantized "curvature fluxoid" which is on this level quite similar to a magnetic fluxoid in a type II superconductor.

The role of the gauge curvature (magnetic field) is taken by the Riemannian curvature while the disclination is like the quantized vortex of the superfluid.

This object is actually an example of a "wedge fluxoid", that will be a point of departure of the systematic development starting in section VI. This will rest on a precise mathematical fundament, leaving no doubt that this intuitive story is correct. "Intuition" refers here to the way that our human visual system processes this affair: we literally "see" the formation of the geometrical fluxoid. This highlights our earlier statement that crystal gravity is remarkably easy to understand for the simple reason that biological evolution has taken care of a deep empirical understanding by the human mind of the only state of matter breaking symmetry spontaneously that is critical for our success as biological species.

## H.  The overview: the structure of crystal gravity and the main results.

We are done with the preliminaries and let us now explain how this paper is organized, summarizing the main results.

Once again, we include an extensive discussion of elasticity anticipating that it may be unfamiliar to many readers. This starts with a primer containing an elementary discussion of the topological defects of crystals (Appendix C)– a must read first for those who are not familiar with dislocations, disclinations and grain boundaries. We have done our best in the main text to render it to be self-contained with regard to Kleinert's rather intimidating repertoire, emphasizing and explaining in detail those results that are needed in crystal gravity. It may still be useful to have his book at hand – we will refer occasionally to passages in this book where some detailed results are derived. We will assume that the reader has at least a basic knowledge of GR, on the level of e.g. the Carroll text book [30, 31].

But the main reason for this text to be long is that there is much going on. "Stationary crystal gravity" is a remarkably rich affair and we have much to explain even in this first exploration that we do not claim to be exhaustive at all. Although it goes hand in hand with GR, the way that the case will develop is guided by what we like to call "crystal geometry": the geometrical view on the nature of solid matter. The order that the various aspects are introduced is for this reason different from what is found in the typical GR text book. After the preliminaries (Sections II, III) the reader will first meet the crystal gravity incarnation of *linearized* gravity in Sections IV, V where the dynamical space-time background only contributes in the form of the gravitational waves. In the second part (Sections VI-IX) the focus will be on curvature – "proper" GR.

The grand symmetry principle controlling space-time is mercilessly on the foreground in crystal geometry, in a way more evidently than in GR itself – the Poincaré group. This insists that one has to consider translations being in *semi-direct* relation to rotations (and boosts). Semi-direct means that translations and rotations are not independent. The infinitesimal translations governing the linearized theory have an independent existence. But since two *finite* translations correspond with the (non-Abelian) rotations, the latter are controlling the full non-linear theory.

In crystal geometry the infinitesimal translations are associated with *shear* rigidity, while the "gauge curvature" associated with these are entirely captured by the *dislocations*. These are close sibblings of the vortices associated with the global $U(1)$ symmetry controlling superfluids. When the latter are gauged these turn into the magnetic fluxoids absorbing the magnetic fields. Similarly, when the geometrical background is considered to be dynamical the dislocations "merge" actually with the geometrical *torsion* of Cartan-Einstein gravity! The outcome will be that the shear rigidity described by standard elasticity together with the dislocations combines with the gravitons and torsion in one package discussed in Sections IV-V.

It is perhaps not widely realized how easy it is to describe the *non-linear* extension of crystal geometry. This revolves around the spontaneous breaking of the *rotational* symmetry of space by the crystallization. There is in fact an emerging rotational "torque" rigidity but as a consequence of the semi-direct nature of the Galilean group this is *confined* in the solid in a flat background, in a surprisingly literal analogy with the confinement of colour charge in QCD. The associated topological current is called the *defect current*. Dislocations "know" about the rotations in the form of their topological charge which is the Burger's *vector* taking values set by the pointgroup symmetry of the crystal. The defect current is associated with an *infinity* of dislocations, the topological expression of the non linearity. The *disclinations* are a special part of the defect current that have a minimum core energy, characterized by a topological invariant (the Franck vector) taking values precisely quantized in terms of the point group symmetry (see Appendix C).

The issue is that these topological defect currents embody the Riemannian *curvature* in crystal geometry. These merge with the geometrical curvature of Einstein theory into a new wholeness. This package of rotations, torque rigidity, the rotational topology captured by defects currents and the geometrical curvature of Einstein theory is the subject of the sections VI-IX.

Having explained the principle underlying the organization of the paper let us now present an overview of the main results.

### 1.  The basics: elasticity and frame fixing (Section II)

The foundations of the mathematical theory are laid down in section II. Higgsing departs from matter (the Higgs field, electrons in solids, etc.) breaking symmetry

spontaneously. Combining this with gauge fields, this matter imposes a "preferred frame" ("fixed frame", whatsoever) having the physical ramification that the *gauge curvature* is expelled, the field strength has to vanish. The gauge fields are forced to be "pure gauge", only invariant under "passive gauge transformation". The gauge curvature can only re-enter in the form of massive topologically quantized fluxes: the magnetic fluxoids of the type II superconductor, as well as the Polyakov- 't Hooft monopoles of the standard model.

This has a vivid image in crystal gravity (Section IIA). Crystal geometry departs from the notion that the positions of atoms in a crystal span up a *coordinate system.* For instance, a cubic crystal is like a Cartesian coordinate system. This can be as well described of course in terms of arbitrary mathematical coordinate systems such as spherical- and cylindrical-, coordinate systems. These are the "passive" diffeomorphisms in this context.

In GR the metric tensor $g_{\mu\nu}$ is a gauge-variant quantity. However, in the geometrical interpretation of the crystal the symmetric rank 2 strain tensors take the role of the metric and the action depends explicitly on this metric: the theory of elasticity. The role of the phase stiffness of the superfluid is taken by the *shear rigidity* encapsulated by the shear modulus $\mu$ of the solid. In close analogy with the Stueckelberg (Josephson) construction for Yang-Mills fields, one can now lift this to deal with a dynamical geometrical background with the background metric tensor taking the role of the gauge field, the Einstein-Hilbert action that of the Yang-Mills gauge theory and the strain tensor being the analogy of the phase gradients, see Eq. (10).

This is the point of departure for the further developments. In the remainder of section II we collect textbook material. We already alluded to the "maximal" breaking of Lorentz invariance because time cannot be involved in the breaking of translations. Next to being a major complication in the formulation of the dynamical theory this has unusual consequences even in stationary set ups that cannot be stressed enough in the present context. Among others, we will find that the *dynamical* gravitons of the background couple exclusively with the modes of *static* elasticity (Section IIB). Finally, dealing with the linearized theory the simple theory of *isotropic* elasticity suffices and this is reviewed in Section IIC.

### 2. Vortex-boson duality: the analogy (Section III).

The key insight behind Kleinert's machinery is in the observation that the weak-strong "vortex-boson" duality of the $U(1)$ Abelian-Higgs system applies equally well to elasticity, be it that one has to generalize it to rank 2 tensor fields. In Section III we will review this $U(1)$ affair, as a template for what follows. The reader who is at home may still have a look to check the particular notations we will be using.

In short summary, employing a straightforward Legendre transformation the phase-action of the superfluid is transformed in a dual representation where the phase field turns into a $U(1)$ gauge field expressing that the superfluid currents do propagate forces. These are in turn exclusively sourced by the quantized vortices: in 2+1D this turns into a literal Maxwell theory where the vortex "particles" take the role of quantized electrical charges. This is effortlessly extended to the gauged superconducting case: the "supercurrent-" and the EM gauge fields couple by a simple BF term, see Eq. 25. Turning this into a gauge invariant form by employing helical projections anything that is desired is computed effortlessly by Gaussian integrations (Sections IIIB, IIIC). This is especially convenient for the construction of the magnetic fluxoid (Section IIID).

Precisely this machinery is in generalized form filling the engine room for crystal gravity. The novice should be particularly aware of the special status of the "linearized sector". This acts way beyond the usual infinitesimal amplitude limit. It is instead associated with the "self-linearizing" Goldstone modes implied by the spontaneous symmetry breaking. Everything non-linear is shuffled into the topological excitations. Also dealing with a non-linear theory like gravity this motive continuous to be valid which is the key to the ease by which crystal gravity can be charted. This "topologization" of the Higgs phase is the greatly simplifying circumstance, highlighting some simple insights in the intricacies of gravity itself.

### 3. Linearized crystal gravity (Section IV).

In this section the first steps are taken in the development of the theory. We unleash the same weak-strong duality as in Section III to the gauged strain elasticity action of Section IIA. This just amounts to the strain-stress duality for the matter sector that is overly well known among elasticians but now sourced by the metric fields associated with the background (Section IVA). The conserved stress fields are parametrized in terms of rank 2 symmetric tensor gauge fields as introduced by Kleinert that we call "stress gravitons". These parametrize the capacity of the actually static elastic medium to propagate the mechanical stress. These turn out to be on the same footing as the usual gravitons in that they couple through a BF type term, in the same guise as the "supercurrent-" and electro-magnetic photons of Section III.

The helical decomposition is particularly useful dealing with these tensor gauge fields (Sections IVB,C). The outcome is that a spin 2 sector is identified describing the simple linear mode coupling between the "shear" gravitons and the "gravitational" gravitons: Eq. (66) (Section IVD). All what remains to be done to address the physical consequences revolves around simple Gaussian integrations.

As anticipated by the earlier attempts, a most natural consequence of the "frame fixing" is the fact that the (gravitational) gravitons acquire a mass, in close analogy

with the generation of mass in the standard model (Section IVE). In the case of superconductors this translates into the London penetration depth. We find an expression for the "gravitational penetration depth" that is so simple that we reproduce it here,

$$\lambda_G = \frac{c^2}{\sqrt{16\pi G\mu}} \tag{1}$$

where $c$ is the velocity of light and $G$ is Newton's constant. Apparently this quantity is not known. It can actually be deduced merely on basis of dimensional analysis. The rigidity associated with the spontaneous symmetry breaking is uniquely captured by the shear modulus $\mu$, and together with $G$ and $c$ there is just one way that these can be combined in a quantity with the dimension of length: Eq. (1). The issue is that compared to other forces Newton's constant is extremely small and thereby $\lambda_G$ is very large. Filling in the shear modulus of steel, it follows that $\lambda_G$ is of order of a lightyear!

The implication is that solid matter of cosmological dimensions is seemingly required to realize physical consequences of crystal gravity. Off and on it has been speculated that dark matter could be elastic. Besides a graviton mass, this would imply that dark matter has to be characterized by *shear* rigidity, which does not appear to be widely acknowledged in this cosmological literature. Since dark matter couples gravitationally to the visible baryonic matter it should be that the suppression of shear strains should imprint on the distribution of stars and galaxies. It could be of interest to search in astronomical surveys for such effects

The meaning of $\lambda_G$ is that it represents the length where the fixed frame of the crystal starts to back react on the spacetime. At smaller distances the coupling of the solid to the gravitational background has no consequences. This is the same wisdom that applies to a superconducting island with a dimension smaller than the London penetration depth. The magnetic field then behaves as in vacuum, while the superconductor behaves as a neutral superfluid – in the present context, solids just obey standard elasticity. In fact, it is easy to show that at scales larger than $\lambda_G$ solids become "infinitely brittle" (Section IVF). In superconductors the phase mode of the neutral system turns into a "longitudinal photon" characterized by a Higgs mass. This translates in crystal gravity into a completely rigid response to external static shear stress.

An important motive is here the "maximal" breaking of Lorentz invariance, having as ramification that these Higgsing effects are entirely restricted to the *static* elastic responses. The propagating phonons are completely decoupled: these stay massless! We will address this in detail in Section IVG. This was actually all along accounted for in the design of Weber bar class of gravitational wave detectors. In this section the reader may also get an impression of the technical complications one faces when one attempts to formulate a fully dynamical

crystal gravity theory.

### 4. Translational topology: dislocations and Cartan torsion (Section V).

What are the internal topological sources for the "stress gravitons"? Proceeding in close analogy with superfluids one finds these to be the *dislocations* (Section VA). As the shear modes, their topological currents are rank 2 symmetric tensors taking values in the point groups of the crystal (Section VB), and we show how to construct these currents in a dynamical 3+1D setting as well (Section VC). Dislocations are well known to accelerate in the presence of an external field of static shear stress. Since the latter couples to the gravitons, we show that gravitational waves are actually dissipated when they propagate through a solid containing dislocations, while a perfect crystal would be perfectly transparent for them (Sections D,E). This may be of interest in the context of black hole mergers: nearby rocky planets (littered with dislocations) may explode when the gravitational wavefront passes by.

As in superconductors one expects that the material topological defects (vortices) merges with background curvature (magnetic field) in fluxoids where the latter curvature is topologically quantized. Dislocations do couple to the gravitational waves of Einstein theory but these have an exclusive dynamical existence and are irrelevant in this context. The geometrical curvature of GR is yet to be identified at this stage. What is then the nature of the "gauge curvature" that can merge with the dislocation? The answer is: the *geometrical torsion* as introduced by Cartan (Section VF). Nothing will happen to dislocations dealing with standard Einstein gravity. Torsion has to be promoted to a dynamical property of the space-time manifold as is accomplished by Einstein-Cartan gravity. This theory in turn implies that torsion may well be absent in the background for dynamical reasons, and there does not seem to be much of physical relevance to be discovered here. The dislocations will however much later in the development (Section VII) make a glorious come back.

### 5. The topologization of curvature: torque gravitons and the defect density (Section VI).

Einstein theory revolves around the geometrical *curvature* of Riemannian manifolds and there has been no mention of it yet arriving at this point halfway this treatise. But this not an accident. Crystal geometry is characterized by a hierarchy that it shares with GR. All we encountered in the preceding sections are the gravitons, the infinitesimal perturbations of the metric. Although realized among relativists, it is not emphasized in elementary textbooks that individual gravitons have nothing to say about any form of curvature. An infinity of gravi-

tons is required to construct a curved manifold – GR is intrinsically non-linear.

This originates in the semidirect relation between infinitesimal Einstein translations and rotations/boosts of the Poincaré group. In the "fixed frame" crystal geometry the consequences become exquisitely transparent. A first elementary question one should ask, where are the Goldstone bosons associated with the spontaneous breaking of the *isotropy* of the manifold by the crystal? It appears to be not widely appreciated that one can identify such rotational modes that we call "torque" modes, or "torque gravitons" on the stress side of strain-stress duality. Torque is the stress associated with rotations. The key insight is that torque is *confined* in a solid, having the same meaning as in confinement of colour charge in quantum chromo dynamics although the mechanism is of a completely different kind. The physical manifestation of it is yet again overly familiar: apply a torque to one end of a shaft in an engine and it will be transferred to the other end in a completely rigid fashion.

In the elasticity literature a highly convenient way to identify these confined torque modes was identified, called "double curl gauge stress fields" – in the present context we rename them "torque gravitons". Using the same procedure of identifying the multi-valued rotational field configurations it is then straightforward to isolate the internal topological sources: the so-called "defect densities". These are in turn uniquely associated with the *Riemannian intrinsic curvature* in the geometrical interpretation. In a flat background these are confined, in the same guise as that quarks are confined as sources of the confined gluons. In fact, this is behind the paper cone classroom experiment of Section I G: the crumbling of the paper illustrates this confinement.

Using this machine it is straightforward to couple in a dynamical background and effortlessly the main result of crystal gravity is derived (Section VIA). This is encapsulated by an elegant formula. The curvature in the background and the topologized curvature of the crystal have to satisfy the simple constraint equation in three space dimensions (see Eq. 112),

$$G_{ab} = -\frac{1}{2}\eta_{ab} \qquad (2)$$

$G_{ab}$ refers to the spatial components of the *Einstein tensor* enumerating the curvature in the background. $\eta_{ab}$ refers to the spatial components of the symmetric rank two *defect density* tensor representing the rotational topological density of the crystal. This constraint is rigorously imposed by the confinement: a violation costs an infinite amount of energy. The remainder of the paper deals with exploring the consequences of this simple result.

The defect density $\eta_{ab}$ is (somewhat implicitly) the main actor in the soft matter pursuit studying the rigid 2D curved manifolds. It is conceptually simple and insightful also in relation to GR itself. We already alluded

to the fact that an infinite number of gravitons is required to "construct" curvature. The defect density is the algebraic topology image of this affair. Dislocations are the defects associated with the infinitesimal translations. Defect density is literally constructed by "piling up" an infinite number of dislocations *with equal Burgers vectors* (Section VIB and Appendix C).

These dislocations can however be organized in different ways, pending the dynamics of the solid. This flexible nature of the rotational topology/curvature will be crucial for the further developments. The most costly way to accomodate the curvature in the crystal is by just invoking a structureless "gas" of equal Burgers vector dislocations. Next to the fact that one has to cope with the finite density of energetically costly dislocation cores, "equal sign" dislocations repel each other by long range strain mediated interactions. The latter can be avoided by organizing the dislocation in planes. These are called "grain boundaries" which are present at a high density in nearly every solid that does not form a single crystal. However, when such a grain boundary ends somewhere in the 3D solid the line where this happens represents an instance where the curvature is localized (see Fig. **??** in Appendix C). This is literally what we accomplished in 2D by constructing the cone: the seam where we glued the sheet of paper in the cone is precisely like a grain boundary and the singularity at the tip represents the end of the seam.

Finally, one can make this seam disappear completely so that only the line (in 3D) of "tip singularities" remains. This is the *disclination*, see Fig.'s (7, **??**). Only in this case the rotations and the curvature are subjected to a precise topological *quantization*. Opening angles etcetera can take arbitrary values dealing with the grain boundaries and isolated dislocation but the condition that the seam disappears implies the topological invariant called the Franck vector governed by the point group of the crystal.

In the remainder of this section aspects of this fundamental curvature part of crystal gravity in both three (Section VIC) and two space dimensions(Section IVD) is further detailed establishing contact with the soft matter program.

### 6. Curvature fluxoids and the gravitational obstruction (Section VII).

We proceed by zooming in on the consequences of the "master equation" Eq. (2). The interest is in first instance in circumstances where the geometrical curvature is topologically quantized, involving the disclinations. This puts the paper cone intuition of Section I G on a firm mathematical ground. The material cone with quantized opening angle (like the famous "five ring" graphene disclination, Fig. 7) merges with a conical singularity in the background with the same opening angle into a "wedge" curvature fluxoid (Section VIIA). Such a curvature flux-

oid is similar to a magnetic fluxoid in a superconductor merging the quantized circulation of the superfluid with the gauge curvature into a quantized magnetic flux. One difference is that the core size of the geometrical fluxoid is not set by the (gravitational) penetration depth but instead by the lattice constant reflecting the confinement.

These 2D solutions are trivially extended to 3+1D, where the point like (in space) wedge fluxoids are pulled in *lines* propagating along lattice directions. The background geometry is literally identified with the standard *cosmic strings*, well known to correspond with a string of conical singularities in the spatial plane orthogonal to the propagation direction. This is only part of the story and in section IX we will generalize it further.

But at this instance the Lorentzian time-axis critically interferes. Conical singularities and cosmic strings are *gravitating* objects. Eq. (2) only involves spatial directions affected by the spontaneous symmetry breaking. However, one also has to satisfy the temporal Einstein equations. These in turn insist that a gravitating mass should reside at the "tip of the cone", in simple relation to the opening angle. Given the topology of the solid, the curvature quanta have to be "of order one". This requires that the curvature fluxoids require a mass- (2+1D) or string tension (3+1 D) set by the Planck scale localized in a volume corresponding with the lattice constant of the crystal (Section VIIB)! Considering mundane crystalline matter such Planck scale stuff required to "decorate the cores" is not available. We conclude that these quantized curvature fluxoids cannot be formed in the physical universe.

It should still be possible to accommodate a crystal in a spatially curved background manifold. At this instance, the intricacies of the "true" topological *defect* current that acts on the r.h.s. of Eq. (2) as discussed in Section VI come to help..The obstruction preventing the formation of the curvature fluxoids is coincident with the background curvature becoming effectively rigid – it behaves like the soft matter rigid surfaces. Any attempt to localize the curvature will run in the Planck scale brick wall and instead the crystal has to adapt to the slowly varying background curvature. We will argue that in the limit of relevance – curvature radii being very large compared to the lattice constant – there is a unique solution in the form of a dislocation gas (Section VIIC).

### 7. The Platonic incarnation: the gravitational Abrikosov lattice and the polytopes (Section VIII).

The Lorentzian time axis of the physical universe spoils the fun. A mathematical germ is lying in wait: crystal gravity appears to relate different branches of modern mathematics in a way that to the best of our understanding has not yet been realized. It requires some degree of mathematical idealization. In the first place, just omit the Lorentzian signature time axis of physics and focus in on 3D geometrical manifolds with Euclidean signa-

ture. In addition, assert that the crystal is perfect while the curvature of the background manifold has to be accommodated by the quantized curvature fluxoids. The situation is then analogous to type II superconductors where the gauge curvature can only be accommodated in magnetic fluxoids: the outcome is a regular, typically triangular Abrikosov lattice of fluxoids.

The mathematical challenge is then as follows. Insist that the geometrical curvature fluxoids have to form a lattice that is as regular as possible, how does such a "type II" lattice looks like given a crystal characterized by a particular spacegroup and a background geometry having a particular isometry and topology that has to be "absorbed" by the fluxoids?

So much is clear that this question has a direct bearing on the grand differences between two- and three dimensional geometry. The former was already understood in the 19-th century while the latter is still of contemporary interest. The outcome for the (tractable) 2D problem case is entertaining. Consider the generic closed manifold $S^2$, the ball that is easily visualized by embedding in three dimensions (Section VIIIA). The outcome is that the lattice of wedge fluxoids corresponds with a *discrete* geometry manifold that dates back to Greek antiquity: these correspond with the surface of the *regular polyhedra*. For instance, departing from a square lattice the curvature fluxoids are characterized by $\pi/2$ opening angles. These span up precisely a cube in the three dimensional embedding. The edges of the cube do not represent intrinsic curvature, and neither do the sides. The curvature is localized at the 8 corners that coincide with the fluxoids.

What happens in 3D? To give a first taste, in section VIIIB we present a minimal example. We depart now from a cubic crystal and ask what to expect when this is Higgsing the three dimensional ball $S^3$. In addition, we only employ the wedge fluxoid the cosmic string like object of section VII. The outcome is greatly entertaining: it corresponds with the surface of a *tesseract*, the generalization of a cube to 4 embedding dimensions! It is now easy to count the number of wedge fluxoids that are required: these now correspond with the 32 edges of the tesseract.

The tesseract is an example of a *polytope*, the generalizations of the polyhedra to higher dimensions. These are part of the general subject of discrete geometry and the classification of polytopes is a subject that is still pursued at the present day. The take home message is that the intricate subject of 3D topology and Riemannian geometry acquires a relation with discrete geometry in 3D by the "topologization" due to the Higgsing. Is it possible to associate the still evolving classification of polytopes due to contemporary giants like Conway and Coxeter with the 230 space groups in 3D, as well as the intricate mathematical art of geometry and topology of 3D Riemannian manifolds?

TABLE I. The route map of the duality landscape.

| Abelian Higgs (2+1 D) | CG: translations/torsion (3D) | CG: rotations/curvature (3D) |
|---|---|---|
| Josephson action (phase, $\phi$) $(\partial_\mu \phi - A_\mu)^2$ | Elasticity (strains, $w_{ab} = (\partial_a u_b + \partial_b u_a)/2$ ) $(w_{ma} + h_{ma}/2)C_{mnab}(w_{nb} + h_{nb}/2)$ | Elasticity (strains, $w_{ab} = (\partial_a u_b + \partial_b u_a)/2$ ) $(w_{ma} + h_{ma}/2)C_{mnab}(w_{nb} + h_{nb}/2)$ |
| Conserved supercurrent $\partial_\mu j_\mu = 0$ | Conserved stress $\partial_m \sigma_{ma} = 0$ | Conserved stress $\partial_m \sigma_{ma} = 0$ |
| Current photon $b_\mu$ $j_\mu = \epsilon_{\mu\nu\lambda}\partial_\nu b_\lambda$ | Shear graviton $b_{kl}$ $\sigma_{ma} = \epsilon_{mnk}\partial_n b_{ka}$ | Torque graviton $\chi_{kl}$ $\sigma_{ma} = \epsilon_{mnk}\epsilon_{abc}\partial_n \partial_b \chi_{kc}$ |
| BF term: EM and $b$ EM field strength $F$ $\epsilon_{klm}b_k F_{lm}$ | BF term: torsion and $b_{ka}$ Torsion tensor $S$ $\sqrt{|g|}\epsilon_{klm}b_{ka}S_{mla}$ | BF term: curvature and $\chi$ Einstein tensor $G$ $i\chi_{kc}G_{kc}$ |
| London penetration depth $\lambda_L = \frac{c}{\omega_p}$ | Gravitional penetration depth $\lambda_G = \frac{c^2}{\sqrt{16\pi G\mu}}$ | Confinement |
| Vortex current $J_\mu^V = \epsilon_{\mu\nu\lambda}\partial_\nu \partial_\lambda \phi^{MV}$ $ib_\mu J_\mu^V$ | Dislocation density $J_{ka} = \epsilon_{klm}\partial_l \partial_m u_a^{MV}$ $ib_{kl}J_{kl}$ | Defect density $\eta_{ka} = \epsilon_{klm}\epsilon_{abc}\partial_l \partial_b w_{mc}^{MV}$ $i\chi_{kl}\eta_{kl}$ |
| Top. constraint $J_a^V = \epsilon_{alm}F_{lm}$ Top. charge: flux quantum $\Phi_0$ | Top. constraint $J_{ka} = -\sqrt{|g|}\epsilon_{klm}S_{mla}/2$ Top. charge: Burgers vector $B_a$ | Top. constraint $\eta_{kl} = -G_{kl}/2$ Top. charge: Frank vector $\Omega_a$ |

*8. But there is also twist in three dimensions (Section IX).*

In 2D the wedge fluxoids exhaust the repertoire of topological curvature densities. But in 3D there is more going on, beyond pulling such wedge defects in strings. These wedge fluxoids are characterized by Franck vectors that are parallel to the propagation direction of the string, corresponding with the diagonal components of the defect density tensor. But pending the space group the Franck vectors can also point in other directions, relating to off-diagonal components of the defect density and the Einstein tensor, Eq. (2).

Taking the simple cubic crystal as an example, in Section IXA we zoom in on the construction of such "twist fluxoids". We are much helped by the fact that the corresponding background geometry is known. This forms by itself a barely chartered affair which is ruled by the non-Abelian nature of the 3D point groups. We study the holonomies associated with the set of twist and wedge fluxoids revealing the non-Abelian nature: for instance, geodesic transport will be dependent on the order one encounters the various fluxoid strings.

Closed homogeneous manifolds in 3D are famously captured by the Thurston classification, insisting that different from 2D there is more going on than only spherical-, hyperbolic- and flat manifolds. This will surely add fur-ther layers of richness to the gravitational Abrikosov lattice question. In Section IXB we take a short look to get an idea what can happen. We consider the "Kantowski-Sachs" Thurston class, corresponding with a 3D cylinder type geometry. We establish that this is topologized in terms of a simple bundle of wedge fluxoids. Then we revisit the maximally symmetric $S^3$ combined with the cubic crystal lattice, with the awareness of the existence of twist fluxoids. We argue that still a tesseract is formed but there are now a total six different ways to absorb the curvature using strictly degenerate configurations of twist- and wedge fluxoids. This affair is further enriched by degeneracy in the overall topology that is rooted by the non-Abelian nature descending from the rotational symmetry, that begs for further mathematical exploration.

*9. The conclusions (Section X).*

We will finish with an assessment of the significance and relevance of our findings. Those readers who perceived the above overview as more or less comprehensible may desire to first have a look at the conclusions before delving into the main text. We point at various opportunities for follow up work that may appeal to particular potential stake holders within the rather diverse selection of subjects that play a role.

## I. The route map of dualities.

As stressed in the above, the duality structures that are at the heart of our exposition are close siblings of the Abelian-Higgs duality in 3D that may be quite fa-miliar to part of the readership. To help the reader with keeping track of this rich affair we constructed the table (I H 9) summarizing the analogies between the various cases. The first column refers to the Abelian-Higgs template, the second column refers to the "translational"

sector (Sections IV, V) and the third column to the "rotational" part (Sections VI-IX).

### J.  A note on conventions.

Dealing with curved manifolds and/or Lorentzian signature one has to pay tribute to covariant- and contravariant quantities, raising an lowering indices using the metric tensor. Delving into the "crystal geometry" the focus will be most of the time on the spatial manifold with its Euclidean signature. Moreover, given the central principle that the solid will expel the spatial curvature (and torsion) as well, concentrating it in the topological excitations, one encounters here invariably as flat metric with Euclidean signature. Accordingly, there is no difference between co- and contra variant quantities and we will follow e.g. Kleinert in this context just ignoring the "upper" indices throughout the text.

However, there are a few occasions where the time axis enters: the gravitons in Section IV and implicitly when dealing with the core of the curvature fluxoids in Section VII where we will pay full tribute to the indices when the need arises.

Notice that we are in this regard just plainly sloppy in section III dealing with the vortex-boson duality – although we use greek indices these refer to 2+1D *Euclidean* space time where yet again there is no need to raise indices.

## II.  FRAME FIXING: ELASTICITY AS THE HIGGS FIELD OF GRAVITY.

The first part of this section is dedicated to the derivation of the "central principle" of crystal gravity Eq. (10), demonstrating the remarkably close analogy with the usual Higgs mechanism. In spirit, this is coincident with the established notion that crystals are Higgsing dynamical geometry. However, in much of this literature one deals in a rather casual way with elasticity itself, it seems because of unfamiliarity (and underestimation) of this theory. Only rather recently proper derivations appeared, in the holography inspired literature [20]. Our presentation here is strictly equivalent, but we have done our best to expose the bare essence of the mechanism. This may appear at first sight as rather intuitive but this is misleading. The Higgs mechanism is taught in the rather abstract setting of the internal symmetries of Yang-Mills theories. But crystal gravity is about the coordinate frames familiar from high school that get "frozen" by the presence of the crystal having the implication that the geometrical curvature (and torsion) are expelled. Upon getting used to the idea, it is by far the easiest way to explain the Higgs mechanism to students, assuming that they know the basics of the general relativity.

This we will accomplish first. In the remainder of this section we will then emphasize the breaking of Lorentz invariance (Section II B). We finish with a discussion of isotropic elasticity, including the spatial angular momentum decomposition that is of particular significance in the gravitational context (Section II C).

### A.  The first law of crystal gravity.

Space-time is as a fabric — this popular metaphor implicitly involves a notion that space-time may have something to do with a solid. Of course, every physicist realises that this metaphor should not be taken literal. But where precisely lies the difference? In fact, in the 'modern' tradition of elasticity, where we take the treatise of Kleinert in the 1980s as benchmark [5], it is formulated as a geometrical theory, in the same sense that GR is geometrical. Surely, the nature of this "crystal geometry" is in a crucial regard very different from the Riemannian geometry underlying gravity. Here we are focussed on the combination of the two where the space-time is sourced by elastic matter. Then it is quite helpful to have them both formulated in the same mathematical language.

The foundation of crystal geometry is very easy: it is just Einstein theory "in a preferred frame" (or "prior geometry", "fixed background"). The notion of a preferred frame is blasphemy for the relativist. The point of departure of Riemannian differential geometry is that since coordinates are an auxiliary device facilitating computations, the geometry as such cannot possibly depend on the choice of frame. Accordingly, diffeomorphism invariance (general covariance) is the ruling symmetry principle in the effective field theory strategy employed by Einstein to construct GR. The metric tensor $g_{\mu\nu}$ in coordinate representation is frame-dependent and should therefore not appear in the action. The lowest-order diffeomorphic invariant that qualifies to determine the action is the Ricci scalar $R$. This inspired Hilbert to rewrite the space-time parts of Einstein's equations of motion in terms of the action,

$$S_{\mathrm{EH}} = \int \mathrm{d}t\mathrm{d}^d x \, \sqrt{-|g|} \, (R + \Lambda + \cdots) . \qquad (3)$$

Including the cosmological constant $\Lambda$ while $\cdots$ refers to higher order curvature corrections such as $R_{\mu\nu}R^{\mu\nu}$ where $R_{\mu\nu}$ is the Ricci tensor. What should an effective field theorist write down instead when the geometry would be governed by a particular coordinate system which is for whatever reason becoming a physical observable? Let's call this frame $\hat{x}^\mu$ with metric $\mathrm{d}s^2 = \mathcal{G}_{\mu\nu}\mathrm{d}\hat{x}^\mu\mathrm{d}\hat{x}^\nu$ in terms of a *preferred* metric tensor $\mathcal{G}_{\mu\nu}$. The metric $g_{\mu\nu} \to \mathcal{G}_{\mu\nu}$ has turned into a physical observable and to lowest order in gradient expansion the minimal action becomes,

$$S_{\text{ff}} \sim \int \mathrm{dt d}^d x \, \sqrt{-|\mathcal{G}|} \left( -\frac{1}{2} \mathcal{G}_{\mu\nu} \mathcal{C}^{\mu\nu\kappa\lambda} \mathcal{G}_{\kappa\lambda} + \cdots \right), \quad (4)$$

where the tensor $\mathcal{C}$ containing coupling constants is constrained by the global symmetries (rotations, boosts) associated with the preferred frame. How can such a preferred frame spring into existence? A practitioner of the theory of elasticity may already have recognized that Eq. (4) may have dealings with this theory. Observing it with X-ray diffraction one immediately discerns that the crystal spans up a coordinate system. A square lattice in 2D is just like a simple Cartesian coordinate system. But now we combine it with the dynamical geometry of GR. The atoms localized on the points of this lattice carry mass and will back react on the geometry. Although this back reaction is extremely small on the scale of the lattice constant this will accumulate when the size of the crystal is growing to become consequential on the scale of the gravitational penetration depth $\lambda_G$, Eq. (1). On scales large compared to $\lambda_G$, by the imprinting of the crystal lattice the dynamics of space-time as otherwise governed by GR will be altered according to the consequences of Eq. (4). This is the essence of crystal gravity.

Although it is all about the dynamics of space time captured in the geometrical language of GR this "Higgs mechanism" that we just described is a close analogue of the Higg's mechanism of Yang-Mills theories. Let us remind first the reader of the way that this Higgs mechanism works. Higgsing means that matter characterized by its order parameter forces the gauge fields to become pure gauge expelling the gauge *curvature* corresponding with the physical field strength from the vacuum. Consider for example a scalar order parameter field $\Psi = |\Psi| \mathrm{e}^{\mathrm{i}\phi}$ minimally coupled to a $U(1)$ gauge field $A_\mu$. The physics deep in the relativistic abelian Higgs phase where the amplitude $|\Psi|$ is frozen is described by the Stueckelberg (Josephson) effective field theory (suppressing dimensionfull constants),

$$S_{\text{Jos}} = \int \mathrm{dt d}^d x \Big[ -\frac{1}{2} |\Psi|^2 (\partial_\mu \phi - A_\mu)(\partial^\mu \phi - A^\mu) - \frac{1}{4} F_{\mu\nu} F^{\mu\nu} \Big]. \quad (5)$$

In the neutral system (superfluid) this would reduce to the action describing the phase mode or "second sound" $\sim (\partial_\mu \phi)^2$, the Goldstone boson arising from breaking of the global $U(1)$ symmetry associated with conserved particle number. Upon gauging, the EOM following from Eq. (5) implies the gauge invariant condition $A_\mu = \partial_\mu \phi$, the gauge field *has* to be the gradient of a scalar function and thereby the gauge curvature $F_{\mu\nu} = \partial_\mu A_\nu - \partial_\nu A_\mu$ is vanishing. The field strength can re-enter above the Higgs mass, the energy scale where the spontaneous symmetry breaking is loosing control, or in a static setting in the form of the quantized vortices of the superfluid

turning into quantized magnetic fluxes (fluxoids) after gauging.

A greatly simplifying circumstance of spontaneous symmetry breaking is that through the Goldstone theorem the theory can be enumerated resting on the linearized sector. Focussing on the large wavelength Goldstone modes the relative displacements on the microscopic scale become infinitesimal and these are perfectly captured in a gradient expansion: to discern the fate of the photons one only has to consider infinitesimal phase fluctuations $\delta\phi$ in Eq. (5). The non-linearities are entirely collected in the sector of topological excitations (the fluxoids) and these can be captured by considering the parallel transport in large loops encircling the defects accumulating again phase differences that are infinitesimal on the microscopic scale.

Having this wisdom in mind, how to derive the gravitational analogue of the Josephson/Stueckelberg action Eq. (5)? The role of the matter breaking spontaneously symmetry should now be related to the formation of the crystal lattice forming the material "preferred frame". The Goldstone modes are now described by the theory of elasticity. Let us first derive its form in elementary textbook style [4, 5].

Consider everyday solids formed from an array of atoms, and let us first focus on static elasticity only involving the spatial coordinates of the atoms labeled with latin indices. The point of departure is an equilibrium state where the atoms form a perfect periodic lattice where atom $i$ is localized at a position $\mathrm{R}_i^0$. It requires a finite potential energy to change these positions and this is parametrised in terms of the displacements $\mathbf{u}_i$: $\mathbf{R}_i = \mathbf{R}_i^0 + \mathbf{u}_i$. In the continuum limit $\mathbf{u}_i \to \mathbf{u}(x) = u^a(x)$. Only relative displacements matter and these are captured by the *strains*, corresponding to symmetric rank-2 tensors [4]:

$$w_{ma} = \frac{1}{2}(\partial_m u_a + \partial_a u_m). \quad (6)$$

In leading-order gradient expansion the gradient potential energy becomes,

$$S_{\text{pot}} = \int \mathrm{dt d}^d x \, \left[ -\frac{1}{2} w_{ma} C_{mnab} w_{nb} \right]. \quad (7)$$

The rank-4 tensor $C_{mnab}$ contains the elastic moduli (the coupling constants), with a form imposed by invariance under space group operations as we will discuss in a moment. This is equivalent to the theory of the phase mode of a superfluid $\sim (\partial_\mu \phi)^2$.

It was however already realized in the era of Euler and Lagrange (see e.g. Landau and Lifshitz [4]) that there is a direct geometrical meaning to the theory of elasticity. One imagines an 'internal observer' living in the crystal that measures distances by hopping from lattice site to lattice site; such an observer will experience the

(deformed) crystal as a geometry. For convenience, consider a (hyper)cubic lattice and choose a Cartesian coordinate system with its axes coincident with the cubic lattice directions. The metric of the internal observer measuring distances in the equilibrium crystal is obviously $ds^2 = \delta_{mn}dx_m dx_n$ — flat space. Deforming the metric slightly by $\vec{x'} = \vec{x} + \vec{u}(\vec{x})$ it becomes,

$$(ds')^2 = (dx_m + du_m)^2 \qquad (8)$$

In leading order of the gradients,

$$(ds')^2 = \delta_{mn}dx_m dx_n + 2w_{mn}dx_m dx_n \equiv \mathcal{G}_{mn}dx_m dx_n \qquad (9)$$

with $w_{mn}$ defined in Eq. (6).

This "crystal geometry" is still invariant under coordinate transformations. Every practioner of elasticity knows that it is a good idea to choose coordinates that are suitable for the problem. Dealing with a cube Cartesian coordinates may be most convenient but for a beam one better uses cylinder coordinates. One can look up in the elasticity textbook how to transform the action Eq.(7) from one to the other coordinate system. In this regard elasticity is still invariant under this restricted set of coordinate transformations but as in Yang Mills theory these have the status of "passive" gauge transformations. In direct correspondence with the usual Higgsing of gauge theories, the matter field will however expel the now geometrical "curvature" (it actually includes Cartan torsion) as will become clear very soon.

This is still coincident with the textbook formulation of elasticity – we have just given a geometrical name to the strain fields. There is yet a caveat ; $\mathcal{G}_{mn}$ is *not* referring to the metric of the fundamental space-time (as in Eq. 4) in which the crystal resides – for instance, a neutrino that is not interacting with baryons or electrons will have no clue that this metric exists. However, eventually the atoms will back react on space-time since these are gravitating objects. We have to find out how the crystal geometry imprints on the space-time geometry via this backreaction .

Given that we are now dealing with a dynamical space-time the background metric should have the a-priori freedom to change according to the GR rule book. As for the superfluid phase, the Goldstone fields (the strains $w_{\mu\nu}$) are "automatically" linearized in the flat background. It is now crucial to anticipate that the condensate will expel the spatial curvature. Flatness of the background is enforced and this has the consequence that the strains being infinitesimal fluctuations of the elastic medium pair merely with the *linearized* excitations of the background.

These are the $h_{ij}$ "graviton" degrees of freedom of textbook linearized gravity, $g_{ij} = \eta_{ij} + h_{ij}$ with $\eta_{ij}$ being the Minkowski metric. Given the geometric identification of elasticity in the above, this implies that we have to modify the elasticity action to incorporate the fact that background metric is dynamical,

$$\mathcal{W}_{ma} = w_{ma} + \frac{1}{2}h_{ma},$$

$$S_{\text{pot}} = \int dt d^d x \left[ -\frac{1}{2}\mathcal{W}_{ma}C_{mnab}\mathcal{W}_{nb} \right]. \qquad (10)$$

This simple equation will be the working horse of crystal gravity.

We are now in the position to appreciate the close analogy with the usual Higgs mechanism. The strains $w_{ab}$ take the role of the phase field $\partial_\mu\phi$ while the metric fluctuation $h_{\mu\nu}$ is like the gauge potential $A_\mu$. At distances large compared to the London penetration depth the unitary gauge $\partial_\mu\phi = 0$ is practical, and the Stueckelberg term reduces to $\sim A_\mu^2$: the term giving mass to the photons. In the same way, at distances large compared to $\lambda_G$ one can employ the crystal gravity version of the unitary gauge $w_{ab} = 0$: the crystal lattice is shuffled in the geometry $\sim h_{\mu\nu}C_{\mu\nu\kappa\lambda}h_{\kappa\lambda}$. In this flat background the Hilbert-Einstein action reduces to Fierz-Pauli linearized gravity and one may already envisage that the outcome is that the gravitons will acquire a mass which is indeed the case (section IV). In fact, one may now insist on restoring the full metric $h_{\mu\nu} \to \mathcal{G}_{\mu\nu}$ to recover Eq. (4). But there is no need since the linearized fields suffice to keep track of everything that can happen. As in the Yang-Mills Higgs phase, anything non-linear will turn into topological excitations and the linearized fields suffice for their identification as we will find out.

With Eq. (10) we have identified the fundamental equation governing crystal gravity and the remainder of this paper is dedicated to study its consequences. Notice that in fact he basic conception of frame fixing leading to a Stueckelberg-type action is the same as in the GR tradition dealing with e.g. massive gravity and the various constructions introduced in the holography community [1, 18, 38?, 39]. The difference is that in the above we depart from the tensor structure as is spelled out in elementary elasticity textbooks rooted in the space-group symmetry governing real crystals (the strain tensors), instead of improvised constructions given in by computational convenience. It seems that it has been overlooked just because of unfamiliarity with elementary elasticity theory.

### B. The brutal breaking of Lorentz invariance.

All what remains to do is to find out the form of the $C_{\mu\nu\kappa\lambda}$ tensor that is determined by the residual symmetry after matter has broken translations and rotations – the space group of the crystal. But we have to consider space as well as *time*. As we emphasized in the introduction, the *universal* consequence of crystallization is a *maximal* breaking of Lorentz invariance.

Surely, finite density and/or finite temperature already suffices to destroy the space-time isotropy. We will be

mainly focussed on stationary circumstances where matter is in equilibrium. Consider as an example a finite temperature fluid under such circumstances. One can then safely Wick rotate to Euclidean signature and Lorentz invariance is broken by the fact that imaginary time in a circle with radius $\hbar/k_B T$: at long times it is inequivalent to the non-compact space directions and this will be remembered in Lorentzian signature.

Consider now a crystal at zero temperature in Euclidean signature. Space and time seem to be a-priori equivalent and one could now contemplate a crystal that is isotropic in Euclidean space-time, a "world crystal" [22] that is respecting Lorentz-invariant modulo the frame fixing. At first sight this looks reasonable – see for instance the quantum liquid crystal [24–26] that is asserted to be "Lorentz-invariant" in this sense [23]. Among the peculiarities is that the sound mode of the non-relativisic nematic fluid becomes massive. However, to arrive at such a world crystal one has to break time translations. This seems unproblematic dealing with an Euclidean time axis at zero temperature. A similar line of reasoning lead Wilczek to postulate the existence of a time crystal characterized by a spontaneous breaking of Euclidean time translations [32].

However, physical time is characterized by Lorentzian signature and breaking Lorentzian time translational invariance means sacrificing unitarity. Soon after Wilczek launched his idea it was demonstrated that such a time crystal cannot exist as equilibrium state [33]. One has to resort to driven systems characterized by special conditions (e.g. many body localization) to suppress quantum thermalization.

For these fundamental reasons it is therefore impossible to realize such isotropic space-time crystals. A physical crystal viewed in Euclidean space-time is therefore *maximally* anisotropic, characterized by translational symmetry breaking in space directions while the time direction is entirely translational invariant. This is similar to the classical smectic liquid crystalline order (solid-like in one direction, liquid in the perpendicular plane) but actually even more anisotropic. The "classical" Euclidean signature analogue in a first quantized representation in 2+1D is like an Abrikosov vortex lattice formed from *incompressible lines* forming a regular array in the "space-like" plane perpendicular to their "time-like" propagation direction, see e.g. [34].

In terms of the effective theory, the crucial ingredient is that worldlines cannot be compressed in the time direction with the implication that the time-like displacement vanishes : $u_\tau = 0$. This means that strain fields are strictly transversal to the time direction, $w_{\tau\tau} = 0$ and only $w_{\tau,i} \to \partial_\tau u_i \neq 0$. We recognize the velocity and we conclude conclusion that in the fixed frame action in so far the time-axis is involved this amounts to a simple kinetic term, in Lorentzian signature

$$\mathcal{L}_{\text{kin}} = -\frac{1}{2}\rho(\partial_t u_a)(\partial^t u_a) = -\frac{1}{2}\rho(\partial_t u^a)^2, \qquad (11)$$

where $\rho$ is the mass density. The demise of Lorentz invariance has the effect that the "temporal strain $w_{\tau,i}$" is no longer a symmetric tensor. This has a detrimental consequence for the economy of the formalism. The symmetric nature of the elasticity tensors is at the heart of the harmonious marriage with GR, as we will exploit in the remainder dealing with stationary circumstances. We found out how to deal in principle with the dynamics, in the context of the (non-gravitational) duality constructions [25, 26] but the formalism is cumbersome and hazardous in combination with a dynamical background. This is the main reason for us to shy away from dynamical aspects in this exposition. We will shortly touch on this again in Section IV G.

But even in stationary circumstances this maximal breaking of Lorentz invariance has unusual- and counter-intuitive consequences that become particularly manifest in the combination with gravity as will be highlighted in the later sections.

## C.  Isotropic elasticity and the shear modulus.

All what remains to be done is to determine the form of the all-spatial tensor $C_{mnab}$ by imposing invariance under the space group transformations. The outcomes are tabulated (see e.g. chapter 1 of ref. [5]). The reference point is the theory of *isotropic* elasticity which is the standard in engineering books. Single crystals that show the ramifications of the full space group on the macroscopic scale such as crystal faces are actually very rare. Solids are typically infused with all kinds of defects with the outcome that their macroscopic elastic properties are effective isotropic: these are governed by the $SO(3)$ group of *global* spatial rotations.

Relative to the isotropic case, the discrete point group operations translate in "anisotropy moduli". With the exception of extreme cases like the van der Waals solids (such as graphite) the anisotropy corrections are usually small as compared to the shear- and compression moduli of the isotropic limit. We will ignore these anisotropic moduli for no other reason than that these are rather non-consequential in the crystal gravity, while having the effect of rendering the theory to become more laborious and less transparent. Notice that at the moment we start to deal with the topological excitations the space group data become crucial because these determine the topological charges. But this is easy to restore departing from the isotropic theory.

As in linearized gravity we take a flat background and a co-moving frame encoding for the frame where the action can be decomposed in terms of spatial $s = 0, 1, 2$ angular momentum contributions. Given the breaking of Lorentz invariance this is now the only natural frame. Introduce projectors $P^{(s)}_{mnab}$ on the space of $(0,2)$ tensors under $SO(d)$ rotations where $d$ refers to the dimensionality of the spatial manifold [5, 25, 26]. In Cartesian coordinates,

$$P_{mnab}^{(0)} = \frac{1}{d}\delta_{ma}\delta_{nb}$$

$$P_{mnab}^{(1)} = \frac{1}{2}(\delta_{mn}\delta_{ab} - \delta_{mb}\delta_{na})$$

$$P_{mnab}^{(2)} = \frac{1}{2}(\delta_{mn}\delta_{ab} + \delta_{mb}\delta_{na}) - \frac{1}{d}\delta_{ma}\delta_{nb} \qquad (12)$$

Since local rotations cannot change the potential energy of the crystal $\mathcal{L}_{\text{pot}}$ does *not* contain the spin-1 components and therefore the action of an isotropic solid is determined by:

$$C_{mnab} = d\kappa P_{mnab}^{(0)} + 2\mu P_{mnab}^{(2)} \qquad (13)$$

The quantities $\kappa$ and $\mu$ are the compression- and shear modulus, respectively, that completely specify the static responses of the isotropic solid. Notice that the compression modulus multiplies the trace while shear is traceless. The particular way in which this will talk to the gravitons is already shimmering through: this has to do with the spin 2 shear sector.

The potential energy density of the isotropic solid can be written explicitly in $d$ space dimensions in terms of Cartesian coordinates as,

$$\mathcal{L}_{\text{pot}} = -\frac{1}{2}\left(2\mu(w_{ma})^2 + \lambda(w_{mm})^2\right), \qquad (14)$$

in terms of the Lamé coefficient $\lambda = \kappa$. Another useful parameter is the Poisson ratio $\nu$ defined as

$$\kappa = \frac{2\mu}{d}\frac{1+\nu}{1-(d-1)\nu}. \qquad (15)$$

This is the classical elasticity theory derived in the 19th century, describing the *static* responses of the isotropic elastic medium. By adding the kinetic term Eq. (11) elasticity acquires the status of semiclassical quantum field theory [25, 26] with a quantum partition sum in Euclidean signature ($\tau = \text{i}t$ is imaginary time),

$$Z_{\text{elas}} = \int \mathcal{D}u^a \, e^{-\frac{S_{\text{E,elas}}}{\hbar}},$$

$$S_{\text{E,elas}} = \int \text{d}\tau \text{d}^d x \, \mathcal{L}_{\text{E,elas}},$$

$$\mathcal{L}_{\text{E,elas}} = \mathcal{L}_{\text{E,kin}} + \mathcal{L}_{\text{E,pot}},$$

$$\mathcal{L}_{\text{E,kin}} = \frac{1}{2}\rho(\partial_\tau u^a)^2,$$

$$\mathcal{L}_{\text{E,pot}} = \frac{1}{2}\left(2\mu(w_{ma})^2 + \lambda(w_{mm})^2\right), \qquad (16)$$

describing among others the acoustic phonons in the guise of the solid-state textbook harmonic solid. Of crucial importance in the context of gravity, the bad breaking of Lorentz invariance has as consequence that the transversal phonons are *spin-1* (helical $(2,\pm1)$ while the longitudinal phonon is spin 0 (mixture of $(0,0)$ and $(2,0)$) under spatial rotations, rooted in the temporal derivatives $\partial_t u^a$.

To make this explicit, let us compute the displacement-displacement propagators associated with the phonons. Fourier transform to frequency-momentum space is indicated by $\mathbf{q}$ with magnitude $q$ for spatial momenta and Matsubara frequencies $\omega_n$. The equations of motions are trivial to solve for this linear problem and it follows,

$$\langle u_a \, u_b \rangle = \frac{1}{\rho}\left[\frac{P_{ab}^{\text{L}}}{\omega_n^2 + c_{\text{L}}^2 q^2} + \frac{P_{ab}^{\text{T}}}{\omega_n^2 + c_{\text{T}}^2 q^2}\right]. \qquad (17)$$

defining the longitudinal and transverse projectors $P_{ab}^{\text{L}} = q_a q_b/q^2$, $P_{ab}^{\text{T}} = \delta_{ab} - P_{ab}^{\text{L}}$. The longitudinal- and transverse velocities are,

$$c_{\text{L}} = \sqrt{\frac{\kappa + 2\frac{d-1}{d}\mu}{\rho}} = \sqrt{\frac{2\mu}{\rho}\frac{1-(d-2)\nu}{1-(d-1)\nu}}, \qquad (18)$$

$$c_{\text{T}} = \sqrt{\frac{\mu}{\rho}}. \qquad (19)$$

We infer one longitudinal acoustic phonon with velocity $c_{\text{L}}$ and $d-1$ transverse acoustic phonons with velocities $c_{\text{T}}$, the Goldstone modes associated with the spontaneous breaking of $d$ spatial translational symmetries.

## III.   INTERMEZZO: ABELIAN-HIGGS DUALITY, THE MASS OF THE PHOTON AND THE FLUXOID.

As we announced in the introduction we are much helped by the circumstance that all the mathematical machinery we need was collected and perfected by Kleinert in his 1980's book [5]. Eventually this revolves around the realization that the analogy between the Abelian Higgs problem and the much richer crystal gravity continues to hold on a much deeper level than what we just discussed. The key is that the general structure of the weak-strong "vortex-boson" (or "particle-vortex", "Abelian-Higgs") duality applying to the former generalizes straighforwardly to the latter. The 2+1D version as formulated in the late 1970's [35] is a well known affair. We will present in this section a short tutorial of the vortex-boson duality, as a template for the generalization to crystal gravity (for recent results in 3+1D, see ref. [36]). This is standard material and the reader who is familiar with this machinery can safely skip this section altogether.

The summary of how it works is as follows. In a first step the Goldstone (phase) modes of the superfluid are dualized in the conserved supercurrents that are subsequently recognized as gauge fields that propagate forces. These are sourced by "multivalued gauge field" configurations that are recognized as the topological excitations of the superfluid, the vortices. One ends up with

the dual theory written in terms of the vortex degrees of freedom having a long range interaction that at least in 2+1D is identical to QED. The same dualization strategy is equally powerful when unleashed on the much richer crystal gravity. The outcome is of the same kind, in the form of an effective theory encapsulated entirely in terms of the topological defects, now of the crystal, interacting through "gauge" fields that morph naturally with the fields of GR. In III A the dualization procedure is explained, while in III B-III E we highlight the various tricks employed to get an answer to specific questions that we will meet later.

## A. The superconductor in dual representation.

The central wheel in the field-theoretical vortex–boson duality (or Abelian-Higgs duality) is just stating that the (Abelian) superfluid in 2+1 dimensions is the Kramers–Wannier (weak–strong) dual of the (gauged) superconductor. The superfluid may be viewed as the Coulomb phase of the charged dual, where the superfluid vortices take the role of the charged matter. Similarly, the superconductor (Higgs phase) may be viewed as the dual of a superfluid where the fluxoids act as neutral bosons that upon condensation form the neutral superfluid. This 'direction' of the duality is the one that relates to crystal-gravity.

Let us consider the $U(1)$ Stueckelberg action, Eq. (5) in $d = 3$ overall Euclidean dimensions. This may be viewed as the relativistic theory in 2+1D with Euclidean signature. This is in turn equivalent to the classical theory dealing with the static responses in three space dimensions. It relates to crystal gravity in 3+1 dimensions in so far as static aspects of the elastic medium are at stake.

The conventional way to proceed is to choose a unitary gauge fix, $\partial_\mu \phi = 0$ and one reads off immediately that the photons acquire a mass $m_H^2 \propto |\Psi|^2$. However, using the dualization technology it becomes easy to enumerate all the consequences of the theory. For convenience we will use here a short-hand notation, by suppressing all dimensionful quantities, taking also the Higgs mass $|\Psi|^2 = 1$. In addition, we suppress the integral signs and keep it implicit that the action appears in the path integral. Any reference to "integrating out" refers to standard Gaussian integrations in the path integral. We consider the relativistic case characterized by a single velocity of light both for matter and the gauge field.

In order to be able to write all terms with spacelike indices and the same sign, the temporal components need to be redefined by a factor of i, but we leave this here implicit since this section is primarily heuristic(for details see Ref. [25]). The phase (Josephson-, Stueckelberg) action in Euclidean signature then becomes,

$$\mathcal{L}_{\text{E,phase}} = \frac{1}{2}(\partial_\mu \phi - A_\mu)^2 + \frac{1}{4}F_{\mu\nu}F_{\mu\nu}; \quad \phi = \phi + 2\pi, \quad (20)$$

where the mod $(2\pi)$ refers to the fact that the phase field is compact. The first step is to dualize by Legendre transformation, which can be accomplished by introducing the auxiliary field $j_\mu$ such that Eq. (20) can be rewritten as,

$$\mathcal{L}_{\text{E,dual}} = \frac{1}{2}j_\mu j_\mu + i j_\mu (\partial_\mu \phi - A_\mu) + \frac{1}{4}F_{\mu\nu}F_{\mu\nu}. \quad (21)$$

According to the standard Hubbard–Stratonovich Gaussian integration identity this reduces to Eq. (20) by integrating out $j_\mu$.

The configurations of the phase field can now be decomposed in smooth and multivalued pieces $\phi = \phi_{\text{sm}} + \phi_{\text{MV}}$, where the latter is defined as the *non-integrable*, singular part due to the compactness of the matter field. We will see in a moment that this relates to the vortices.

The smooth part is integrable and this implies that $j_\mu \partial_\mu \phi_{\text{sm}} = -\phi_{\text{sm}} \partial_\mu j_\mu$ modulo a total derivative. Then $\phi_{\text{sm}}$ acts as a Lagrange multiplier that can be integrated out, imposing the conservation law

$$\partial_\mu j_\mu = 0 \quad (22)$$

This is just the continuity equation governing the conservation of the supercurrent $j_\mu$. In the absence of singular ("multivalued") configurations the action in terms of the current variables can be written as,

$$\mathcal{L}_{\text{E,dual}} = \frac{1}{2}j_\mu j_\mu - i j_\mu A_\mu + \frac{1}{4}F_{\mu\nu}F_{\mu\nu}$$
$$\partial_\mu j_\mu = 0 \quad (23)$$

Now comes the magic: the conservation law Eq. (22) can be imposed in 3 dimensions by expressing the current in terms of a non-compact $U(1)$ 1-form gauge field $b_\mu$,

$$j_\mu = \varepsilon_{\mu\nu\lambda}\partial_\nu b_\lambda \quad (24)$$

and it follows that the kinetic energy of the super current $j_\mu j_\mu = f_{\mu\nu} f_{\mu\nu}$, where $f$ is the field strength of $b$: $f_{\mu\nu} = \partial_\mu b_\nu - \partial_\nu b_\mu$. It is equivalent to Maxwell electromagnetism: one may identify the phase mode of the superfluid with the single propagating photon of 2+1D Maxwell theory!

But we are not done yet since the multivalued part of the phase field has still to be addressed. This is the non-integrable part: $j_\mu \partial_\mu \phi_{\text{MV}} = \varepsilon_{\mu\nu\lambda}\partial_\nu b_\lambda \partial_\mu \phi_{\text{MV}} = b_\mu \varepsilon_{\mu\nu\lambda}\partial_\nu \partial_\lambda \phi_{\text{MV}}$. This implies,

$$\mathcal{L}_{\text{E,dual}} = \frac{1}{4}F_{\mu\nu}F_{\mu\nu} + \frac{1}{4}f_{\mu\nu}f_{\mu\nu} + i b_\mu J_\mu^{\text{V}} - i A_\mu \varepsilon_{\mu\nu\lambda}\partial_\nu b_\lambda,$$
$$J_\mu^{\text{V}} = \varepsilon_{\mu\nu\lambda}\partial_\nu\partial_\lambda \phi_{\text{MV}}. \quad (25)$$

By Stokes theorem it follows that $J^{\text{V}}$ represent the vortex currents, being lines in 3 Euclidean dimensions,

corresponding to the wordlines of 'vortex particles' in the 2+1D quantum theory. To see this explicitly, depart from the familiar statement defining the winding number $N$ of the vortex,

$$\oint_C d\phi(x) = \oint_C dx_\mu \partial_\mu \phi(x) = 2\pi N. \tag{26}$$

Convert this using Stokes' theorem into a surface integral of the curl of the integrand over the surface $S$ enclosed by $C$,

$$\int_S dS_\lambda \, \varepsilon_{\lambda\nu\mu} \partial_\nu \partial_\mu \phi(x) = 2\pi N \tag{27}$$

which is satisfied when

$$J_\lambda^V(x) = \varepsilon_{\lambda\nu\mu} \partial_\nu \partial_\mu \phi(x) = 2\pi N \delta_\lambda^{(2)}(L, x). \tag{28}$$

Here $\delta_\lambda^{(2)}(x)$ is a two dimensional delta function in the plane orthogonal to $\lambda$, where we use he definition of the delta function on the defect line $L$ parametrized by $s$, given by

$$\delta_\lambda^{(2)}(L, \vec{x}) = \int_L ds \, \partial_s x_k^L(s) \delta^{(d)}\big(x - x_k^L(s)\big). \tag{29}$$

which is non-zero at the origin – if the vortex is not centered at the origin, the argument of the delta function is shifted. This establishes $J^V$ as the (conserved) topological (vortex) current related to the non-integrable part of the phase field $\phi$.

Here is the benefit of this dualization procedure: all the non-linearities associated with the physics of the ordered state are encapsulated by the topological excitations. Departing from the linear Goldstone sector this dualization procedure captures the topological sector automatically by identifying multi-valued order parameter configurations. Remarkably, gauge theory arises as the natural mathematical language. This dualization principle overrules all the differences between simple $U(1)$ Abelian Higgs and crystal gravity as we will see in the remainder.

Summarizing, in this dual description the neutral superfluid is described by 2+1D Maxwell theory, where the vortices take the role of 'charges' that interact with each other via 'photons' $b_\mu$ that represent the induced supercurrents. In superconductors these 'current photons' are coupled to the physical photon gauge field $A_\mu$ via the BF-term $\sim A\varepsilon\partial b$. This is the blueprint of the machinery that we will use as well in crystal gravity. By integrating out either the $A_\mu$ or $b_\mu$ fields in the presence or absence of vortex sources one can compute effortlessly anything of physical interest happening in the superconductor.

Let us now turn to the benefits of the helical representation. This is explicitly constructed in Appendix B: it departs from a coordinate system associated with the longitudinal- and transverse directions relative to spacetime momentum. The helical representation then arises by considering the eigenvectors in momentum space under rotations. In three dimensions, these are in the $s = 1$ representation with eigenvalues $h = -1, 0, +1$, where 0 is the longitudinal direction. In the gauge theory, the longitudinal components $A^{(0)}$, $b^{(0)}$ do not contribute to their respective field strengths $F_{\mu\nu}$ and $j_\mu \sim f_{\mu\nu}$ since these are pure gauge. In the helical representation these components are thereby automatically vanishing, and all that remains are the physical photons $A^{(\pm 1)}$ and 'current photons' $b^{(\pm 1)}$. Such a decomposition will become particularly convenient in the context of the rank-2 tensors of elasticity.

To illustrate how this works, let us decompose our dual BF action in helical components:

$$\mathcal{L}_{E,dual} = \sum_{h=\pm 1} \frac{1}{2} \left( p^2 |A^{(h)}|^2 + p^2 |b^{(h)}|^2 \right) + ip(b^{(-1)\dagger}A^{(-1)} - b^{(+1)\dagger}A^{+1}) + i(b^{(+1)\dagger}J^{V(+1)} + b^{(-1)\dagger}J^{V(-1)}). \tag{30}$$

Here $p$ is the magnitude of the spacetime momentum $p_\mu = (\frac{1}{c}\omega_n, \mathbf{q})$, where $c$ is the velocity of light. Furthermore $b^{+1\dagger}A^{+1}$ is a short-hand for $\frac{1}{2}(b^{+1\dagger}A^{+1} + A^{+1\dagger}b^{+1})$ etc.

This is a highly transparent expression: Eq. (30) corresponds to a simple linear mode coupling problem in terms of the physical, gauge-invariant fields. The operations that we will use in crystal gravity are straightforward generalizations of this superconductivity affair.

## B. The Higgs mechanism.

In the helical representation of the dual superconductor, the mechanism that gives photons a mass becomes particularly transparent. The crucial step is the dualization of the Goldstone variable (the phase mode) into the supercurrent, and the current photons $b_\mu$. It is elementary that the EM gauge fields $A_\mu$ are associated with the propagation of *forces*. By parametrizing the supercur-

rents (associated with momentum) in terms of the 'auxiliary' gauge fields $b_\mu$, one isolates similarly the capacity of the superfluid to propagate *forces* given its emergent phase rigidity. In terms of this force representation one compares apples with apples, and the outcome is that the superfluid and EM gauge bosons are subjected to a simple linear mode coupling which in turn takes care of the emergence of the Higgs mass as we will now show.

Take Eq. (30) without vortex current $J^V$, and integrate out the $b_\mu$ field to obtain,

$$\mathcal{L}_{\text{E,Higgs,EM}} = \sum_{h=\pm 1} \frac{1}{2c^2}(\omega_n^2 + c^2q^2 + m_{\text{H}}^2)|A^{(h)}|^2, \quad \text{or}$$

$$\mathcal{L}_{\text{Higgs,EM}} = \sum_{h=\pm 1} \frac{1}{2c^2}(\omega^2 - c^2q^2 - m_{\text{H}}^2)|A^{(h)}|^2. \quad (31)$$

In the second line we have analytically continued to real time/frequency to arrive at the standard form of the massive photon propagator. The Higgs mass in these dimensionless units is 1, but it is proportional to the superfluid density. Expressed in explicit units, $m_{\text{H}} = c/\lambda_{\text{L}}$ with $\lambda_{\text{L}}$ the ordinary London penetration depth. This Lagrangian describes the massive photons.

Alternatively, we may integrate out the EM fields from Eq. (30) to find

$$\mathcal{L}_{\text{Higgs,SC}} = \sum_{h=\pm 1} \frac{1}{2c^2}(\omega^2 - c^2q^2 - m_{\text{H}}^2)|b^{(h)}|^2 \quad (32)$$

Compared to the neutral superfluid with a single, massless Goldstone mode described by a free vector gauge field $b_\mu$, now two massive degrees freedom arise that can be represented either by the photon field $A_\mu$ or by the current gauge field $b_\mu$. In 2+1D the EM vacuum has one physical photon polarization, but by coupling to the superconducting condensate the longitudinal photon becomes a second propagating degree of freedom due to the Anderson–Higgs mechanism resulting in Eq. (31). This is consistent with the standard argument employing the unitary gauge.

But we can also represent the same degrees of freedom by the $b_\mu$ field. The transverse polarization of this field represents the original Goldstone mode, picking up a mass when coupled to electromagnetism. In a neutral superfluid, the longitudinal part of $b_\mu$ represents the static (non-propagating) Coulomb interaction between vortex particles. In the helical representation it becomes transparent that due to the coupling to the EM field it turns into a propagating, massive degree of freedom. It is even possible to integrate out the longitudinal parts of $A_\mu$ and $b_\mu$, keeping track of the transverse polarizations of both. The remaining $A$- and $b$-component now represent the massive photon and Goldstone mode, respectively.

## C. Interactions between vortices.

Vortices in a neutral superfluid interact with an effective Coulomb interaction. The circulating currents associated with a vortex fall off algebraically causing a repulsion (attraction) between vortices with the same (opposite) sense of circulation. This can be directly read off from Eq. (25): suppress the EM fields $A_\mu$ and it reduces to a Maxwell action for current photons $b_\mu$ and vortex charges $J_\mu^V$. Upon adding back the EM field, this interaction becomes screened on the scale of the London length, and this is easy to compute in the dual representation. After integrating out $A_\mu$, we have Eq. (32) supplemented with vortex currents as sources,

$$\mathcal{L}_{\text{E,vortex,SC}} = \sum_{h=\pm 1} \left( \frac{1}{2}(p^2 + \frac{1}{\lambda_{\text{L}}^2})|b^{(h)}|^2 + ib^{(h)\dagger}J^{V(h)} \right). \quad (33)$$

Here $\lambda_{\text{L}}$ is (again) the London penetration depth inversely proportional to the Higgs mass. Let us interpret this in a static 3D space interpretation where $p_\mu \rightarrow \mathbf{p}$ is the 3D spatial momentum and $J^V$ represents static vortex lines. Upon integrating out the $b_\mu$ field,

$$\mathcal{L}_{\text{E,vortex,SC}} = \sum_{h=\pm 1} \frac{1}{2} J^{V(h)\dagger} \frac{1}{\mathbf{p}^2 + \frac{1}{\lambda_{\text{L}}^2}} J^{V(h)}. \quad (34)$$

The vortex correlation function is of the Yukawa form, with the familiar result in real space,

$$\int \frac{d^3p}{(2\pi)^3} \frac{1}{\mathbf{p}^2 + \frac{1}{\lambda_L^2}} e^{i\mathbf{p}\cdot(\mathbf{x}-\mathbf{x}')} = \frac{1}{4\pi|\mathbf{x}-\mathbf{x}'|}e^{-\frac{|\mathbf{x}-\mathbf{x}'|}{\lambda_{\text{L}}}}. \quad (35)$$

We see that the vortex lines interact via a $e^{-r/\lambda_{\text{L}}}/r$ potential; the Coulomb interaction of the superfluid is screened on the scale of the London length in the superconductor. This result can also be interpreted in two spatial dimensions upon taking the static limit $\omega \rightarrow 0$; the vortex "particle" static interaction acquires the form of a Bessel function that for large $r$ approaches $\exp(-r/\lambda_{\text{L}})/\sqrt{r/\lambda_{\text{L}}}$.

## D. Fluxoids in the charged superconductor.

A complementary way of viewing the vortex in the superconductor is by focussing on the electromagnetic field $A_\mu$ to find out how this is sourced by a vortex $J_\mu^V$. This will reconstruct the Abrikosov vortex/fluxoid: the line-like topological defect in a superconductor corresponds with magnetic field lines combined with screening currents localized within a radius $\sim \lambda_{\text{L}}$ carrying an overall topological quantum of magnetic flux $\Phi_0 = h/e^*$, $e^*$ being the microscopic charge quantum ($2e$ for ordinary Cooper pairs).

We will present here a less familiar way to understand how this fluxoid forms; this view will turn out to be quite informative dealing with the "confining" geometric curvature fluxoids discussed in section VI.

Depart from the action Eq. (25). Modulo a total derivative the BF term can be as well written in terms of the combination of the supercurrent gauge field and the electromagnetic field strength, $A_\mu \varepsilon_{\mu\nu\lambda} \partial_\nu b_\lambda \to b_\mu \varepsilon_{\mu\nu\lambda} \partial_\nu A_\lambda$. The action becomes,

$$\mathcal{L}_{\text{E,dual}} = \frac{1}{4} F_{\mu\nu} F_{\mu\nu} + \frac{1}{4} f_{\mu\nu} f_{\mu\nu} + \mathrm{i} b_\mu \left( J_\mu^{\text{V}} - \varepsilon_{\mu\nu\lambda} \partial_\nu A_\lambda \right). \tag{36}$$

The glueing of flux and vorticity is a static affair and therefore interpret this as the theory in three space dimensions: $\varepsilon_{\mu\nu\lambda} \partial_\nu A_\lambda \to \nabla \times \mathbf{A} = \mathbf{B}$ is the magnetic field. Integrate out the supercurrent gauge field $\mathbf{b}$ and express the outcome in terms of the magnetic field strength and the density of vortex lines in space $\mathbf{J}^{\text{V}}(\mathbf{x})$) as,

$$\mathcal{L}_{\text{E,fluxoid}} = \frac{1}{2} \mathbf{B} \cdot \mathbf{B} - \frac{1}{2} (\mathbf{J}^{\text{V}} - \mathbf{B}) \frac{1}{\lambda_{\text{L}}^2 \nabla^2} (\mathbf{J}^{\text{V}} - \mathbf{B}), \tag{37}$$

noticing that $\nabla \cdot \mathbf{B} = 0$ and $\nabla \cdot \mathbf{J}^{\text{V}} = 0$, while $\frac{1}{\nabla^2}$ is a short hand for the (unscreened) vortex-Coulomb interaction. By varying the action with respect to the vortex current $\mathbf{J}^{\text{V}}$ it follows immediately that

$$\mathbf{J}^{\text{V}} = \mathbf{B}. \tag{38}$$

This implies that the fluxoid carries the magnetic flux quantum,

$$2\pi N = \int_S dS_k J_j^{\text{V}} = \int_S dS_k B_k = \Phi_B \tag{39}$$

where $\Phi_B$ is the magnetic flux through $S$ in natural units.

This is of course familiar, but Eq. (37) reveals a less familiar wisdom. What is the "force" glueing precisely the right amount of quantized flux to the circulating current? One reads this off Eq. (37): it is a Coulomb potential, when vortex current and magnetic field do not compensate one runs into a Coulomb catastrophy associated with the incompatible response of the supercurrents.

In order to determine how this flux is distributed in space vary Eq. (37) to the magnetic field to obtain the equation of motion,

$$-\lambda_{\text{L}}^2 \nabla^2 B_k + B_k = J_k^{\text{V}} = \frac{h}{e^*} N \delta_k^{(2)}(\vec{x}) \tag{40}$$

where we restored the units of (effective) electrical charge $e^*$ and $\hbar$ and used Eq. (28). This is precisely the textbook equation for the fluxoid: the solution is $B(r) = \frac{\hbar N}{e^* \lambda_{\text{L}}^2} K_0(r/\lambda_{\text{L}})$, where $r$ is the radial coordinate and $K_0$ is a modified Bessel function, which falls off as $\exp(-r/\lambda_{\text{L}})$ for large $r$ [2].

For future use it is instructive to see how this works in the helical representation. Integrate out the $b_\mu$ field from Eq. (30), in the static interpretation of Eq. (33),

$$\mathcal{L}_{\text{E,fluxoid}} = \sum_{h=\pm 1} \left( (p^2 + \frac{1}{\lambda_{\text{L}}^2}) |A^{(h)}|^2 + \frac{1}{\lambda_{\text{L}}^2 p^2} |J^{\text{V}(h)}|^2 \right) + \mathrm{i} \frac{1}{\lambda_{\text{L}}^2 p} (A^{-1\dagger} J^{\text{V}-1} - A^{+1\dagger} J^{\text{V}+1}) \tag{41}$$

The equation of motion obtained follows from varying to $A^{(h)\dagger}$,

$$(\lambda_{\text{L}}^2 p^2 + 1) p A^{\pm 1} = \mp J^{\text{V}\pm 1} \tag{42}$$

In this 3D static setting the gauge fields are entirely associated with the magnetic fields $B_k = \varepsilon_{klm} \partial_l A_m$. Substituting $B^{\pm 1} = \mp p A^{\pm 1}$ and $p \to -\mathrm{i}\nabla$, one recognizes that this is the same equation as Eq. (40).

### E. Fluxoids in 3+1 dimensions.

One better be aware that the perfect match between Maxwell theory and the vortex dual is special to 2+1D, or equivalently the static physics in 3D space. However, in 3+1D the analogy is severed and instead the vortex dual now involves *2-form* or *Kalb–Ramond* gauge fields. The

reason is that the vortices are lines in 3D, turning into Nielsen–Olesen *strings* in 3+1D. This is also the case for the dislocations and disclinations associated with elasticity, and this foreshadows intricacies with the formulation of dynamical crystal gravity in 3+1D space-time. Although manegable, it involves harder work to match the tensor structure of GR and the topological currents.

The 3+1D generalization of the vortex–boson duality for superconductors is enumerated in detail in Ref. [36]. Let us only highlight here the main differences with the 3+0D case. Instead of the parametrization Eq. (24), in 3+1D the conservation law is in imposed by,

$$j_\mu = \varepsilon_{\mu\nu\kappa\lambda} \partial_\nu b_{\kappa\lambda} \tag{43}$$

where $b$ is now an (antisymmetric) 2-form gauge field. The supercurrent kinetic energy is associated with the 2-form field strength $j_\mu^2 \to h_{\mu\kappa\lambda}$ with $h_{\mu\kappa\lambda} = \partial_{[\mu} b_{\kappa\lambda]}$),

while the gauge field is sourced by 2-form vortex currents $\mathcal{L}_{\text{int}} = ib_{\kappa\lambda}J_{\kappa\lambda}^{\text{V}}$ where

$$J_{\kappa\lambda}^{\text{V}} = \varepsilon_{\kappa\lambda\mu\nu}\partial_\mu\partial_\nu\phi_{\text{MV}} \tag{44}$$

In 3+1D the vortices become *strings* and this 2-form is just the natural way to parametrize their worldsheets. This can be alternatively written as

$$J_{\kappa\lambda}^{\text{V}}(x) = 2\pi N\delta_{\kappa\lambda}^{(2)}(x), \tag{45}$$

in analogy with Eq. (28) where the delta function now refers to an infinitesimal world sheet element. The 2-form gauge fields acquire a BF type mode coupling with the 1-form EM fields $\sim b_{\kappa\lambda}\epsilon_{\kappa\lambda\mu\nu}\partial_\mu A_\nu$ and the remainder can be enumerated completely using similar procedures as in 3D.

## IV. LINEARIZED CRYSTAL GRAVITY: GRAVITONS AND STATIC SHEAR STRESS.

All the pieces are now lying ready for the first stage of the evaluation of the theory of crystal gravity. This first step deals with the fully *linearized* sector. As we argued in section II A the crystal will force in a locally flat co-moving frame, while the linearized nature of the "Goldstone" strain fields $w_{mn}$ leads naturally to the "first law" Eq. (10) expressing that these "pair" with the infinite metric fluctuations $h_{nm}$ being coincident with the text book gravitational waves. For the reasons explained in section II B we depart from isotropic elasticity Eq. (14). We will ignore for the time being the kinetic term Eq. (11) (see section IV G) for the perhaps counterintuitive reason that it *does not play* any role in the linearized theory. The alert reader may have already anticipated that the propagating phonons involve spatial spin 1, while the coupling to the background is in the spin 2 sector encapsulated by the static part of elasticity.

Hence, the point of departure is the part of the action associated with the rigidity of the solid medium,

$$S_{\text{stat}} = \int \mathrm{d}t\mathrm{d}^3x \left[-\mu W_{ma}W_{ma} - \frac{\lambda}{2}(W_{mm})^2\right] + S_{\text{EH}},$$

$$W_{ma} = w_{ma} + \frac{1}{2}h_{ma}. \tag{46}$$

It should be obvious that this is in essence nothing more than the rank 2 symmetric generalization of the phase action Eq. (20) that formed the input for the dualization procedure highlighted in Section III A. This is the key insight behind Kleinert's "single curl gauge field" machinery that we will follow closely in this section. Given that this may be quite unfamilar for some of our readers we will go slow. We will first highlight the dualization procedure that has here the vivid physical interpretation of the classic (in elasticity) stress-strain duality (section IV A). Subsequently we will step back in section IV B to the textbook treatment of gravitational waves highlighting the benefits of the helical decomposition. In section IV C we will introduce Kleinert's stress gauge fields and demonstrate that shear forces are captured by quite literal "shear gravitons". After these preliminaries we will expose in section IV D the gravitational Higgs mechanism that becomes a simple mode coupling affair in this language. We then explore the consequences both for space time and the properties of the solid in section IV E deriving the graviton mass and in section IV F presenting the amusing gravitational hardening effects, respectively. The discussion of the internal topological sources in this "translational sector" will be taken up in sections V and VI.

### A. Stress–strain duality and gravity.

The development will closely follow the "Abelian Higgs" template of the previous section. The first step as explained in Section III A did amount to a Legendre transformation. The gradients of the phase degree of freedom $\partial_\mu\phi$ are transformed in momenta which in turn represent the supercurrent. The conserved supercurrents represent the "force fields" encoding the emergent rigidity associated with the spontaneous breaking of the $U(1)$ symmetry. Exploiting the local constraint in the form of the continuity equations these could then be expressed in terms of $U(1)$ gauge fields. By simple manipulations we derived the dual action Eq. (25) expressing a simple linear mode coupling to the external EM fields thought a BF-term, identifying the vortices as the internal sources for the emergent gauge fields. The equivalents of the latter will be the subject of the next sections and here we will expose only the linearized sector.

Technically, the elastic version is nothing more than the rank-2 tensor generalization of the vector story of the previous section. Physically it is actually much closer to daily experience. In our human existence we never encounter the forces propagated by supercurrents. However, we are surrounded by elastic emergent rigidity. In strain representation elasticity represents that it costs energy to deform a solid medium. But we know that this implies that the medium is "pushing back" when exposed to an external force that is causing this deformation.

The same Legendre transformation turns the strain formulation into one that is exposing the propagation of elastic forces. This is the stress-strain duality that was understood long before superconductivity was discovered. The emergent rigidity associated with solids is the reactive response to shear stress. As for the superfluid these can be captured in the language of gauge fields. The amazing fact highlighted in the Kleinert treatise is that the gauge fields associated with shear are precisely like gravitons. Accordingly, we will find a linear mode coupling in terms of the BF coupling generalized to rank 2

symmetric tensor fields between the "stress" and "gravitational" gravitons. This is then exploited to exploit the portfolio of the physics of the Higgs phase in close analogy with the exposition in the previous section.

Let us first focus on the stress-strain duality. Departing from the strain action Eq. (46) let us execute the Legendre transformation in Hubbard-Stratonovich style. The generalization to tensor fields is straightforward. Introduce an auxiliary tensor field $\sigma_{ma}$ and the dual action becomes,

$$S_{\text{E,stat}} = \int \mathrm{d}\tau d^3x \left[ \sigma_{ma} C^{-1}_{mnab} \sigma_{nb} + i\sigma_{ma}(w_{ma} + \frac{1}{2}h_{ma}) \right] + S_{\text{EH}}, \tag{47}$$

reducing to the strain action Eq. (46) by integrating out the $\sigma_{ma}$ fields under the condition that $C^{-1}$ is the inverse of the elastic tensor, defined explicitly below. We identify $\sigma_{ma}$ as the *stress tensor* capturing the response of the medium to an imposed strain according to the equation of motion,

$$\sigma_{ma} = -i\frac{\partial \mathcal{L}_{\text{E}}}{\partial w_{ma}} - iC_{mnab}w_{nb} \tag{48}$$

ignoring the external stress captured by $h_{ma}$; the factor of $-i$ is due to our conventions associated with imaginary time (as in Eq. (21).

As for the Abelian-Higgs case, we have to distinguish between the smooth (i.e. integrable) and multivalued displacement field $u_a$ configurations given that $u_a$ takes the role of the phase field $\phi$. The topological sources associated with the non-integrable parts will be the subject of the next section and here we focus on the smooth configurations.

Modulo a boundary term we have $\sigma_{ma}w_{ma} = \sigma_{ma}(\partial_m u_a + \partial_a u_m)/2 = -u_a\partial_m\sigma_{ma}$. The smooth displacement field $u_a$ acts as a Lagrange multiplier, and after integrating it out it imposes the Bianchi identity

$$\partial_m \sigma_{ma} = 0. \tag{49}$$

This Bianchi identity has the physical meaning that the overall mechanical stress, including the background 'geometrical forces' associated with $h_{ma}$, is conserved.

Under the condition that the total stress is conserved while topological sources are absent the dual action becomes,

$$S_{\text{E,stat}} = \int \mathrm{d}\tau d^3x \left[ \sigma_{ma} C^{-1}_{mnab} \sigma_{nb} + i\frac{1}{2}\sigma_{ma}h_{ma} \right] + S_{\text{EH}}. \tag{50}$$

This is equivalent to Eq. (23), the working horse in the linearized part of the vortex-boson duality. The stress tensor $\sigma_{ma}$ takes the role of the supercurrent $j_\mu$, the Einstein–Hilbert action the role of Maxwell, while with

regard to the coupling between matter and background the linearized metric fluctuation $h_{ma}$ takes, remarkably, the role of the EM gauge field $A_\mu$. As for the superconductor this amounts to a simple linear mode coupling.

Specializing to the isotropic case, the static elasticity part takes the form in terms of the stress fields,

$$S_{\text{E,Iso}} = \int \mathrm{d}\tau d^3x \frac{1}{4\mu} \left[ \sigma_{ij}\sigma_{ij} - \frac{2\nu}{1+\nu}(\sigma_{ii})^2 \right] \tag{51}$$

Let us now inspect the various pieces in detail.

## B. Linearized gravity in helical projection.

As we discussed at length in section II A, a ramification of the 'frame flattening' imposed by the crystal on the background is that the Minkowski frame is the natural choice for the infinitesimal metric fluctuations $h_{\mu\nu}$ as they appear in Eq. (50): $g_{\mu\nu} = \eta_{\mu\nu} + h_{\mu\nu}$. This is a convenience because this is the same set up as used in textbooks for the elementary derivation of the gravitons [30, 31]. One inserts this metric Ansatz in the Einstein-Hilbert action to obtain the Fierz-Pauli action,

$$S_{\text{FP}} = -\frac{c^4}{64\pi G} \int \mathrm{d}t d^d x \, \eta^{\mu\nu}\partial_\mu h_{\rho\sigma}\partial_\nu h^{\rho\sigma}. \tag{52}$$

this still contains the ten independent (in 3+1D) components of $h_{\mu\nu}$. According to the EOM's (linearized Einstein equations) $\Phi = -h_{00}/2$ is the Newtonian gravitational potential while the spatial trace part $\Psi = -\frac{1}{6}\delta^{ij}h_{ij}$ is also determined by the distribution of rest mass. The vectors $w_i = h_{0i}$ may be of interest in non-stationary geometries since these are sourced by the spin-1 phonons (section IV G) but we leave this for further study. The (linearized) degrees of freedom that are left behind describing the dynamics of space-time itself are then associated with the traceless part of the spatial components $s_{ij} = \frac{1}{2}(h_{ij} - \frac{1}{3}\delta^{kl}h_{kl}\delta_{ij})$. There is still gauge redundancy: only the spatially transverse components are physical in empty space. There are only two of these, the transverse-traceless "$h^{\text{TT}}_{\mu\nu}$" components. The gravitational wave action becomes,

$$S_{\text{GW}} = -\frac{c^4}{16\pi G} \int \mathrm{d}t d^d x \, \eta^{\mu\nu}\partial_\mu s_{ij}\partial_\nu s_{ij}. \tag{53}$$

while the Einstein equations reduce to,

$$G_{ij} = \frac{8\pi G}{c^4} T_{ij},$$
$$G_{ij} = -\eta^{\mu\nu}\partial_\mu\partial_\nu s_{ij}. \tag{54}$$

Specifically, the gravitons are spin-2 relative to spatial rotations, the same spin 2 that is associated with the *shear rigidity* of isotropic elasticity, Eq. (13).

Taking apart the linearized metric in the textbook style as we just described is more argumentative than necessary: especially dealing with tensor fields the helical representation is highly convenient. Although there is some initial overhead it is so much more transparent that it is recommended for the gravity textbook. One will find that $\Phi, \Psi$ are $s = 0$, while the space-time dynamics resides in the $s = 2$ part (see underneath). Technically it revolves around spatial rotations in momentum space as we already illustrated in the context of Abelian-Higgs in Eq.(30); for computational details see Appendix B.

The helical deomposition works in detail as follows. Depart from three mutually orthogonal cartesian directions in 3D momentum space L, R, S with unit vectors $\hat{e}_{\rm L}$, $\hat{e}_{\rm R}$, $\hat{e}_{\rm S}$, where L is parallel to spatial momentum $\mathbf{q}$. The linear combinations $\hat{e}^{\pm 1} = (\mathrm{i}\hat{e}_{\rm R} \pm \hat{e}_{\rm S})/\sqrt{2}$ of the two transverse directions are eigenvectors of the helicity matrix

$$H_{ij} = q_m (S_m)_{ij} = -\mathrm{i} q_m \epsilon_{mij}, \qquad (55)$$

with eigenvalues $\pm 1$, while $\hat{e}^0 = \hat{e}_{\rm L}$ is an eigenvector with eigenvalue 0; these eigenvectors depend on $\mathbf{q}$, obviously. $S_m$ is the usual generator of 3-rotations around the axis $m$. Rank 2-tensors are constructed by taking the tensor product of two of these eigenvectors; a basis of the 9-dimensional space is then given by Clebsch–Gordan decomposition in terms of eigentensors $\hat{e}_{mn}^{(s,h)}$ with $(s,h)$ eigenvalues set by $s = 0, 1, 2$ and $h = -s, \ldots, s$, see Appendix B. A tensor field $t_{mn}$ can then be decomposed as,

$$t_{mn} = \sum_{s,h} \hat{e}_{mn}^{(s,h)} t^{(s,h)}, \qquad (56)$$

with $t^{(s,h)} = \sum_{mn} \hat{e}_{mn}^{(s,h)*} t_{mn}$.

Let us first decompose the graviton action in this helical representation. In the transverse–traceless gauge fix $h_{t\nu} = 0 \; \forall \nu$ and $h_{mm} = 0$. This leaves only the five $s_{mn}$ components, spanning precisely the $s = 2$ subspace in the helical decomposition. The transversality condition $\partial_m s_{mn} = 0$ removes the $s = 2, h = 0, \pm 1$ components, leaving only $s^{2,\pm 2}$. These are recognized as the familiar $+$ and $\times$ graviton polarizations from linearized gravity [30, 31]. In terms of these physical components, the linearized gravity Lagrangian Eq. (53) becomes,

$$\mathcal{L}_{\rm E,GW} = \frac{c^4}{64\pi G}\left(\frac{1}{c^2}\omega_n^2 + q^2\right)(|h^{(2,2)}|^2 + |h^{(2,-2)}|^2), \qquad (57)$$

in Euclidean momentum space $(\omega_n, \mathbf{q})$ where $q = \sqrt{q_x^2 + q_y^2 + q_z^2}$.

## C.  The stress action and the spin 2 shear gravitons.

Let us now return to the matter part of the linearized crystal-gravity action Eq. (50). Here we follow Kleinert's derivation closely as presented in chapter 4 of his book [5], revolving around the introduction of stress gauge fields and the helical projections, eventually leading to the simple result Eq. (64) that was obtained by Kleinert (his Eq. 4.113).

The inverse $C_{mnab}^{-1}$ of the elastic tensor $C_{mnab}$ in Eq. (13) is defined by $C_{mkac}^{-1} C_{kncb} = \delta_{mn}\delta_{ab}$. The dual stress action Eq. (50) is therefore governed by the helical decomposition,

$$C_{mnab}^{-1} = \frac{1}{d\kappa} P_{mnab}^{(0)} + \frac{1}{2\mu} P_{mnab}^{(2)}. \qquad (58)$$

From this very definition of isotropic elasticity one already infers directly that the spatial scalar (trace part, $s = 0$) is associated with standard pressure (compressibility in elasticity), having the same role as in fluids. The difference is in the spin-2 part, which is now governed by a reactive response quantified by the shear modulus $\mu$. One anticipates that this shear modulus will take the role of the superfluid density (the $m_{\rm H}$ from Eq. (32)) in the Higgsing of the gravitational field.

It will turn out to be convenient to introduce the generalization of the 'current photon' $b_\mu$ of Abelian Higgs, Eq.(24). How to accomplish this for rank-2 stress tensors $\sigma_{mn} = 0$? The constraint $\partial_m \sigma_{ma} = 0$ can be enforced by introducing rank 2 tensor gauge fields $b_{ka}$ ('stress gravitons', Eq. 4.1 in [5]):

$$\sigma_{ma} = \varepsilon_{mnk}\partial_n b_{ka}. \qquad (59)$$

The stress gauge field is invariant under the gauge transformations

$$b_{ka}(x) \to b_{ka}(x) + \partial_k \Lambda_a(x), \qquad (60)$$

where $\Lambda_a$ are arbitrary smooth vector fields. The stress tensor $\sigma_{ma}$ is symmetric ("Ehrenfest constraint") and this implies $\epsilon_{kma}\sigma_{ma} = 0 = \partial_a b_{ka} - \partial_k b_{aa}$ and we have to add three constraints compatible with Eq. (60),

$$\partial_a b_{ka} = \partial_k b_{aa}. \qquad (61)$$

Note that the stress gauge field $b_{ka}$ is not necessarily symmetric in $k \leftrightarrow a$.

Let us proceed by expressing $\sigma_{ma}$ and $b_{ka}$ in helical representation. The symmetry of $\sigma_{ma}$ means that the $s = 1$ components must vanish. Furthermore, the conservation of stress $\partial_m \sigma_{ma} = 0$ removes $\sigma^{2,\pm 1}$ as well as the combination $(\sqrt{2}\sigma^{2,0} + \sigma^{0,0})/\sqrt{3}$. The physical components of the stress tensor are therefore in three space dimensions $\sigma^{2,\pm 2}$ and the combination $\sigma^{\rm c} = (-\sigma^{2,0} + \sqrt{2}\sigma^{0,0})/\sqrt{3}$. The stress action becomes in terms of the helical components,

$$\mathcal{L}_{\rm E,elas} = \frac{1}{4\mu}\left(|\sigma^{2,2}|^2 + |\sigma^{2,-2}|^2 + \frac{1-\nu}{1+\nu}|\sigma^{\rm c}|^2\right). \qquad (62)$$

where $\nu$ is the Poisson ratio defined in Eq. (15). The $\sigma^{2,\pm2}$ parts encapsulate the purely transversal shear response of the isotropic crystal. The compressional response is not exclusively determined by the compression modulus $\kappa$ (Eq. 13): $\sigma^{\mathrm{c}}$ also contains the $\sigma^{2,0}$ shear component. This is the meaning of the Poisson ratio: upon applying pressure to a solid, next to a volume change it will also induce a shear deformation.

Let us now consider the helical composition of the "shear gravitons" $b_{ka}$. Using the relations in Appendix B for the curl,

$$\sigma^{2,2} = qb^{2,2}, \quad \sigma^{2,-2} = -qb^{2,-2}, \quad \sigma^{\mathrm{c}} = -qb^{1,0}. \quad (63)$$

It is easy to check that of the nine possible components $b^{s,h}$, three are pure gauge and three are removed by Eq. (61). Inserting this in Eq.(64) the stress Lagrangian in terms of the physical stress gravitons takes the form,

$$\mathcal{L}_{\mathrm{E,elas}} = \frac{1}{4\mu}q^2\left(|b^{2,2}|^2 + |b^{2,-2}|^2 + \frac{1-\nu}{1+\nu}|b^{1,0}|^2\right). \quad (64)$$

This affair becomes now quite transparent. Using the gauge field representation of the stress dual we have managed to obtain the force carrying capacity associated with the emergent shear rigidity on the same footing as the force carrying capacity of spacetime itself – it is all about the $(2,\pm2)$ helical sector. What remains to be done is to inspect the "BF" like mode coupling between the spacetime and stress gravitons, anticipating that this will be a simple linear mode coupling affair.

### D. Coupling static stress with the gravitons: the gravitational Higgs mechanism.

We departed from the observation that there is just a simple linear mode coupling between the gravitons and the stress fields of the form $\mathrm{i}\sigma^{mn}h_{mn}$, Eq. (50). This is of the BF kind, involving the "field strength" of the matter ($\sigma_{mn}$, like $f_{\mu\nu}$) and the "gauge" field ($h_{mn}$, like $A_\mu$).

In the previous section we got enlightened regarding the "shear-gravitons" recognizing the similarity with the way that the generation of Higgs mass arises in Abelian Higgs in dual representation, by the linear BF mode coupling between the current- and EM photons in Eq. (25). It is just the symmetric rank 2 tensor generalization of the vector story in Section III. The main difference is elucidated by the helical projections: instead of the coupling between matter and EM "photons" in the spin $(1,\pm1)$ sector the matter and background gravitons couple to each other in the helicity $(2,\pm2)$ channel.

It follows immediately from the above that this crystal-gravity BF term takes the very simple form in helical representation where as usual $h^\dagger(q) = h(-q)$,

$$\begin{aligned}\mathcal{L}_{\mathrm{E,E-G}} &= \mathrm{i}\frac{1}{2}\sigma_{ma}h_{ma} \\ &= \mathrm{i}\frac{1}{2}q(h^{2,+2\dagger}b^{2,+2} - h^{2,-2\dagger}b^{2,-2}). \quad (65)\end{aligned}$$

This is just the spin-2 generalization of the spin-1 (vector) BF coupling of Abelian-Higgs (Eq. 30) in helical representation.

It is now explicit that the *shear* stress takes the role of the supercurrent, expelling the geometrical 'curvature' from the background, having the same role as the gauge curvature (magnetic field) in the superconductor. We put 'curvature' in quotation marks since the literal curvature of GR is truly non-linear. Eq. (65) reveals that gravitons are actually shear-like in the language of crystal geometry and we will see that on this level it is actually geometrical torsion that is expelled. Riemannian curvature is associated with an infinity of gravitons, going hand-in-hand with the "rotational" topological defects of the crystal, as we will start discussing in section (VI).

Collecting all the pieces of Eq. (50) in the helical (stress) graviton representation, Eqs. (57),(64) and (65),

$$\mathcal{L}_{\mathrm{IEG}} = \sum_{\alpha=\pm2}\left[\frac{c^4}{64\pi G}(\frac{1}{c^2}\omega^2 - q^2)|h^{2,\alpha}|^2 - \frac{q^2}{4\mu}|b^{2,\alpha}|^2 - \mathrm{i}q\frac{1}{2}\mathrm{sgn}(\alpha)h^{2,\alpha\dagger}b^{2,\alpha}\right] - \frac{q^2}{4\mu}\frac{1-\nu}{1+\nu}|b^{1,0}|^2, \quad (66)$$

where we have restored all dimensionful quantities; $q$ refers to the magnitude of the *spatial* momentum and we have written the action in Lorentzian signature, $\omega$ refers to real frequency. This expresses that the gravitons couple exclusively to the shear stress while the compressional stress $\sim b^{1,0}$ is not communicating with the gravitational background. In other regards, the structure of this effective action is indeed similar to the Abelian-Higgs result Eq.(30). Next to the decoupled compressional part there is only one other qualitative difference. In Abelian-Higgs both matter and gauge fields are governed by the same momentum, $c^2q^2 \to c^2q^2 - \omega^2$. As we already emphasized repeatedly, the oddity rooted in the bad breaking of Lorentz invariance is that the propagating gravitons couple exclusively to the *static* material shear stress. The propagating modes of matter (the phonons) carry the wrong spin 1. This will have interesting consequences as we will see soon.

## E. The mass of the graviton.

The linearized crystal gravity system Eq. (66) is a close cousin of the Abelian Higgs analogue Eq. (30) and we will now retrace the repertoire of phenomena that we exhibited in the Abelian Higgs section. The first exercise was in this context to elucidate the origin of the Higgs mass, section III B. In so far the mass of the graviton is invoked it is similar.

Upon integrating out the shear graviton $b^{2,\pm2}$ a simple constant mass term arises since both the coupling and the stress propagators involve only spatial momenta. This results in the effective theory governing the gravitons,

$$\mathcal{L}_{\mathrm{MG}} = \sum_{\alpha=\pm2} \frac{c^2}{64\pi G}(\omega^2 - c^2q^2 - m_{\mathrm{G}}^2)|h^{(2,h)}|^2 \quad (67)$$

$$m_G = \sqrt{16\pi G\mu/c^2} \quad (68)$$

Here $m_G$ is the mass (in units of frequency) of the graviton due to the crystal with shear modulus $\mu$. The corresponding length scale is,

$$\lambda_G = \frac{c}{m_G} = \frac{c^2}{\sqrt{16\pi G\mu}}. \quad (69)$$

The 'gravitational penetration depth' that we announced in the introduction.

Once again, it seems that this graviton mass due to the Higgsing by crystalline matter is not commonly known. In fact, the coupling between the elastic stress tensor and the graviton field has long been known (ref. [37], for recent work see e.g. [38?, 39]). Integrating out the stress tensor field itself already immediately yields this mass. It follows actually from elementary dimensional analysis. The dimension-ful quantity associated with the rigidity of the crystal is the shear modulus $\mu$ with dimension of pressure $\mathrm{kg/m\,s^2}$. The backreaction on the background space is governed by Newtons constant $G$ having dimension $\mathrm{m^3/kg\,s^2}$. The combination $\sqrt{\mu G/c^2}$ is then the dimension uniquely associated with inverse time.

Let us estimate the order of magnitude of the graviton mass. Consider a universe filled with a sturdy solid like steel. The shear modulus is of order $\mu \simeq 10^2\,\mathrm{GPa} = 10^{11}\,\mathrm{kg/m\,s^2}$. Given the values of the natural constants, this corresponds with a gravitational penetration depth of $\lambda_{\mathrm{steel}} \simeq 3.10^{15}\,\mathrm{m}$, roughly equal to 1 lightyear. In order to screen a gravitational wave detector from gravitational waves one has to put it in the middle of a ball made out of steel with a radius of a lightyear! The most sturdy form of elastic matter may be formed in the crust of the neutron star, characterized by a shear modulus $\mu \simeq 10^{21}\,\mathrm{GPa}$. This implies a screening length of order of $10^7\,m$, like the radius of the earth, while the radius of the neutron star is only $\sim 10$ km. For these very good reasons the effects of solid matter on the nature of space time has been ignored in the long history of the subject!

For any noticeable effect elastic matter should be present on cosmological scales. Off and on, cosmologists have been playing with the idea that *dark matter* could behave elastically, e.g. ref.'s [1, 10]. On cosmological scales dark matter is distributed homogeneously and on sufficiently large scales it should then impose a mass on the graviton. How does this relate to the observations? The LIGO collaboration claims a lower limit to the graviton Compton wavelength $\lambda_G = 10^{16}$ m [40], of the same order of magnitude to be expected when the universe would be filled with a solid as strong as steel.

## F. Gravitational hardening of solids.

By integrating out the EM fields in the relativistic Abelian-Higgs case one finds that the Goldstone boson (phase mode) turns into the massive longitudinal photon having an identical dispersion relation as the transversal ones, Eq. (32). This works in essence in the same way in this gravitational context, except that the mass generation only pertains to the *static* stress. This has the amusing consequence that crystals become 'infinitely brittle' with regard to their response to static shear stresses on scales large compared to the gravitational penetration depth $\lambda_G$.

In analogy to Abelian-Higgs, to find out the response of the matter fields integrate out the gravitons from Eq. (66),

$$\mathcal{L}_{\mathrm{SH}} = \sum_{h=\pm2} -\frac{1}{4\mu}\left(q^2 + \frac{1}{\lambda_G^2(1 - \frac{\omega^2}{c^2q^2})}\right)|b^{(2,h)}|^2 - \frac{q^2}{4\mu}\frac{1-\nu}{1+\nu}|b^{(1,0)}|^2. \quad (70)$$

Consider the static limit ($\omega = 0$) and we discern a propagator of the form $1/(q^2 + 1/\lambda_G^2)$. This is the same as one would find for the currents in the static limit in a superconductor, where it has the meaning that external forces (the magnetic field) do not set currents in motion at length scales larger than the London penetration depth. This translates in this elastic context to the statement that an external shear stress is screened on the length scale $\lambda_G$ when it penetrates the solid, due to the presence of the dynamical space-time background. Given the stress–strain duality $w_{mn} = 2\mu\sigma_{mn}$ (Eq. (48)) this has in turn the implication that the solid becomes *unde-*

*formable* when external shear stress is applied ($w_{mn} = 0$) with a strength less than the one associated with $\lambda_G$! Upon applying such a small shear stress, the bulk of the crystal will not respond at all. It is behaving like the crankshaft when a torque is applied but now with regard to shearing. When the external shear stress exceeds the critical value set by $m_G$ the crystal will suddenly deform. This is the gravitational equivalent of the critical current of the superconductor.

In everyday life mechanical engineers are unaware of this 'gravitational stiffening effect' for the obvious reason that the typical dimensions of solids are minute compared to $\lambda_G$. This is analogous to dealing with small superconducting particles having a linear dimension which is small compared to the London penetration depth. These just behave like superfluids, and in the same vein one can ignore the 'Higgsing by gravity' of solid substances in our universe because its effects become noticeable only when these acquire a linear dimension of order of lightyears. It is however amusing to contemplate a world where $G$ would be larger by 40 orders of magnitude or so, such that the gravitational penetration depth would become of order of micrometers. The mechanical engineering manuals would surely have a quite different content.

There is yet another highly peculiar effect, which appears to be unique for this gravitational Higgsing of the shear rigidity. Although we consider here stress that is strictly static in the absence of gravity, it *acquires a dynamical response* by the coupling to the gravitons: the mass term in Eq. (70) is frequency dependent. This is engrained in the 'imbalance' between elasticity and gravity with regard to the loss of Lorentz invariance in the former where the emergent shear rigidity is exclusively tied to space directions. The effect is that spin-2 is only associated with *static* shear. On the other hand, in gravity spin-2 is associated with the *propagating* gravitational waves that only exist as modes by the virtue that the deformations of space oscillate in time. The bottom line is encapsulated by Eq. (57) exhibiting the highly unusual phenomenon that a static force is coupling to a propagating excitation, while the coupling only involves spatial gradients since these are born in the stress sector. Notice that this is quite different from the way that the non-relativistic limit affects the Higgsing of the laboratory electron superconductors. Here the Fermi velocity $v_F$ of the electrons is much smaller than the velocity of light, and it follows immediately that the (time like) electrical 'penetration depth' (Thomas-Fermi screening length) is smaller by a factor $v_F/c$ than the (space like) magnetic (London) penetration depth.

The 'emergent' dynamical nature of the static stress is in principle measurable. Consider the transverse strain correlation functions, which can be shown to be related to the stress correlation function via [24, 25]

$$\langle w^{2,\pm 2}\, w^{2,\pm 2}\rangle = \frac{1}{2\mu} - \frac{1}{4\mu^2}\langle \sigma^{2,\pm 2}\, \sigma^{2,\pm 2}\rangle$$
$$= \frac{1}{2\mu} - \frac{1}{4\mu^2}q^2\langle b^{2,\pm 2}\, b^{2,\pm 2}\rangle, \qquad (71)$$

where we used $\sigma^{(2,\pm 2)} = \pm q b^{(2,\pm 2)}$. This correlation function vanishes for the crystal because $w_{ab} \propto q_a u_b + q_b u_a$, and this cannot have both $a$ and $b$ to be transversal to $q$. But in the presence of gravity, we calculate from Eq. (70)

$$\langle w^{2\pm 2}\, w^{2\pm 2}\rangle = \frac{1}{2\mu}\frac{m_G^2}{c^2q^2 + m_G^2 - \omega^2}. \qquad (72)$$

showing that this is characterized by poles associated with the graviton dispersion $\omega = \sqrt{c^2q^2 + m_G^2}$, with a pole strength of $m_G/2\sqrt{c^2q^2 + m_G^2}$. This quantity can in principle be measured by e.g. inelastic neutron scattering but yet again the scale is set by $\lambda_G$. In the "steel universe" detectors should be a a light year size and capable of detecting absorptions at a frequency $\simeq c/1\text{ly} \simeq 10^{-8}$ Hz.

One may however contemplate possible ramifications on the cosmological scale. Assuming again that dark matter is elastic, there is a surprise: on large scales it will no longer deform when exposed to shear stress! One may imagine that the evolution due to Newtonian gravity to inhomogeneous matter distributions effective shear forces may arise acting on the dark matter that become subjected to the "shear undeformability" on large scales. We envisage that it could well be possible to detect the absence of such shear deformations in the dark matter distributions. We leave it to the astronomers to explore this further in case that the need arises to (dis)prove the assertion that dark matter is elastic.

Finally, yet another difference with the usual Higgs mechanism is that there is just more room for structure in elasticity given its rank-2 tensor nature. A simple but striking example is in the fact that the *compressional* stress is not at all affected according to Eq.(70). The response to an isotropic, hydrostatic stress would be as usual since the isotropic pressure is a scalar that can communicate only with gravitational scalars such as $\Phi$ and $\Psi$ .

## G. The asymmetry of time and the role of the phonons.

The reader may be puzzled: where are the phonons, the ubiquitous excitations of the solid? As we will show here, it is a ramification of the bad breaking of Lorentz invariance that at least in the (quasi) stationary setting these behave as spectators when the solid is homogeneous. This has the counterintuitive ramification that the phonons are *not* affected by the Higgsing. Although

the solid will no longer deform when subjected to a *static* shear stress on scales larger than $\lambda_G$, the phonons continue to behave as massless Goldstone bosons.

Having accepted the wisdom that only the symmetry of space is spontaneously broken, while the emergent shear rigidity has to invoke two space directions (spatial spin 2, quadrupolar deformation), the reason that dynamical phonons "escape" the Higgsing is obvious. As dynamical excitations these occur in the plane spanned by time and one space direction: these are spatial vectors. More precisely, transversal phonons are spatial helicity $(2, \pm 1)$ while longitudinal phonons are combinations of $(0,0)$ (pressure) and $(2,0)$, and these do not couple to the "gravitational" $(2, \pm 2)$ helicity modes. Let us enumerate this in more detail, using the occasion to illustrate the complications in the formalism that arise from the loss of Lorentz invariance. We will present the general strategy to tackle this as developed in ref.'s [24–26] dealing with the full weak-strong dualities relating crystals to quantum liquid crystals where this "asymmetry motive" is crucial. For the phonons this is still easy but these difficulties greatly complicate the formulation of the full dynamical theory – the main reason that we limit ourselves to stationary settings in this paper.

Let us step back to the action of isotropic elasticity as discussed in section (II B). There we already emphasized that the time axis is different because time translations cannot be broken. A greatly simplifying circumstance that we exploit all along dealing with stationary situations is the symmetric nature of the spatial rank two tensors of elasticity, shared with Einstein theory. But this is no longer the case when the time axis comes into play as we already discussed at length in section II B: this we summarized by the statement that the effective theory in Euclidean space time is lacking 'time-like displacements" $u^\tau$, rendering thereby the "time-like" strains (velocities) to be unsymmetric: $\partial_\tau u_a + \partial_a u_\tau \to \partial_\tau u_a$.

We wish to formulate the theory in stress representation. We can identify the stress dual of the "temporal strain" from Eq. (11),

$$\sigma_\tau^a = -\mathrm{i}\frac{\partial \mathcal{L}}{\partial(\partial_\tau u^a)} = -\mathrm{i}\rho\partial_\tau u^a, \tag{73}$$

which is obviously the linear momentum of the system, a spatial vector. The spatial stress tensors are symmetric as the equivalent gravitational tensors and Kleinert found out how to exploit this by the definition of the shear gravitons Eq. (59) (as well as the torque gravitons Eq. 106) parametrizing stress in terms of symmetric rank 2 gauge fields. This rendered the formalism in the above to be efficient and simple. But the asymmetric nature of the "temporal stress" (momentum) disrupts this match having as consequence that the formalism becomes quite messy.

One is forced [24–26] to introduce non-symmetrized spatial stress tensors $\sigma_a^b$ as the duals of $\partial_a u^b$, using upper- and lower indices to keep track of the asymmetry. Sub-

sequently, one has to symmetrize the spatial stresses by hand using the so-called Ehrenfest constraint,

$$\varepsilon_{cma}\sigma_m^a = 0 \tag{74}$$

that may be imposed using Lagrange mulitpliers. In this non-symmetric formulation the dual elasticity action is,

$$\mathcal{L}_{\mathrm{E,stress}} = \frac{1}{2\rho}(\sigma_\tau^a)^2 + \frac{1}{2}\sigma_m^a C_{mnab}^{-1}\sigma_n^b. \tag{75}$$

which is for the isotropic solid,

$$\mathcal{L}_{\mathrm{E,stress}} = \frac{1}{2\rho}(\sigma_\tau^a)^2 + \frac{1}{8\mu}\left[\sigma_m^a\sigma_m^a + \sigma_m^a\sigma_a^m - \frac{2\nu}{1+\nu}\sigma_a^a\sigma_b^b\right]. \tag{76}$$

becoming identical to Eq. (58) when the Ehrenfest constraints are satisfied. The Bianchi-identity associated with the "conservation of total stress" are as before,

$$\partial_\mu\sigma_\mu^a = 0, \tag{77}$$

Given the asymmetry of momentum one is now forced to introduce 2 form (in 3+1D) gauge fields "with a flavor" [26]. Instead of the shear gravitons (Eq. 59), the non symmetric stress is parametrized by,

$$\sigma_\mu^a = \varepsilon_{\mu\nu\kappa\lambda}\partial_\nu b_{\kappa\lambda}^a \tag{78}$$

in 3+1D space time, where we have to use two form gauge fields $b_{\mu\nu}^a$ with a "flavor" label a. The dislocations and disclinations that source these stress "photons" are now strings explaining why two form gauge fields are required (see section V B). In addition, one has now to impose the Ehrenfest constraint as well as the eerie glide ("fracton") constraints on the motion of the topological defects (Section V C). The outcome is a rather baroque and laborious affair that we worked out in the context of the theory of quantum melting of a solid in quantum liquid crystals [24–26]. This machinery may also applicable in this gravitational context but we leave this to a future effort.

In so far the linear modes are concerned, the spatial helical decomposition suffices to understand the role of the phonons. This works well also in the asymmetric formalism: the antisymmetric components of the spatial stress tensor that have to be projected out using the Ehrenfest constraint are easy to identify in this representation [26]. The outcome is that the anisotropic "space-time crystal" is characterized by six physical stress components. The three static components are already listed in Eq. (64). This includes the spin 2 static shear contributions that couple to the gravitons $\mathcal{L}_{\mathrm{E},2,\pm 2} = \frac{1}{4\mu}\sum_{\alpha=\pm 2}|\sigma^{2,\alpha}|^2$. In addition, there are three extra stresses associated with the planes spanned by the time axis and the three space

directions: the phonons. Defining the transversal phonon velocity as $c_{\mathrm{T}} = \sqrt{\mu/\rho}$ the transversal acoustic (TA) phonons are in terms of Matsubara frequency $\omega_n$,

$$\mathcal{L}_{\mathrm{E,TA}} = \frac{1}{4\mu} \sum_{\alpha = \pm 1} (1 + \frac{c_{\mathrm{T}}^2 q^2}{\omega_n^2}) |\sigma^{2,\alpha}|^2 \qquad (79)$$

As we announced, this reveals that the propagating phonons are spin $(2, \pm 1)$ excitations under spatial rotations that therefore do not couple to the quadrupolar spin 2 gravitons.

For completeness, both the longitudinal phonons and -static stresses involve compression and shear at finite wave vectors. To address both the static and propagating longitudinal modes it is convenient to invoke the following helical components for the longitudinal sector: $\sigma^c = (-\sigma^{2,0} + \sqrt{2}\sigma^{0,0})/\sqrt{3}$ and $\sigma^{c'} = (\sqrt{2}\sigma^{2,0} + \sigma^{0,0})/\sqrt{3}$. The trace part $\sigma^{0,0}$ is associated with compressional stress (pressure) that will play its usual role in Einstein theory. In addition, the shear rigidity enters the longitudinal sector via the scalar $2, 0$ component that does not couple to gravitons either. The longitudinal sector becomes,

$$\mathcal{L}_{\mathrm{L}} = \frac{1}{4\mu}\frac{1}{1+\nu} \times$$
$$\begin{pmatrix} \sigma^{c'\dagger} \\ \sigma^{c\dagger} \end{pmatrix}^{\mathrm{T}} \begin{pmatrix} 1 + 2(1+\nu)\frac{c_{\mathrm{T}}^2 q^2}{\omega_n^2} & -\sqrt{2}\nu \\ -\sqrt{2}\nu & 1 - \nu \end{pmatrix} \begin{pmatrix} \sigma^{c'} \\ \sigma^c \end{pmatrix}. \quad (80)$$

Upon diagonalization one will recognize both the static longitudinal mode $\sigma^c$ of Eq. (64) as well as the longitudinal phonon.

This is not news. The fact that gravitons do not couple to the phonons of a homogeneous solid was already recognized by Dyson in the 1960's [37], using in essence the same argument as in the above. The ramification is that gravitons only couple to the lattice vibrations at the *surface* of the solid: here the helical decomposition fails. Such a coupling to dynamical modes is a necessary condition for the detection of gravitons since the latter have to dissipate their energy which requires that these couple to dynamical excitations. This fact was fully acknowledged in the design of the solid body "Weber bar" gravitational wave detectors [41].

## V. DISLOCATIONS, SHEAR STRESS, TORSION AND GRAVITONS.

We are now in the position to address the way that the internal topological sources for the stress fields as introduced in section IV are identified. Having the realization in the back of the mind that this addresses the translational sector we anticipate these to be the *dislocations*. We will assume in the main text that the reader is well informed regarding the basic facts of the topological excitations associated with crystalline order. From this point onward these will be on the main stage. We recommend the reader to have a look in the Appendix C to make sure that he/she is aware of the elementary wisdoms pertaining to the dislocations, disclinations and grain boundaries.

This is yet again evolving in close analogy with the Abelian-Higgs case. We identified the vortex current $J_\mu^V$ to represent the multivalued (non-integrable) phase field configurations (Eq.'s 25, 26). In the solid we have to focus in on the multivalued configurations of the displacement fields $u_a$ instead. This amounts to a straightforward generalization of the vector theory to rank 2 tensors (Section V A) with the main novelty that the topological charge of the dislocations are now the Burgers vectors instead of the winding numbers of the vortices. Dealing with dislocations one can no longer ignore the specific space-group symmetry of the crystal lattice (Section V B) and subsequently we sketch the way to handle dynamical dislocations revolving around two-form gauge fields (Section V C). We then focus in on solid bodies with a spatial extend that is small compared to $\lambda_G$, deriving the relevant action involving dislocations, shear stress and the gravitons as external sources (Section V D), discovering that gravitons exert forces on the dislocations mediated by the stress gravitons. This has the ramification that contrary to a common believe gravitational waves are dissipated in the bulk of solids when these contain a finite density of mobile dislocations, a condition fulfilled in any malleable solid (Section V E). Finally, we consider what happens when the solid is large compared to $\lambda_G$. In analogy with the Abrikosov vortices *fluxoids* are formed characterized by a quantized geometrical flux. The geometrical quantity that is quantized is the *torsion* of Cartan-Einstein theory (Section V F).

### A. The dislocation currents sourcing the stress photons.

In section IV C where we introduced the "stress-graviton" gauge fields $b_{mn}$ we considered on purpose only the smooth displacement fields. However, as in the case of Abelian Higgs there may be also multivalued displacement field configurations. As for the vortices the non-integrability condition will translate via Stokes theorem into a topological invariant, the Burgers vector. To keep matters transparent let us focus on static elasticity first as in the previous section. Later in this section we will sketch how this generalizes to the dynamical version in 3+1D.

As we discussed in section IV A, one finds after the Hubbard-Stratonovich transformation a term that is schematically $\sigma \partial u$ where we assumed the displacement $u$ to be integrable such that $\sigma \partial u \to -u \partial \sigma$. These smooth displacement field configurations turn into Lagrange multipliers that impose the conservation of total stress. However, as for the phase fields one has to allow also for the multivalued configurations (section 2.9, ref. [5]). Em-

ploying the stress graviton's $b_{ij}$ defined in Eq.(59),

$$\sigma_{ma}(\partial_m u_a^{\mathrm{MV}} + \partial_a u_m^{\mathrm{MV}}) = \varepsilon_{mlk}\partial_l b_{ka}(\partial_m u_a^{\mathrm{MV}} + \partial_a u_m^{\mathrm{MV}})$$
$$= b_{ka}J_{ka}$$
$$J_{ka} = \varepsilon_{klm}\partial_l \partial_m u_a^{MV} \qquad (81)$$

the $J_{ka}$ are symmetric rank two tensor densities that enumerate the dislocations.

These can be written as

$$J_{ka} = \varepsilon_{klm}\partial_l\partial_m u_a^{\mathrm{MV}} = B_a\delta_k^{(2)}(x) \qquad (82)$$

where $\delta_k^{(2)}(x)$ was defined in Eq. (29). In direct analogy with the vortices, Eqs.(26-28) the Burgers loop is expressed as

$$\oint_C \mathrm{d}u_a(x) = \oint_C \mathrm{d}x_k\ \partial_k u_a(x) = B_a, \qquad (83)$$

This is the analogue of the starting point of the description of the Abrikosov vortices Eq. (25), and it will be the working horse for the remainder of this section.

### B. Dislocation currents and the space groups.

For reasons explained in section II B we have in a rather cavalier fashion dealt with the non-isotropic nature of real solids. Dealing with the topological currents the full information of the space group governing the spontaneous symmetry breaking becomes crucial again. Given the questions we will ask in this section the need for this information is not obvious but this will change later on when curvature is addressed. It is straightforward to restore this information without even invoking the (secondary) effects of the anisotropic moduli on the stress fields.

It is obvious that in three space dimensions static dislocations are *lines* in the lattice. These lines propagate along the *directions* in the crystal lattice which are set by the point-group symmetries. However, the Burgers vectors are also governed by this information. Since the dislocation corresponds with a plane of unit cells coming to and end in the crystal, the associated Burgers vector points along a lattice direction having a magnitude set by the lattice constant in this particular orientation. Dealing with a crystal with a low symmetry there is quite some accounting to do. For instance, consider a hexagonal crystal like graphite; this has a six-fold discrete rotation

where $B_a$ is the component of the Burgers vector in the $a$ direction. Using Stokes' theorem, convert this into a surface integral of the curl of the integrand over the surface $S$ enclosed by $C$,

$$\int_S \mathrm{d}S_k\ \varepsilon_{klm}\partial_l\partial_m u_a(x) = B_a \qquad (84)$$

and we recognize that this is satisfied by Eq. (82).

In precise analogy with the vortices forming the internal sources for the supercurrent gauge fields ( Eq. 25) this demonstrates that the dislocation densities form the exclusive internal sources for the stress gravitons introduced in the previous section. Combining this with Eq.'s (50,51) and the definition of the shear gravitons $b_{ij}$ (Eq. 59) we arrive at the result,

$$\mathcal{L}_{\mathrm{E,Matter}} = \frac{1}{4\mu}\left[\sigma_{ij}\sigma_{ij} - \frac{2\nu}{1+\nu}(\sigma_{ii})^2\right] + ib_{ij}J_{ij} + i\frac{1}{2}h_{ij}\sigma_{ij} + \mathcal{L}_{\mathrm{E,EH}}. \qquad (85)$$

tion axis $C_6$ associated with the honeycomb lattice and both Burgers- and propagation vectors may point in one of these six direction. In the direction perpendicular to the plane there is only a two fold $C_2$ axis.

The dislocation current is a rank 2 tensor because it has to keep track of both the local propagation direction and the Burgers vector. It has to be a symmetric tensor since a dislocation line propagating in lattice direction $\vec{a}$ and Burgers vector $\vec{B}$ is symmetry wise equivalent to one that is propagating in the $\vec{B}$ direction with Burgers vector $\vec{a}$.

The simplest "spherical cow" crystal lattice that is genuinely representing such space group data in the present context is the simple cubic crystal. In this case one can rely on a simple Cartesian frame where the indices of the current $J_{ka}$ refer to unit vectors pointing in the $x, y, z$ directions. We will use this case to illustrate matters in the remainder.

There is one characteristic that is generic: when the propagation direction is orthogonal to the Burgers vector one is dealing with an *edge* dislocation, while when both directions are parallel one encounters a *screw* dislocation, see Appendix C. Dislocations lines that occur spontaneously in malleable solids form typically closed loops. It is easy to find out that edge dislocations turn into screw dislocations and the other way around when one goes around such a loop; see, e.g., ref. [42] for some general examples.

## C. Dynamical dislocation currents in a 3+1D space time.

Although we will stay away from addressing dynamical phenomena in any detail, let us present here a short account of how the static theory in the above can be generalized to include time. We already highlighted the complications arising from the breaking of Lorentz invariance in section IV G rendering the strain- and stress tensors in space time to become asymmetric. In terms of the asymmetric stress tensors $\sigma_\mu^a$ in 3+1D (we are sloppy with the metric fluctuation $h_{\mu\nu}$),

$$S_{\mathrm{E}} = \int \mathrm{d}\tau \mathrm{d}^3 x \Big[ \sigma_\mu^a C_{\mu\nu ab}^{-1} \sigma_\nu^b +$$
$$\mathrm{i}\sigma_\mu^a (\partial_\mu u_{\mathrm{sm}}^a + \partial_\mu u_{\mathrm{MV}}^a + \frac{1}{2} h_{\mu a}) \Big], \quad (86)$$

to be augmented by the Ehrenfest constraint $\sigma_a^b = \sigma_b^a$. Integrating out the smooth displacement fields for every flavour separately yields $\partial_\mu \sigma_\mu^a = 0$. Implementing this in 3+1D implies that we have to introduce two form gauge fields of the kind discussed in section III E

$$\sigma_\mu^a = \varepsilon_{\mu\nu\kappa\lambda} \partial_\nu b_{\kappa\lambda}^a \qquad (87)$$

and the action Eq. (86) becomes schematically in terms of these two form "stress photons" as

$$S_{\mathrm{E}} = \int d^3 x d\tau \Big[ \epsilon_{\mu\rho\kappa\lambda} \partial_\rho b_{\kappa\lambda}^a C_{\mu\nu ab}^{-1} \epsilon_{\nu\sigma\tau u} \partial_\sigma b_{\tau u}^a$$
$$+ \mathrm{i} b_{\kappa\lambda}^a J_{\kappa\lambda}^a + \mathrm{i} \frac{1}{2} h_{\mu a} \varepsilon_{\mu\nu\kappa\lambda} \partial_\nu b_{\kappa\lambda}^a \Big] \quad (88)$$

where

$$J_{\kappa\lambda}^a = \varepsilon_{\kappa\lambda\mu\nu} \partial_\mu \partial_\nu u_{\mathrm{MV}}^a. \qquad (89)$$

This is the dislocation current in 3+1D. One infers that this is closely related to the "worldsheet" vortex current in 3+1D, Eq. (44) and it can be as well written as in Eq. (45),

$$J_{\mu\nu}^a(x) = B^a \delta_{\mu\nu}^{(2)}(x) \qquad (90)$$

recognizing again the Burgers vector representing the quantized topological charge. The dislocations are just like the vortices, forming strings in 3+1D, with the difference that the dislocation currents are "flavored by their

Burgers vectors", the upper label $a$. In principle the theory Eq. (88) can be completely enumerated. One proceeds by choosing an appropriate (Coulomb) gauge fix for the stress gauge fields, imposing the Ehrenfest constraint "afterwards" as explained in section IV G to deal with the "asymmetry of time". Last but not least, dislocations are subjected to an uncommon constrained kinematics: the "glide constraint", insisting that dislocations can at zero temperature only move in the "slip plane" [43]. This can be easily understood by insisting that no "free atoms" are present – a condition that can be rigorously fulfilled at zero temperature. At finite temperatures away from the melting this density of "substititutional/interstitial defects" is exponentially suppressed and typically very low. But it is possible for the dislocation to move by "breaking and restoring bonds" without invoking extra matter, see Fig. (6) in Appendix C 1. Only quite recently it was recognized [44] that this is an example of the general field theoretical "fracton" notion [45]. The "interstitials" can be viewed as dipolar bound states formed from dislocation-anti-dislocation pairs and since the interstitials are themselves conserved the special fracton constraints follow for the motion of single dislocations in the form of the glide constraint.

Eventually one can even construct the "stress superconductor" dual of the 3+1D crystal, turning out to be a superconducting quantum liquid crystal. The details can be found in ref. [26]: it is in principle straightforward but the "unsymmetric" formalism that is required to accomodate the time axis is inherently laborious.

## D. Gravitons accelerate dislocations.

Let us zoom in on the problem of the static stress fields and static dislocations living in a 3+1D space time as described by linearized gravity: Eq. (85). Conceptually this is a straightforward affair: a field of (shear) stress exerts a force on the dislocations while gravitons couple to this stress as well. Hence, gravitons interact with the dislocations. According to the analysis of the linearized modes in Section IV it is helpful to employ the helical stress-graviton representation We found there three physical stress gravitons $b^{2,\pm 2}$ and $b^{1,0}$. The dislocation densities can be decomposed in the same way – predictably one finds the same components: after all these are the unique internal sources for the stress gravitons. An explicit derivation is presented in ref. [5] (see Eq.'s 4.113, 4.124).

The action Eq. (85) becomes in terms of the helical gravitons $h$, stress gravitons $b$ and the dislocation densities $J$,

$$\mathcal{L}_{\text{disl}} = \sum_{\alpha=\pm 2} \left[ -\frac{q^2}{4\mu}|b^{2,\alpha}|^2 - \mathrm{i}q\frac{1}{2}\mathrm{sgn}(\alpha)h^{2,\alpha\dagger}b^{2,\alpha} \right] - \frac{q^2}{4\mu}\frac{1-\nu}{1+\nu}|b^{1,0}|^2 - \mathrm{i}\sum_{\alpha=\pm 2} b^{2,\alpha\dagger}J^{2,\alpha} - \mathrm{i}b^{1,0\dagger}J^{1,0}$$
$$+ \sum_{\alpha=\pm 2} \frac{c^4}{64\pi G}(\frac{1}{c^2}\omega^2 - q^2)|h^{2,\alpha}|^2 \tag{91}$$

We recognize the result Eq. (66) from Section IV but now augmented by the dislocation density sources.

The result Eq. (91) is actually encoding in this helical/field theoretical language a famous, classic result: the Peach-Koehler equation [46]. This is expressing the somewhat complicated way that an external stress exerts a force on a dislocation line segment. In our notation this reads,

$$f_i = \varepsilon_{imj}\sigma_{im}J_{jm}$$
$$\vec{f} = (\vec{B} \cdot \sigma) \times \vec{l} \tag{92}$$

where in the second line we have written it in the conventional form: a force $\vec{f}$ per unit length is exerted on a dislocation line segment propagating in the $\vec{l}$ direction ("sense") with Burgers vector $\vec{B}$ by a stress (tensor) field $\sigma$.

Also the compressional stresses $\sigma_{ii}$ exert a force on the dislocation line. However, these are exclusively acting in the *climb* direction reflecting the principle that this is in-volving a change in volume. The climb motion is however impeded in a typical crystal and therefore the dislocation line cannot accelerate in this field of force: compressional stress will not dissipate. In helical representation this is encoded in the longitudinal current $J^{(1,0)}$. The force exerted by the shear stress $\sigma_{ij}, i \neq j$ acts on the other hand automatically in the glide direction where the dislocation can "freely" move (modulo practical circumstances like pinning), and these correspond with the $J^{(2,\pm 2)}$ currents in helical representation. Upon applying a shear force the dislocation motions will thereby dissipate the external shear stress leading to the plastic deformations characterizing malleable solids. For edge dislocations this is easy to see (see Fig. 6 in Appendix C 1). It works in essentially the same way for screw dislocations but this is more of a challenge to visualize (see e.g. ref. [47]).

Let us now get back to the full problem including the gravitons. We already highlighted in the previous section that gravitons couple exclusively to the spin 2 shear stress, and we just established that the latter accelerate the dislocations in the glide direction. The gravitational action is therefore entirely in this spin 2 sector,

$$\mathcal{L}_{\text{disl},2} = \sum_{\alpha=\pm 2} \left( \frac{c^4}{64\pi G}(\frac{\omega^2}{c^2} - q^2)|h^{(2,\alpha)}|^2 - \frac{1}{4\mu}q^2|b^{(2,\alpha)}|^2 + \mathrm{i}b^{(2,\alpha),\dagger}(\frac{1}{2}\mathrm{sgn}(\alpha)qh^{(2,a)} - J^{(2,\alpha)}) \right) \tag{93}$$

which is recognized to be a close sibling of the Abelian-Higgs system Eq.(30); this spin 2 sector is identical to the spin 1 Abelian-Higgs problem except that the matter sector is now static as related to the "bad" breaking of Lorentz invariance. As for the Abelian-Higgs case (section III B), let us integrate out the shear gravitons $b^{(2,\pm 2)}$ to determine the effective theory describing the interactions between the gravitons and the dislocations,

$$\mathcal{L}_{\text{DG}} = \frac{c^4}{64\pi G} \sum_{\alpha=\pm 2} \left[ (\frac{\omega^2}{c^2} - q^2)|h^{(2,\alpha)}|^2 - \left( J^{(2,\alpha)\dagger} + \frac{1}{2}q\,\mathrm{sgn}(\alpha)h^{(2,\alpha)\dagger} \right) \frac{4}{q^2\lambda_G^2} \left( J^{(2,\alpha)} - \frac{1}{2}q\,\mathrm{sgn}(\alpha)h^{(2,a)} \right) \right], \tag{94}$$

being the equivalent of Eq. (37).

### E.   Gravitons heat malleable solids.

Let us first find out the consequences of Eq. (94) under the practical circumstance that the linear dimension $L$ characterizing the size of the solid is small compared to the gravitional penetration depth $\lambda_G$. We observe that

$1/(q\lambda_G) \sim L/\lambda_G$ is in this regime the small quantity taking the role of coupling constant mediating the interaction between the gravitons and the dislocations. In this regime,

$$\mathcal{L}_{\text{DG}} = \frac{c^4}{64\pi G} \sum_{\alpha=\pm 2} \left[ (\frac{\omega^2}{c^2} - q^2 - \frac{m_G^2}{c^2})|h^{(2,\alpha)}|^2 + \frac{2L}{\lambda_G^2}\text{sgn}(\alpha)(h^{(2,\alpha)\dagger}J^{(2,\alpha)} - J^{(2,\alpha)\dagger}h^{(2,\alpha)}) + J^{(2,\alpha)\dagger}\frac{4L^2}{\lambda_G^2}J^{(2,\alpha)} \right], \quad (95)$$

showing that gravitons exert a force on the dislocations. The consequence is that gravitational waves are actually attenuated in the *bulk* of malleable solids!

The way that gravitons interact with solids in this regime is a classic subject, motivated by the design of gravitational wave detectors. This pursuit started with Weber designing his solid bar detectors [41]. In order to detect gravitational waves, these *have* to dump energy into the measuring device. For this to happen the gravitons have to excite low frequency phonons but as Dyson pointed out first [37] gravitational waves do not interact with phonons in the bulk of the crystal. It is unclear to us whether it was fully realized in the early days that the unusual ways this proceeds is rooted in he breaking of Lorentz invariance – the reason that phonons and gravitons do not interact in the bulk of crystals is that phonons are spin 1 while gravitons are spin 2 as we showed in Section IV G. This helical decomposition fails when the elastic medium is no longer homogeneous. At the surface of the solid the spin 2 gravitons are therefore mixing with the spin 1 phonons and for this reason the modes couple exclusively at the surface, and this is the number one design principle for solid detectors.

Our stress formalism reveals that gravitons do perturb the bulk of the solid but entirely through *static* stress. The key is that the static responses of an elastic solids are entirely reactive: energy is not absorbed by applying static stresses to an elastic medium. However, many solids are malleable. Upon applying a static stress on a sheet of metal in a stamping press it acquires a permanent different shape instead of springing back to its original shape as expected from the dissipation-less elastic response. The reason is that any piece of metal contains many dislocations. Dislocations will accelerate along the glide directions in a field of static shear stress. This absorbs energy and when the dislocation configurations have adapted to the stress the metal will have acquired its new plastically deformed shape. But in the stamping press the metal has heated up.

This reveals the qualitative mechanism behind the coupling Eq. (95). As we showed in Section IV gravitons exhibit a linear mode coupling with the spin 2 static shear stress. This shear stress is in turn exerting the Peach-Koehler force Eq. (92) on the edge dislocations. When these are mobile as is the case in malleable solids these will accelerate in this field of force. Their motion will in turn dissipate the energy of the gravitons with the effect that these are damped.

Consider the passing of a gravitational wave train due to an astrophysical event like a black hole merger through a piece of malleable solid characterized by a finite dislocation density. The characteristic time scale of the GW's will be of order of milliseconds while the time scales associated with the response of the dislocations are microscopic, of order of nanoseconds or so. This is in the adiabatic limit and we can therefore take the stress exerted on the dislocations by the GW's to be static.

The entertaining observation is that the way of the gravitational radiation interacting with solid matter is revolving around the same principles that underly the action of a black smith forging high quality swords. The network of dislocation lines present in the solid will react by glide (or "slip") motions, reconfiguring in a way that relaxes the external shear stress exerted by the gravitational wave.

The outcome is that the solid will be plastically deformed, heating up in the process. Obviously, this will not play any role under the conditions found on the earth. The shear forces exerted by GW's are extremely feeble. Even in the most malleable solids the dislocation networks are eventually pinned, and the force has to exceed a critical value before these start to move. These pinning forces are huge on the scale of the GW forces. This may however play a role under extreme circumstances. Consider e.g. a planet with a solid iron core orbiting a black hole binary; the gravitational wave tsunami released by the black hole merger may give rise a catastrophic heating of the core of such a planet.

Let us finish with an observation that may be of interest to superconductivity experts. We have already emphasized that modulo the intricacies associated with the "Peach-Koehler" directionality of the forces there is gross similarity with the way that vortices interact via the superflow with EM fields. Specifically, compare Eq. (94) with the static case superconducting case, describing vortices responding to an external magnetic field: Eq. (41). What would be the analogue in superconductivity of the physics we just discussed?

This corresponds with a situation that may be beyond the reach of experimental technique – we are not aware that this was considered even theoretically. The first demand is that one has to consider small superconducting grains with a linear dimension $L \ll \lambda_L$; in the gravitational analogue we are dealing with pieces of solid that are tiny as compared to the gravitational penetration depth $\lambda_G$. Under this condition the superconductor

will behave like a superfluid and the analogue of the dislocation is a *superfluid* vortex line. Imagine now that suddenly the magnetic field is ramped up: this will exert a force of the superfluid vortex which is the analogue of the GW force on the dislocation line. Why is this problem not well documented? The reason is practical. Superconductors are formed from electrons and the ramification is that these excel in *self-annealing*: different from solids where one may have to wait centuries before the dislocationhs have annealed away the vortices formed during the phase transition disappear from the superconductor so rapidly that it may be practically impossible to capture them.

### F.   Dislocations and the quantization of torsion.

Let us now turn to the regime $L \gg \lambda_G$. We established already the close correspondence with the superconducting fluxoids. As for the fluxoids, by varying to the dislocation density the constraint equation follows from Eq. (94),

$$J^{(2,\alpha)} = \frac{1}{2}\mathrm{sgn}(\alpha)q\ h^{(2,\alpha)} \tag{96}$$

by varying to the latter one obtains the equivalent of the vortex equation,

$$\frac{1}{2}\left(\lambda_G^2(\frac{\omega^2}{c^2} - q^2) - 1\right)q\ h^{(2,\alpha)} = \mathrm{sgn}(\alpha)J^{(2,\alpha)}, \tag{97}$$

which is nearly identical to Eq. (40) upon associating $|q|h^{(2,\alpha)}$ with the magnetic field. The only difference is in the extra "propagation" term $\sim \omega^2$ that we already discussed at length in Section IV. This equation suggests that in the static limit the gravitational analogue of a fluxoid is formed from a dislocation and a "graviton flux" where the latter is quantized in units of the Burgers vector. But now we face a problem: gravitational waves have an exclusive dynamical existence, there has to be a time axis for them to exist. But the "gravitational fluxoid" is a static affair.

Let us consider instead what happens in a strictly 3D space-only manifold. As a reminder In the Abelian-Higgs case we re-expressed the BF term $iA\epsilon\partial b \to -ib\epsilon\partial A$ finding out that the material gauge field is sourced by the combination of vortex current and EM field strength $\sim b(J^V - \epsilon\partial A)$ (Eq. (36, section III D). By integrating out the $b$ field the magnetic flux is found to merge with the vortex into the magnetic fluxoid. Let us proceed here in the same way but now in terms of the gravitons $h_{ma}$ and stress gravitons $b_{ka}$, departing from Eq. (85). Focussing on the stress-graviton coupling, we can rewrite

$$\sigma_{ma}h_{ma} = h_{ma}\varepsilon_{mlk}\partial_l b_{ka}$$
$$= b_{ka}\hat{\alpha}_{ka}$$
$$\hat{\alpha}_{ka} = \varepsilon_{klm}\partial_l h_{ma} \tag{98}$$

and the action becomes in terms of these single curl gauge fields,

$$S_\mathrm{E} = \int \mathrm{d}\tau\mathrm{d}^3x \left[\sigma_{ma}C_{mnab}^{-1}\sigma_{nb} + \mathrm{i}b_{ka}(J_{ka} + \frac{1}{2}\hat{\alpha}_{ka})\right]$$
$$+ S_\mathrm{E,EH} \tag{99}$$

The gravitational background enters through the tensor $\hat{\alpha}_{ka}$: one infers immediately that this quantity has a similar status as the magnetic field in the case of the electromagnetic fluxoid, Eq. (36). Upon integrating out the stress gauge fields an effective Coulomb potential will be encountered that imposes $J_{ka} = -\frac{1}{2}\hat{\alpha}_{ka}$, suggesting that the dislocation and the "curvature-like" object associated with the gravitational background merge in a single entity carrying a quantized geometrical flux. What is the meaning of the tensor $\alpha_{ka}$ in gravity?

We encoded gravity explicitly only in linearized form, the infinitesimal $h_{ma}$. For topological purposes this suffices. The reason is the same as for the "neutral" defects, where the (linearized) goldstone bosons suffice to identify the topological currents. The local expressions for the topological currents (like $J_{ka}$) are equivalent to the (Burger) loops Eq. (83,84). One can take an arbitrary large loop where the Burgers vector accumulates from locally infinitesimal displacements. Surely, the core of the dislocation cannot be enumerated in terms of the Goldstone modes but this can be addressed independently. When we turn to curvature we will find that the literal *gravitating* aspects of this core structure will have far reaching consequences, but that is not an issue here.

The strategy is to identify the geometrical meaning of $\hat{\alpha}$ in the linearized theory, that can be subsequently promoted to the non-linear level. The key is that we need the generalization of Einstein gravity, allowing for the presence of *geometrical torsion*. This is accomplished by *Cartan-Einstein gravity*. The geometrical *torsion tensor* defined as $\hat{S}_{mla} = (\Gamma_{mla} - \Gamma_{lma})/2$ which is vanishing in standard GR [30, 31] is now allowed to be finite, see e.g. ref.'s [48, 49].

The elastician is warned: the meaning of this geometrical notion of torsion as discovered by the geometer Cartan is completely unrelated to the way it is used in the mechanical engineering context, as in "torsion bar" [50]. This semantic affair becomes quite awkward in this crystal gravity setting: we will see later that the mechanical engineering use of the word torsion has dealings with geometrical *curvature* instead.

Inserting the linearized Christoffel connection $\Gamma_{mla} = \partial_m h_{la}$,

$$\hat{\alpha}_{ka} = \varepsilon_{kml}\hat{S}_{mla} \tag{100}$$

As we argued, one may now associate $\hat{\alpha}$ with the fully non-linear torsion tensor that is expressing the asymmetry of the Christoffel connection. We can therefore substitute,

$$\alpha_{ka} = \sqrt{|g|}\varepsilon_{kml}S_{mla} \qquad (101)$$

for $\hat{\alpha}_{ka}$ in Eq. (99) to obtain the fully co-variantized dislocation action.

We have rediscovered a motive that is regarded as a highlight of crystal geometry [5]: *the dislocation densities have the geometrical meaning of torsion.* The quantized crystal-geometry torsion merges with the torsion of Einstein-Cartan gravity in a "torsion fluxoid", in the same way that the circulating supercurrents of the vortex merge with the magnetic fields to form a fluxoid characterized by a quantized magnetic flux.

The role of torsion in the geometry of fundamental space-time is ambiguous. It is even not clear whether it exists at all. In this framework the Cartan equations supplement the Einstein equations governing the torsion tensor [48],

$$\hat{S}^k_{ij} + g^k_i \hat{S}^l_{jl} - g^k_j \hat{S}^l_{il} = 8\pi G \sigma^k_{ij} \qquad (102)$$

where $\sigma^k_{ij}$ is the *spin* tensor, a property of matter. This equation represents an algebraic constraint rather than a partial differential equation: the torsion is just determined by the spin tensor.

Let us close our eyes for a moment for this fundamental requirement of Cartan-Einstein theory. Assume that the background can accomodate torsion and consider a strictly 3D spatial Euclidean geometry, lacking propagating gravitons. Under these circumstances we can use the same strategy as in section III D: integrate out the matter gauge bosons from Eq. (99) and as for the magnetic fluxoid (Eq. 37) this demonstrates a Coulomb force binding the torsion flux to the dislocation,

$$\mathcal{L}_{torsionflux} \sim (J_{ka} + \frac{1}{2}\alpha_{ka}) \frac{1}{\nabla^2} (J_{ka} + \frac{1}{2}\alpha_{ka}) \qquad (103)$$

As for the fluxoid, it follows immediately that

$$J_{ka} = -\frac{1}{2}\alpha_{ka} \qquad (104)$$

Using Stokes it follows that in direct analogy with the magnetic fluxoid the defect is carrying a topological quantized torsion flux determined by the Burgers vector,

$$B_a = \oint_S dS_\lambda J_{\lambda a} = -\frac{1}{2}\oint_S dS_\lambda \alpha_{\lambda a} \qquad (105)$$

This has the status of an Einstein equation that takes the form of a constraint equation: *when* the background accommodates torsion the dislocation should be bound to a quantized torsion flux to avoid a Coulomb catastrophe.

We have gathered all the pieces, being in the position to arrive at a conclusion. The first possibility is that the geometry of space-time is the one of Einstein gravity that does not allow for torsion. In this case dislocations will be "pushed around" by dynamical, propagating gravitons but static dislocations will not be screened by the space-time background. The dislocations will be the usual ones, characterized by long range strain mediated interactions. The other possibility is that fundamental geometry allows for torsion. Both the spin density of conventional Einstein-Cartan theory (Eq. 102) as the disclinations will source the torsion; the sources of both the dislocations (Eq. 104) and the "fundamental" spin density (Eq. 102) will add up in a "torsion quantum".

One could contemplate to use these observations to construct a "torsion detector". We depart from the "cosmic solid", imaging it to be a perfect single crystal with a spatial extend $>> \lambda_G$. We imagine the presence of "dark spin currents" imposing a net "translational curvature" (Cartan torsion) in the geometrical background. The prediction is that a "torsion type II phase" should be realized, consisting of a lattice of dislocation lines screened by a pile up of the spinning matter into quantized torsion fluxes.

Given the gigantic separation of scales due the smallness of Newton's constant this is a rather unpractical affair. Assuming that the scales are set by the dimensions of atomic physics $\lambda_G$ has to be of order of light years as we showed in section IV. The typical spatial dimension of the networks of dislocation lines that are responsible for the malleable nature of typical solids is of order of microns. In real solids the shear stress caused by individual dislocations lines is averaged out on the millimeter scale. To make this work one would need a perfect crystalline order on the scale of light years, a feat that can surely not be accomplished by known forms of crystalline matter.

## VI.  GEOMETRICAL CURVATURE, TORQUE GRAVITONS AND THE TOPOLOGICAL DEFECT CURRENT.

We have arrived at the heart of crystal gravity. Einstein theory revolves around the curvature of the space-time manifold and in the preceding chapters there was no mention of this curvature. However, in the same way as geometrical torsion has a precise topological status in crystal geometry, so does curvature. General relativity is famously non-linear but we only addressed how its linear sector – gravitational waves – is affected by the solid. We found out that in Einstein gravity only the linearized modes are affected – the gravitons of the background. Cartan-Einstein with its torsion is required for the space-time to respond to the topological excitations of matter in so far the translational sector is involved – the dislocations.

The topological meaning of dislocations is that they restore the translational invariance. When these proliferate spontaneously the crystal become a liquid crystal: a substance characterized by translational invariance that still

breaks rotational invariance [24–26]. Dislocations are at the same time associated with geometrical torsion, but where to identify in the theory of elasticity the information regarding rotational symmetry breaking? Dealing with the "spin 1" phonons one encounters the same question. A solid breaks both translations and rotations, and why is it so that the only Goldstone bosons are the phonons associated with the breaking of translations, where are the "rotational Goldstone modes"?

The answer lies in the semi-direct relation between translations and rotations. These are intrinsically interrelated: finite translations are the same thing as rotations. But for the translations to become finite one needs an *infinity* of infinitesimal translational modes. One finds the mirror-image of this simple principle in GR. The gravitons of Fierz-Pauli are associated with infinitesimal translations and to build up a metric characterized by a finite curvature an infinite number of gravitons are required.

In crystals this is controlled instead by rigidity principle. A "sector" associated with rotations can be identified but it is not physically observable other than in the form of finite size effects – the "engineering torsion" as we will explain later. This sector contains the analogues of the phonons in the form of "rotational Goldstone bosons" but these are not observable. The reason is the same as for the gluons of QCD being not observable: these are literally *confined* [25, 52] although the confinement mechanism has nothing to do with QCD. The main ramification is an obvious, everyday fact. In the gauge field language associated with force propagation rotations are associated with the response to *torque* stress imposed by twisting opposite ends of the solid. Upon applying such a torque stress to one end of a metal bar this is in a completely rigid way propagated to the other end of the bar when the bar is sufficiently thick: the property of e.g. a prop shaft in the drive train of a car.

As the dislocations are the internal topological currents sourcing the (shear) stress gravitons, torque stress is sourced by unique "defect currents" [5]. These have again a precise topological status, being the unique agents associated with the restoration of the *rotational* symmetry breaking. This is non-standard material and we will discuss the intricacies of the defect currents at length in the next section. For the time being these may be interpreted as representing the *disclinations*, which are discussed on an elementary level in the Appendix C.

In QCD, given that the gluons are confined the sources (quarks) of the gauge fields are confined as well. In direct analogy, free disclinations as the internal sources of torque stress have never been observed in a solid living in a flat background since their existence takes infinite energy. The semi-direct relation between translations and rotations is actually reflected in the nature of the topological excitations. A disclination can be viewed as a bound state of an infinite number of dislocations with equal Burgers vector. The elementary dislocation can be viewed in turn as a bound disclination-antidisclination

pair, like a meson in QCD; see the Appendix C.

We will be focussed in this section on the purely static problem. We ignore completely the time axis that will return in the next section: only the 3D (and 2D) spatial manifold are considered.

A crucial result in the present that we borrow from the mathematical elasticity tradition is due to Kröner [51] in the form of the "double curl gauge fields" (see also ref.[5]). This is a powerful mathematical device that grabs this hidden, confined rotational sector departing from the solid. The "(shear) stress gravitons" that were the working horse in the preceding two sections are in this language the "single curl gauge fields" (Eq. 59) being the incarnation of gravitons in crystal geometry, "pairing" with the gravitons of the dynamical background. For lack of a better word we will call the double curl version "torque gravitons". These would be dual to the Goldstone bosons associated with the rotational symmetry breaking were it not that these are confined. In fact, it can be shown that when the crystal undergoes a zero temperature quantum phase transition from a crystal to a quantum liquid crystal (by proliferation of free dislocations [25, 26]), the torque sector deconfines [52] and the rotational Goldstone bosons are liberated. A liquid crystal responds elastically to a torque stress and the disclinations take here the role of the dislocations in the solid.

Turning to the geometric interpretation, this rotational sector is associated with *curvature* in crystal geometry [5]. Embedding this in a dynamical background the crystal curvature "pairs" with the curvature in the background in the same way as Cartan torsion "pairs" with the dislocations. The crystal curvature is entirely encapsulated topologically by the defect currents which are yet again symmetric rank two tensors. One anticipates that "curvature fluxoids" will be formed, in analogy with the magnetic fluxoids in superconductors and the torsion fluxoids of Section V F. This is indeed the case and it is governed by a stunningly simple and elegant Einstein equation. This takes a constraint form and it insists that the *Einstein tensor* capturing the curvature in the background should *coincide with the defect density*, Eq. (112).

We perceive this as the central result of crystal gravity. Despite its simple appearance, its consequences both in physics and mathematics are intricate and far from completely understood. The remaining sections will be dedicated to a first exploration of this result. The groundwork is done in this section by developing the torque graviton formalism.

## A. The confinement of curvature and defect density.

We depart again from the symmetric structure of the elasticity tensors associated with the 3D space manifold. We ignore for the time being the time axis that actually plays a critical role as explained in the next section.

The crucial insight is that the rotational defects reside "one derivative deeper" in the non-integrable sector of the displacement fields. Kröners invention [51] is to catch the "rotational multivaluedness" by invoking a parametrization in terms of so-called *double curl gauge fields*. The conservation of stress $\partial_m \sigma_{ma} = 0$ is imposed in terms of symmetric 1-form tensor gauge fields $\chi_{ij}$ (Chapter 5 in [5]),

$$\sigma_{ma} = \varepsilon_{mnk}\varepsilon_{abc}\partial_n\partial_b\chi_{kc} \qquad (106)$$

by choosing $\chi_{kc}$ to be a symmetric tensor automatically the symmetric nature of the physical stress tensor $\sigma_{ma}$ is imposed. Contrasting this with the definition of the stress gravitons of the translational sector Eq. 59, this no longer resembles the way that local constraints and gauge fields are related in Yang-Mills type gauge theories. The gauge transformations leaving $\sigma_{ma}$ invariant are now,

$$\chi_{kc} \to \chi_{kc} + \partial_k\xi_c + \partial_c\xi_k, \qquad (107)$$

where $\xi_k$ is an arbitrary smooth vector field. This ensures that only three of the six components of the stress gauge fields in 3D are physical. As will become explicit very soon, this is identical to the way that the physical curvature tensors of GR are parametrized in terms of the metric. We will call the $\chi_{kc}$'s "torque gravitons", contrasting with the single curl (shear) stress gravitons.

The stress part of the action takes yet again a Maxwell-like form but now involving a total of four derivatives. We will save the effort of writing this explicitly - as for the single curl gauge fields this will take a pleasingly simple form in terms of helically projected physical torque gravitons, see underneath.

As for the dislocations of section V, the crucial part is associated with the multi-valued displacement fields as well as the background metric. Let us repeat the procedure Eq. (81) but now for the double curl fields. To simplify the notation we define $w_{ma}^{\mathrm{MV}} = (1/2)(\partial_m u_a^{\mathrm{MV}} + \partial_a u_m^{MV})$

$$\sigma_{ma}w_{ma}^{\mathrm{MV}} = \varepsilon_{mnk}\varepsilon_{abc}\partial_n\partial_b\chi_{kc}w_{ma}^{\mathrm{MV}}$$
$$= \chi_{kc}\eta_{kc}$$
$$\eta_{kc} = \varepsilon_{knm}\varepsilon_{cba}\partial_n\partial_b w_{ma}^{\mathrm{MV}} \qquad (108)$$

As before, the second line follows from the first by partial integration. The defect density $\eta_{kc}$ (chap. 2.12 in ref. [5]) captures the topological excitations associated with the rotations. This quantity is more intricate than the dislocation current $J_{ka}$ and we will have a first look in the next subsection. It is obvious however that in 3D it is a rank two tensor which is symmetric since it sources the $\chi_{kc}$ gauge field which is symmetric.

Consider now what happens to the coupling between the stress tensor and the metric fluctuation when we parametrize the former in terms of the torque graviton gauge field. This works in exactly the same way as for Eq. (108) with the metric fluctuation $h_{ma}$ taking the role of the multi-valued strain $w_{ma}^{\mathrm{MV}}$,

$$\sigma_{ma}h_{ma} = \varepsilon_{mnk}\varepsilon_{abc}\partial_n\partial_b\chi_{kc}h_{ma}$$
$$= \chi_{kc}\hat{G}_{kc}$$
$$\hat{G}_{kc} = \varepsilon_{knm}\varepsilon_{cba}\partial_n\partial_b h_{ma} \qquad (109)$$

It is straightforward to check that $\hat{G}_{kc}$ *is precisely coincident with the spatial components of the Einstein tensor* $\hat{G}_{\mu\nu} = R_{\mu\nu} - \frac{1}{2}Rg_{\mu\nu}$ *evaluated in the weak field limit* $g_{\mu\nu} = \eta_{\mu\nu} + h_{\mu\nu}$.

At first sight it may appear hazardous to draw conclusions from the linearized theory pertaining to curvature that is by default non-linear. However, we are dealing with topological excitations and we can mobilise the same logic as we did dealing with identification of the torsion flux in Section V F. The reason that the defect density is a rank 2 tensor is the same as for the dislocation current being rank 2: the topology insists that the defect density is a line in 3D with a propagation director encoded in one vector, and a topological charge which is also a spatial vector: the Franck vector of e.g. the disclination, see Appendix C. We can employ Stokes to convert this into a large loop encircling the defect and on this loop the locally infinitesimal perturbation of the background metric will accumulate in finite overall topological charge. We can therefore substitute the full Einstein tensor for the linearized one $\hat{G}_{kc} \to G_{kc}$ in Eq. (109).

Gathering all the pieces we find for the action,

$$S = \int \mathrm{dt d}^3 x \sqrt{-|g|}\Big[ -\sigma_{ma}C_{mnab}^{-1}\sigma_{nb}$$
$$-\mathrm{i}\chi_{kc}(\eta_{kc} + \frac{1}{2}G_{kc})\Big]. \qquad (110)$$

Supplemented by the definitions Eq.'s (107, 109) and the Einstein-Hilbert action: this may well be the most consequential equation of crystal gravity. It is a close sibling of Eq. (99) demonstrating the pairing of the dislocation density with the torsion in the background geometry, culminating in the torsion fluxoids. But here we find that in a similar way the torque gravitons are now sourced by the sum of the defect density and the Einstein tensor enumerating the curvature in the spatial back ground.

There is however one big difference of principle with the torsion case. In order to find out how the topological defect density and the background curvature relate to each other we should integrate out the torque gravitons. Let us just count derivatives; the "torsion" Eq. (99) has the structure $\sim C^{-1}(\partial b)^2 + \mathrm{i}b(J + \frac{1}{2}\alpha)$ becoming $(J + \frac{1}{2}\alpha)\frac{C}{\partial^2}(J + \frac{1}{2}\alpha)$ (Eq. 103) implying that the dislocation and the torsion flux are bound together by a Coulomb potential, just as the magnetic fluxoid. But

now we are dealing with the torque gravitons and the action Eq. (110) implies $\sim C^{-1}(\partial^2\chi)^2 + i\chi(\eta + \frac{1}{2}G)$. Upon integrating out $\chi$,

$$S \sim \int \mathrm{d}t\mathrm{d}^3x \Big[ -(\eta_{kc} + \frac{1}{2}G_{kc})\frac{1}{\partial^4}(\eta_{kc} + \frac{1}{2}G_{kc}) \Big]. \quad (111)$$

This implies that a combined matter-background *curvature fluxoid* should form, characterized by a topologically quantized geometrical curvature localized on a line. This is similar as the Abrikosov (and torsion) flux lines. The difference is in the way that the material- and gauge fluxes bind to each other. In the magnetic- and torsion cases we found that these are bound by a $1/r$ Coulomb force. However, by power counting one infers directly from Eq. (111) that upon pulling apart the background curvature $\sim G$ from the defect density core $\sim \eta$ the energy increases *linearly* with the separation. Precisely the same behaviour is found in QCD upon pulling apart static quarks: this is the confinement!

Varying to $\eta$ Eq. (111) implies an EOM (Einstein equation) corresponding with a simple constraint equation similar to e.g. Eq. (38),

$$\eta_{kc} = -\frac{1}{2}G_{kc}. \quad (112)$$

The topological current and the background curvature are now glued together much more tightly. When these two quantities do not compenstate each other locally, the potential energy is increasing *linearly* in the spatial separation between the topological defect and the background curvature. This signals the merciless way that a solid topologizes the spatial curvature of fundamental spacetime. In the presence of the solid all the spatial curvature *has* to be collected in curvature fluxoids, and "non-topologized" curvature cannot exist for the same reason that free quarks cannot exist in the confining regime of QCD.

Using the helical projection we will fill in the details in the remainder of this section. But let us have a first look at the meaning of the defect density. (see also the Appendix C 3).

### B. Defect density: dissecting the curvature of a crystal.

The confinement works in either way. In a flat background devoid of curvature the defect density has to vanish since $G_{kc} = 0$. This is surely the case and since disclinations are unobservable in solids there is not much of a literature describing the way it works in detail [62, 63], in contrast with dislocations being behind the macroscopic behaviour of many solid substances. An exception is the study of 2D solids covering the surface of a rigid sphere by the soft matter community [27, 28, 53–57] that we will

discuss in Section VII C. A related subject is the description of microstructure, associated with disclinations that are locally screened by compensating dislocation structure [58–61].

Upon restoring the gravitational time axis we will see in the next section that complications arise that are of a similar kind being in the forefront of the soft matter affair. This will force us to look beyond the elementary disclination wisdoms as reviewed in Appendix C 3. Fortunately, the mathematical prerequisites are available in the form of a rather elegant affair in differential topology: see chapter 2 in Kleinert's book [5] for a thorough analysis. As a first introduction, let us collect here some of the highlights.

The meaning of the defect density is that it keeps track of the line-like (disclinations), plane-like (grain boundaries) and volume like (dislocation gas) ways of organizing infinities of dislocations such that they describe a net curved crystal manifold (see Appendix C 3). How does this relate to the differential geometry language, with its referral to non-integrable multi-valued parts? Instead of focussing on the non-integrability associated with the symmetric strain fields, let us consider the antisymmetric combinations that keep track of local rotations,

$$\omega_{ma} = \frac{1}{2}\left( \partial_m u_a - \partial_a u_m \right). \quad (113)$$

The *disclination density* is defined as,

$$\theta_{kc} = \varepsilon_{knm}\varepsilon_{cba}\partial_n\partial_b\omega_{ma} \quad (114)$$

One infers immediately that compared to the dislocation density $J_{kc} = \varepsilon_{knm}\partial_n w_{ma}$ there is one more derivative, mirrored in the above by the need to introduce the torque gravitons. This disclination density is enumerating the disclinations as they are recognized in the elasticity canon: see Appendix C 2. In short summary, these are like dislocations in the regard that they are line-like textures in 3D characterized by a vectorial topological charge called the Franck vector. The Franck vector is quantized in units of the discrete rotations characterizing the space group. For this reason it is a rank 2 symmetric tensor current, as the dislocation current. Both the propagation direction and the Franck vector are determined by the discrete point group operations associated with the space group.

Comparing this with the defect density Eq. (109) a piece is missing. This can be expressed in terms of a "trace free" dislocation density called "contortion",

$$K_{ak} = -J_{ka} + \frac{1}{2}\delta_{ka}J_{ll} \quad (115)$$

where $J_{ka}$ is the dislocation density. It is easy to check (see Kleinert) that the defect density can be decomposed as,

$$\eta_{kc} = \theta_{kc} - \varepsilon_{cba}\partial_b K_{ak} \qquad (116)$$

where the contortion part collects the non-disclination contributions to the defect density. Observe that this torque graviton formulation contains the stress graviton theory of section V: impose that there is no multivaluedness associated with rotations and it is easy to show that the theory reduces to the one describing single dislocations interacting via the shear gauge fields.

In fact, the basic rule captured by the defect density is that for *any* dislocation configuration formed from a macroscopic number of dislocations characterized by *Burgers vectors pointing in the same direction* ("Burger's vector magnetization") represents crystal curvature, actually invariably localized on a *line* in 3D. A liquid crystal is defined in this topological language [24–26] as a system of free dislocations characterized by "local Burger's neutrality": on the microscopic scale their Burgers vector occur with the same probability pointing in precisely opposite directions. Notice that this underpins a general classification of liquid crystal order as descending from the space-groups of crystals [64, 65].

The disclinations are a special case: these can be viewed as a stack of an infinite number of dislocations organized in such a way that the dislocations can no longer be identified having the ramification that the curvature is topologically quantized (see Appendix C 2). But one may as well organize the "equal Burgers vectors" in the form of planes – the grain boundaries [73, 74] that can be stacked in a way that it absorbs curvature. These are more costly energetically than disclinations since the number of required dislocation cores grows with area. The worst case is the "equal Burgers vector dislocation gas" since the number of dislocation cores required to absorb curvature grows with volume. We will take up this theme at length in Section VII C.

Let us conclude by zooming in on the disclinations as the most intuitive topological curvature agents. Once again, this is a line-like (in 3D) defect propagating along lattice directions. Its topological charge is called the Frank vector, defined as

$$\Omega_c = \oint_{\partial\mathcal{S}} dx^m \, \partial_m \tfrac{1}{2}\epsilon_{cma}\omega_{ma}. \qquad (117)$$

Here $\partial\mathcal{S}$ is an arbitrary closed contour encircling the core of the topological defect, such that the surface area $\mathcal{S}$ enclosed by $\partial\mathcal{S}$ is pierced by the defect line. It is quantized in units set by the pointgroup (see the Appendix C 2). The disclination density tensor Eq. (114) can be written as,

$$\theta_{kc}(\vec{x}) = \epsilon_{kln}\partial_l\partial_n\epsilon_{cma}\omega_{ma}(\vec{x}) = \delta_k(L,\vec{x})\Omega_c, \qquad (118)$$

with the delta function defined in Eq. (29). By using Stokes theorem and integrating the density over the surface $\mathcal{S}$ these identifications can be easily checked.

An immediate consequence of the relation between the dislocations and disclinations is that the former are no longer conserved in the presence of a finite defect density. In simple terms, individual dislocations can be freely "added" or "peeled off" from e.g. the disclinations, while the orientation of the Burgers vector of the dislocation will shift upon encircling a disclination, see e.g. ref. [68]. It is easy to demonstrate that

$$\partial_k J_{ka} = -\varepsilon_{alc}\theta_{lc}. \qquad (119)$$

expressing that the disclination currents act as sinks and sources of the dislocation currents.

## C. Dissecting (crystal) curvature in three space dimensions.

We learned in Section V that regardless the details of the symmetry of the crystal, dislocations occur in two gross categories: edge dislocations and screw dislocations. These could be identified already in the isotropic theory in terms of the traceless- and trace parts of the dislocation density- and shear graviton tensors. A similar subdivision applies to disclinations that can be categorized as being of the "wedge-" or "twist" variety. For a pedestrian introduction we refer to the Appendix C 2; these are obtained by just cutting out a "Volterra" wedge and gluing it together again in such a way that the cutting surface becomes invisible, imposing thereby the quantization of the Franck vector. The Franck vector is *parallel* to the propagation direction dealing with the wedge disclination, corresponding with the trace part of the disclination density. This can be in turn viewed as a bound state of an infinity of edge dislocations (see Fig. 9). However, one may also perform a cut such that the Franck vector is *orthogonal* to the propagation direction: these are the *twist* disclinations.

Departing from the "mother" equation (110) we discovered on basis of power counting (Eq. 111) that the defect density and the Einstein tensor are "confined" to be the same: $\eta_{ij} = G_{ij}$. But how does this looks like in detail, being aware of the way that the disclinations can be "organized" in a twist- or wedge form? A first question is, do the independent components of the background curvature match those of the crystal curvature expressed through the defect density? One may even depart from the curvature degrees of freedom of the 3+1D theory including time. The Riemann tensor is characterized by 20 independent components but given the Higgsing of gravity the 10 Weyl components are frozen out. Only the 10 Ricci components remain. Ignoring the time related components, the six independent spatial components of the Einstein tensor remain. These are in turn constrained by 3 Bianchi identities $\nabla_\mu G^{\mu\nu} = 0$, and in combination with the crystal gravity constraint equation $\eta_{kc} = -\tfrac{1}{2}G_{kc}$ (Eq. (112)) this implies that three independent gravitational fluxoids exist. We conclude that

the match is perfect, all forms of spatial curvature can be absorbed in the curvature fluxoids.

Let us now proceed from isotropic elasticity, to find out what we can learn by employing helical projections, resting on Chap. 5 of ref. [5]. There is a shift in the interpretation as compared to the "translational sector" of Section V since we are now dissecting curvature itself – for instance, it is no longer the case that only spin 2 has dealings with gravity. The transversal components are still decomposed in terms of the $(2, \pm 2)$ polarizations, but a main difference with the dislocations is that the longitudinal polarization of the torque gravitons is now governed by a dreibein tensor $e_{ln}^L = \frac{1}{\sqrt{3}}\left(-e_{ln}^{(2,0)}(\mathbf{q}) + \sqrt{2}e_{ln}^{(0,0)}(\mathbf{q})\right)$ (Eq. (5.13), ref. [5]). In this helical representation the action Eq. (110) becomes in terms of physical torque stress "gravitons" $\chi^{(s,m_s)}$ (compare with Eq.'s 5.22, 5.24, Kleinert),

$$S = \sum_{\omega_n} \int \frac{\mathrm{d}^3 q}{(2\pi)^3} \left[ -\frac{q^4}{4\mu}\left(|\chi^{(2,2)}|^2 + |\chi^{(2,-2)}|^2 + \frac{1-\nu}{1+\nu}|B^L|^2\right) - \mathrm{i}\left(\sum_{\alpha = \pm 2} \chi^{(2,\alpha)\dagger}(\eta^{(2,\alpha)} + \frac{1}{2}\hat{G}^{(2,\alpha)}) + \chi^{L\dagger}(\eta^L + \frac{1}{2}\hat{G}^L)\right) \right] \tag{120}$$

The $\hat{G}$ should represent the Einstein tensor in helical representation. This fits the expectation: the 6 independent components of $\hat{G}$ constrained by the 3 Bianchi identities to a total of three real degrees of freedom. Imposing spatial isotropy these decompose naturally in two traceless spin 2 $G^{(2,\pm 2)}$ parts and a longitudinal $G^L$ component containing the trace.

Upon integrating out the $\chi$ stress gravitons

$$S = \sum_{\omega_n} \int \frac{\mathrm{d}^3 q}{(2\pi)^3} \left( -\sum_{\alpha = \pm 2} (\eta^{(2,\alpha)\dagger} + \frac{1}{2}\hat{G}^{(2,\alpha)\dagger}) \frac{\mu}{q^4} (\eta^{(2,\alpha)} + \frac{1}{2}\hat{G}^{(2,\alpha)}) - \frac{1+\nu}{1-\nu} (\eta^{L\dagger} + \frac{1}{2}\hat{G}^{L\dagger}) \frac{\mu}{q^4} (\eta^L + \frac{1}{2}\hat{G}^L) \right) \tag{121}$$

where we recognize the $1/q^4$ interaction responsible for the confinement. In isotropic 3D Riemannian the curvature can be decomposed into one longitudinal $\hat{G}^L$ and two transversal spin $(2\pm 2)$ sectors, getting topologized in the three sectors of curvature fluxoids. We find that both the Riemannian curvature and the crystal curvature decompose in three dimension in one longitudinal- and two transversal spin 2 sectors. Much of the remainder will be devoted to a further elaboration of this fundamental outcome.

Upon transforming back from helical- to Cartesian components Eq. (121) becomes (compare with Kleinert Eq. 5:30),

$$S = -\mu \sum_{\omega_n} \int \frac{\mathrm{d}^3 q}{(2\pi)^3} \frac{1}{q^4} \left(\eta_{kc}(\vec{q}) + \frac{1}{2}\hat{G}_{kc}(\vec{q})\right)^\dagger \left[\frac{1}{2}(\delta_{kl}\delta_{cd} + \delta_{kd}\delta_{cl}) + \frac{\nu}{1-\nu}\delta_{kc}\delta_{ld}\right] \left(\eta_{ld}(\vec{q}) + \frac{1}{2}\hat{G}_{ld}(\vec{q})\right) \tag{122}$$

One immediately infers that the (longitudinal) trace part $(k = c, l = d)$ corresponds with the wedge disclination: the Franck vector is parallel with the propagation direction. On the other hand, the transversal spin 2 sector refers to a crystal curvature where the Franck vector is orthogonal to the propagation direction: these enumerate the twist sector.

The reader may wonder how this relates to the Riemannian (Ricci) curvature in 3D space. In dimension higher than 2 curvature acquires more structure than the simple and intuitive scalar curvature. We will take this theme up in the final sections, first focussing in on the simpler "wedge curvature" in Section VIII to then generalize it to include the "twist topology" in Section IX. In this last section we will offer a first glimpse on how how twist- versus wedge disclinations relate to the curvature in the 3D background geometry, revealing a mathematical challenge that to the best of our knowledge is completely uncharted: the non-Abelian nature of the "curvature fluxoid" topology as rooted in the rotational symmetry.

## D. Curvature and defect density in two space dimensions.

In physics minimal models play a crucial role: look for the simplest possible set of circumstances where the essence of the physics can still be recognized avoiding secondary complications. In the context of crystal gravity, this role is played by its realization in *two* space dimensions. Only one curvature invariant survives: the scalar (Gaussian) curvature $R(\vec{x})$. On the other hand, the defect density becomes a scalar as well. The defect density turns into a point as well while it is impossible to realize a twist defect. All what remains are the point like wedge "disclinations" characterized by a Frank "scalar" taking quantized values (see Fig. 7). One already infers that $R$ and $\eta$ count in the same way.

We ignored the 2D case when we dealt with the dislocations. The realization that these count in the same way as the disclination may help the reader. In 2D only edge dislocations exist (screw dislocations need a third dimension). Since wedge disclinations can be viewed as a bound state of edge dislocations this explains that the co-dimension of the dislocation and disclinations have to be the same. Accordingly the dislocation density is in 2D also a scalar. This is a far reaching simplification, being behind the work of the soft matter community inherently limited to 2D geometry [27, 28, 53–57].

This goes hand in hand with the parametrization of the stress fields in terms of the single curl stress photons: $\sigma_{ma} = \varepsilon_{mn}\partial_n b_a$, and a transversality condition $\partial_x b_x + \partial_y b_y = 0$ ensuring the stress tensor to be symmetric. As specific for 2D $b_a$ is no longer a proper gauge field, it turns into a scalar field flavored by the Burgers "scalar" label $a$. The stress action then takes the form $\mathcal{L} = -(\partial_k b_a)^2/(4\mu(1+\nu))$, being sourced by $-ib_a J_a$ where $J_a$ is the density of edge dislocations $J_a(\vec{x}) \sim B_a \delta^{(2)}(\vec{x})$, cf. Eq. (82). In 2D (or 2+1D) there are surely no gravitons and in the absence of torsion there is not much left to do.

There is surely more going on in the curvature sector. The torque gravitons, defect currents and geometrical rank 2 tensors are all reduced to scalars. For instance, the only surviving component of the defect density is $\eta_{ij} \to \eta_{33} = \eta$ where "3" refers to the direction perpendicular to the 2D plane. This is of course consistent with the fact that in 2D only the Gaussian intrinsic (= scalar) curvature exists.

In 2D Eq. (109) becomes $\sigma_{ma}h_{ma} = \chi G$, with $G = \epsilon_{nm}\epsilon_{ba}\partial_n\partial_b h_{ma}$, corresponding precisely with *minus* the Gaussian curvature in linear approximation. Stress is now parametrized as $\sigma_{ma} = \varepsilon_{mn}\varepsilon_{ab}\partial_n\partial_b\chi$. Given these simplifications the action Eq. (110) reduces to the simple form,

$$S_{2D} = \int dt d^2x \left( -\frac{1}{4\mu(1+\nu)}(\partial^2\chi)^2 - i\chi\eta - i\frac{1}{2}h_{ma}\sigma_{ma} \right) \tag{123}$$

All what remains to find out is the meaning of the "BF" terms coupling the matter to the background metric,

$$h_{ma}\sigma_{ma} = -\chi\hat{R} \tag{124}$$

where $\hat{R} = -\varepsilon_{nm}\varepsilon_{ba}\partial_n\partial_b h_{ma}$ is the aforementioned linearized Gaussian curvature scalar in 2D

All what remains to be done is to integrate out the stress gravitons. The result is just the "scalar" version of the 3D result Eq. (111) written in the precise form,

$$S_{2D} = -4\mu(1+\nu)\int dt d^2x$$
$$\left( \eta(\vec{x}) - \frac{1}{2}R(\vec{x}) \right)\Gamma_B(\vec{x},\vec{x}')\left( \eta(\vec{x}') - \frac{1}{2}R(\vec{x}') \right) \tag{125}$$

where $\Gamma_B$ is the Greens function of the biharmonic operator

$$\partial^4\Gamma_B(\vec{x},\vec{x}') = \delta(\vec{x} - \vec{x}') \tag{126}$$

implying the confinement between curvature and defect current also in 2D. This is a central result in the soft matter community effort dealing with solids covering curved surfaces (see e.g. Eq's (72,73) in the tutorial ref. [27]). Kleinert's efficient and transparent stress gauge field formalism is not quite realized in this community, but their more laborious methodology is precisely equivalent.

The constraint equation Eq. (112) we found for three spatial dimensions simplifies to the extreme in 2D,

$$\eta(\vec{x}) = \frac{1}{2}R(\vec{x}) \tag{127}$$

The Gaussian curvature of the 2D manifold has to be "equal" to the density of rotational topological excitations of the solid. The soft matter community is focussed on the ramifications of this exceedingly simple formula in the case of a rigid background geometry where $R(x)$ is just a given quantity. Despite the simple appearance of these equations this turns out to give rise in the case of rigid curvature to an exceedingly complex frustration problem that can only be addressed experimentally and by computational approaches [27, 28, 53–57]. We will come back to this in Section VII C.

## VII. CURVATURE FLUXOIDS AND THE PROBLEM OF TIME.

Having identified the principles rooted in the topology of crystals we are now facing the task to find solutions of Eq. (122) (or Eq. 125 in 2D) to find out how the

topologized geometry of crystal gravity looks like when spatial curvature matters.

Up to this point we largely ignored the time dimension. Within the restrictions of stationary solutions, time did not seem to matter much: in the context of dislocations and Cartan's torsion we only stumbled on the odd circumstance of dynamical gravitons "pushing" static shear stress. However, when curvature gets into play the rules change. After all, GR is in the first place a theory dealing with gravitation, the fact that what seems to be the attractive force between bodies with mass of Newtonian gravity is actually about extremizing the length of a path (geodesic) in a curved geometry with Lorentzian signature pertaining to the time dimension [30, 31].

This "gravitating side" of gravity interferes with the usual "topologization logic" pertaining to the Higgs phenomenon. In the presence of background gauge curvature (magnetic field) the superconducting condensate forces this to be absorbed in an Abrikosov lattice of quantized magnetic fluxoids. As we argued in section V F, geometric torsion in the background should merge with the dislocations in 'torsion fluxoids' characterized by a Burgers vector topological quantum. As we learned in the previous section, the *disclination* has the status of matter defect representing the topological quantum of curvature. Henceforth, one may anticipate that geometrical curvature in the background should merge with the disclinations into "curvature fluxoids" forming some kind of analogue of the Abrikosov lattice. In a universe with only space-like dimensions this is what is happening. This is the subject of the next two chapters where we will make the case that this is a quite interesting mathematical affair. In a perhaps unfortunate way, the Lorentzian time of the physical universe spoils this fun.

A relativist will immediately recognize the obstruction. As discussed in section VII A, the background geometry that merges with the disclination is like the conical singularity or cosmic string in 3D and 4D gravity, respectively. The specialty is however that the amount of curvature (opening angle) stored at the core of the defect has to be "of order 1" since this is set by the Franck vector. The bottomline is that in order to find a solution an energy of order of the Planck mass has to be mobilized to stabilize the core (section VII B). Such circumstances cannot possibly be fulfilled dealing with mundane solids.

But "curvature topology" is a subtle affair. The background curvature "confines" with the defect density according to Eq. (122), *not* exclusively with the disclination density that is only a component of the defect density. It will turn out that the situation is closely related to the frustration problem identified by the soft matter community with their rigid curved surfaces [27, 28, 53–57]. In section VII C we will argue that there is a unique solution in the parameter regime of relevance in crystal gravity. This is very simple: the curvature is absorbed by a simple, dilute gas formed from dislocations with shared Burgers vectors. The net outcome is that via this loophole the matter "fails to topologize" geometrical spatial curvature.

## A. Wedge disclinations and gravitational fluxoids in the spatial manifold.

The construction of curvature fluxoids associated with wedge disclinations in two- and three dimensional spatial manifolds is very easy. Let us first focus in on two space dimensions; this generalizes straightforwardly to the wedge disclinations in 3D while we will later learn that the same difficulties plague the other 3D "twist" disclinations as well.

Let us recall the Volterra construction for the elementary disclination in 2D. Take a sheet of solid, cut out a wedge and glue the edges together under the condition that the cutting surface does not alter the crystal lattice. This turns into a literal Kindergarten exercise by embedding this in our 3D spatial universe: a cone is obtained. This is the GR classroom device that we highlighted in the very beginning, illustrating the nature of curvature: the circumference of a circle is no longer $2\pi r$ where $r$ is the radial coordinate with $r = 0$ at the tip of the coin, but instead $2\pi(1 - \alpha)$ where $\alpha$ is the opening angle.

In order to render the cutting seam to be invisible in the crystal geometry the opening angle has to be quantized in units of the discrete rotations associated with the space group of the crystal. A canonical example is found in a natural occurring 2D crystal: graphene with its hexagonal honeycomb lattice formed from carbon atoms. The core of the point-like disclination is formed by a 5-ring accompanied by a perfect hexagonal lattice everywhere else – the anti-disclination is accordingly a 7 ring (see fig. 7). Given the six-fold rotational axis the opening angle $\alpha = 1/6$, which is in turn coincident with the Franck "scalar" (in 2D) $\Omega = 2\pi\alpha$ introduced in the previous section.

Using the definition Eq. (118) for the disclination density and the "Einstein equation" Eq. (127) it follows for a disclination at the origin in 2D,

$$R(\vec{x}) = \eta(\vec{x}) = 2\pi\alpha\delta(\vec{x}) \qquad (128)$$

To avoid the infinities associated with the confinement the scalar curvature in the background geometry $R(\vec{x})$ is entirely concentrated at the disclination core while elsewhere $R$ is vanishing. We recognize directly a GR textbook solution: this is the conical singularity!

The metric is written in terms of radial coordinates with the singularity at $r = 0$ as,

$$ds^2 = dr^2 + (1 - \alpha)2\pi r^2 d\phi^2$$

and it is a textbook exercise to demonstrate that the scalar curvature acquires the form Eq. (128).

This example illustrates vividly the simple principle governing the formation of the curvature fluxoid in 2D.

Consider the classroom cone; we exploited the third dimension to construct it but when we attempt to force it into two dimensions the confinement takes over: the paper cone crumbles when we push the tip to the table. The remedy is to *cut out the same wedge from the 2D space itself*, thereby endowing space itself with a conical singularity curvature.

We can directly proceed generalizing this to a wedge disclination line in 3D. In a 3D crystal the discrete rotations may be different pending the direction. To avoid these complications let us consider a cubic crystal, choosing a Cartesian coordinate system along the lattice vectors. The Franck vectors are now of magnitude $|\Omega| = 2\pi\alpha$ where $\alpha = 1/4$ (right angles) and consider an elementary wedge disclination propagating along the z-direction. This corresponds with a Volterra cut in the xy plane as in 2D with a core which is now pulled out in a line oriented in the z-direction. The only non-zero component of the disclination density is the $zz$ one; it follows from Eq.'s (112, 118) that the only non-zero component of the Einstein tensor is,

$$G_{zz}(\vec{x}) = \eta_{zz}(\vec{x}) = |\Omega|\delta_z^{(1)}(\vec{x}) \qquad (129)$$

where $\delta_z^{(1)}(\vec{x})$ measures the position of the disclination line in the $z$ direction.

This relates yet again a famous GR geometry: such an Einstein tensor is associated with the metric [69] of a *cosmic string*, e.g. ref. [70]. This corresponds with 2D conical singularities pulled out in a line in 3D, in cylindrical coordinates and Cartesian coordinates resp.,

$$ds^2 = dz^2 + dr^2 + (1-\alpha)r^2 d\phi^2$$
$$ds^2 = dz^2 + (1 - \alpha\frac{y^2}{x^2+y^2})dx^2 + (1 - \alpha\frac{x^2}{x^2+y^2})dy^2 + \alpha\frac{2xy}{x^2+y^2}dxdy \qquad (130)$$

where $\alpha = \Omega/2\pi$. Although harder to visualize, this in essence the same affair as in 2D: a 3D Volterra wedge is removed from the solid leaving behind a disclination line and to accommodate this in three dimensions one has to remove the same wedge from space itself. As in 2D, the space is locally flat everywhere except at the disclination core and the combination of wedge disclination and cosmic string geometry is merely *topological*; the geometry is locally flat while the curvature becomes only manifest by measurements that in one or the other way "lasso" the defect /cosmic string. For instance, it is well known that light passing by cosmic strings may exhibit interference fringes.

We will take up this theme again in the final sections, elaborating some mathematical consequences. In the physical universe we meet a complication: there is also a time dimension being responsible for an unpleasant complication.

### B. Curvature fluxoids and the problem of the heavy core.

Once again, the relativist should have been already alerted by the arguments we just presented. GR is about gravity and gravity means that energy affects the nature of space, according to the Einstein equations. This problem arises at the moment that one includes a Lorentzian time axis: conical singularities and cosmic strings are among the simplest examples where one sees this at work.

Consider a static "curvature fluxoid" formed from a wedge disclination confined with a cosmic string metric characterized by a Franck vector opening angle. But let us now account for the fact that gravity requires the time axis. In fact, the conical singularities in 2+1D became part of the GR canon by the seminal work of Deser, Jackiw and 't Hooft in 1984 [71] aimed at elucidating the nature of gravity in this dimension. In two space- and one time dimension the Weyl tensor is vanishing by default and gravity turns in a topological theory similar to crystal gravity in higher dimensions. 't Hooft *et al.* assumed that the geometry is sourced exclusively by point particles, discovering that every particle "binds" to a conical singularity. The system of particles turns into an affair governed by topological interactions which is in principle numerically tractable [72].

The derivation departs from a particle at rest at the origin. One adds the time axis - the metric becomes $ds^2 \to -dt^2 + ds^2$ - to solve the Einstein equations. The solution [71] is textbook material. The result is the conical singularity spatial geometry Eq. (129) but now the opening angle is determined by the rest mass $m$ of the particle,

$$\alpha_{3D} = 4Gm \qquad (131)$$

Let us directly proceed to the cosmic string in 3+1D which is equally well known. Yet again one adds the time axis to solve subsequently the Einstein equations now sourced by a thin tube of gravitating matter characterized by a string tension (mass per unit length) $\rho$, yielding for the opening angle

$$\alpha_{4D} = 8\pi G\rho_{fluxoid} \qquad (132)$$

In order to exist in the physical universe the wedge curvature fluxoid should be a solution to *both* the usual "time like" Einstein equations, as well as the additional crystal gravity Eq. (112). The opening angles given by Eq.'s (129,132) should match and this implies for the cosmic string,

$$\rho_{fluxoid} = \frac{|\Omega|}{8\pi G} \tag{133}$$

The trouble becomes now manifest. Cosmic strings are usually viewed [70]as a remnant topological defect associated with a GUT scale phase transition, where the core energy/mass of the string is set by the GUT scale. Nevertheless, the opening angle is then only of order of $10^{-6}$ radiants. But the Franck vector of the disclination implies an opening angle that is $O(1)$ radiants! The trouble is that one now needs a mass density stabilizing the curvature at the "tip of the cone" that is of order a Planck mass per Planck length. Let us estimate this in explicit units. Assert that $|\Omega| \sim 2\pi$, restore a factor $c^2$ for an answer in terms of kg/m: $\rho_{flux} \sim c^2/4G \simeq 10^{26}$ kilogram per meter. One has to mobilize roughly 100 earth masses concentrated in a core area with a dimension set by the lattice constant (because of the confinement) to stabilize one meter of curvature fluxoid. This signals an obvious problem with the realization of such perfect disclination fluxoids in the physical universe. In hindsight this is not surprising. As we will show in the next section only a handful of such disclination fluxoids are required to concentrate all the spatial curvature of a closed universe: it is obvious that one needs some quite heavy gravitational weaponry to accomplish this goal.

### C. The resolution: the splintering of the topological quantization of curvature

To set the mind, let us consider a cosmological thought experiment. Imagine a universe shortly after the big band which is spatially closed and characterized by a homogeneous and isotropic spatial curvature in the guise of FLRW cosmology. In this early epoch a phase transition happens where some form of matter crystallizes – proponents of dark matter as an elastic substance may view it as the present day relict of this primordial crystallization. How would the present day universe look like?

This is analogues to a "field cooled" type II superconductor: the magnetic field is the analogy of the curvature, and upon cooling through the phase transition a lattice of magnetic fluxoids is formed. The literal crystal gravity analogue would be that some lattice of quantized curvature fluxoids (corresponding with the disclination-cosmic string assemblies) would spring in existence. The precise nature of such "gravitational Abrikosov lattices" is an entertaining mathematical affair that we will illustrate in the final sections. However, we just became aware that in the physical universe a Planck mass has to be mobilized

to stabilize the "macroscopic" curvature concentrated in a microscopic volume associated with the core of the curvature fluxoid.

One can contemplate crystallization associated with the Planck scale itself where perhaps such a feat can be accomplished. But given the bounds implied by e.g. the graviton mass (Eq. 1) this is unrelated to the present epoch of the universe. The conclusion is that when the "cosmic crystal" is related to "mundane" matter it is impossible for it to form topologically quantized curvature fluxoids!

At first sight this may look similar as to type I superconductors where a Meissner phase is formed, expelling the magnetic flux altogether. The effect would be that all spatial curvature would be expelled when the solid forms – an alternative for inflation as resolution of the flatness problem. But this analogy fails. First, one has to find out a mechanism explaining why the core size (coherence length of the superconductor) becomes larger than the penetration depth $\lambda_G$. But much worse, the topological current associated with the crystal curvature (the defect density) as introduced in section VI B is more "flexible" than a vortex topological current.

Ironically, the net effect is that the gravitational version of this problem is actually closely related to the work of the soft matter community addressing solids covering rigid curved surfaces [27, 28, 53–57]. Because of the "absence of core mass" fundamental space-time becomes effectively rigid: it is just impossible to concentrate the overall curvature in microscopic areas. Dealing with the rigid surfaces the curvature in the background is fixed, take as example a sphere by a constant Gaussian curvature set by the radius of the sphere. The energetically most favourable way for the solid to accommodate this overall curvature would be to form a lattice of a handful of dislocations. However, the curvature in the background is distributed smoothly, and this cannot be matched with the localized curvature of the disclinations, thereby grossly violating the constraint Eq. (127).

However, as we stressed in section VI B, the quantity that is paired with the Einstein tensor is the defect density and this contains the contortion piece next to the disclination density, Eq. (115). To understand how this works it is useful to depart of the notion that a macroscopic assembly of dislocations with Burgers vectors pointing in the same direction occurring at a finite density suffices to represent crystal curvature. Using these as building blocks one can now distinguish the various curvature defects on basis of the "dynamics" of this dislocation system.

The way this works is discussed in the Appendix C 3 in a pictorial fashion. Consider first the wedge disclination in 2D using the cone folding procedure. When the opening angle is chosen to be the Franck scalar, away from the tip of the cone the cut-and-glue seam can be made invisible. The crystal lattice is everywhere restored except at the core: one has only to pay a core energy associated with this zero dimensional point.

One can however also decide to cut and glue such that the opening angle is smaller than the Franck scalar. It is no longer possible to "hide the seam": the best solution with regard to repairing the lattice as much as possible is now the *grain boundary*. This consists [73] of a linear array of approximately equally spaced dislocations with parallel Burgers vectors, see Fig. 10. The opening ("misfit") angle $\alpha$ is found to be,

$$\sin(\frac{\alpha}{2}) = \frac{B}{2h} \tag{134}$$

where $B$ is the amplitude of the Burgers vector (a lattice constant in this example) and $h$ the spacing of the dislocations which will be an integer number of lattice constants. This is energetically less favourable than the disclination since now the energy grows with the linear dimension of the system in 2D. Instead of the core being a point, it is now a line. Similarly in 3D, the disclination is a line like defect while the grain boundary is like a domain wall. But paying this prize one can avoid the topological quantization of the curvature flux as set by the Franck vector – the opening angle can be anything utilizing grain boundaries.

Fact is that with the exception of single crystals everyday solids are *littered* with grain boundaries, typically due to growth conditions: literal grains start to form first in the melt that meet each other. After annealing these relax into the array's of dislocations. These form closed manifolds that do not represent curvature: the cone example illustrates that as for the disclination the curvature is localized at the point in 2D (line in 3D) where the grain boundary comes to an end. A first take home message is that a grain boundary curvature defect is surely more costly than a disclination but in normal solids this energy difference is surmountable.

One already discerns that a similar feat can be accomplished by an "unorganized" gas of equal Burgers vector dislocations. Why is a grain boundary preferred? Generically, equal Burgers vector dislocations repel each other by the strain mediated ("stress graviton") interactions, raising the energy. It is easy to see that for a linear array (the grain boundary) these interactions are *screened*: there are no longer strains present away from the grain boundary (see ref. [74] for a recent precision analysis).

The moral is that very different from the topology at work in normal gauge theories the quantized topological defect (the vortex of a superfluid, here the disclination) can now "splinter" in building blocks which costs energy while the topological charge (the Franck vector) diminishes in the process. The central theme in the study of the "rigid balls" of the soft matter community is to find the best compromise between this energy cost and the requirement that the background curvature is smoothly distributed. The crucial control parameter is the ratio of the lattice constant $a$ to the curvature radius $L$ (e.g., radius of the rigid ball). A disclination lattice is the best solution when these are comparable; a famous example is the soccer ball or "buckyball" molecule that corresponds with a lattice of 5-ring dislocations in a hexagonal (six ring) crystal, see the next section. The soft matter community has been focussed on the study at intermediate $a/L$: complicated texture is found consisting of grain boundary fragments ("scars"), clouds of dislocations and so forth, e.g. ref.'s [53, 54, 56].

We are interested in the limit $a/L \to 0$ and there appears to be again a simple, universal solution that may not have been recognized by the soft matter community. This is easy to construct. Take an arbitrary point of the background manifold characterized by a near-infinitesimal local curvature that is frozen because of the 'lack of Planck mass", and associate it with the end of a grain boundary. This grain boundary has a diminishing misfit angle, and according to Eq. 134 the distance $h$ between the dislocations is diverging, and thereby the attractive interactions keeping the dislocation in the linear array configuration is diminishing. With a vanishing energy cost the dislocations can now glide freely away from the grain boundary – the direction perpendicular to the grain boundary corresponds with a slip plane. The results is a structureless gas of equal Burger vector dislocations. Repeat this operations at all other points in the manifold and the outcome is that the smooth background curvature is absorbed in a homogeneous dislocation gas that actually fully satisfies the confinement condition Eq. (112)

This concludes the physics part of crystal gravity, in so far we got with analyzing stationary situations. It is somewhat of an anti-climax. Getting at the heart of the GR interest – curvature – we established that all what happens is that the solid is infused with an extremely dilute gas of dislocations when the background manifold carries whatever form of spatial curvature. Physically this is not profound – any piece of steel carries in comparison a rather high density of dislocations albeit organized in a different way (grain boundaries, dislocation-antidislocation loops). This problem of core mass is in a way quite unfortunate since a universe where genuine curvature fluxoids can exist is a place of remarkable mathematical beauty and elegance. Let us conclude this treatise with an attempt to entice mathematicians to have a closer look.

## VIII.   THE IDEAL CRYSTAL GRAVITY UNIVERSES.

As we just argued, it is an unfortunate circumstance that in the physical universe the "topologization of curvature" is inhibited by the requirement that a Planck mass is necessary to stabilize the cores of the geometric curvature fluxoids. But the question arises, how would the analogue of the Abrikosov lattices of magnetic fluxoids as found in type II superconductors look like for geometrical curvature when these could form?

Although likely irrelevant for physics, this affair may

be of interest as a mere mathematical question. As we will explain it appears to point at surprising links between different branches of contemporary mathematics: discrete geometry, differential geometry/topology and algebraic topology. We will get so far that we can formulate some questions that appears to require mathematical machinery that is not available, suggesting however that a methodology may be around the corner that delivers answers as of intrinsic mathematical interest.

As point of departure, let us formulate the problem in the guise of Platonic perfection. The crucial assumption is to erase time and everything else is mathematical idealization:

1. Consider 2D and 3D space only-manifolds, avoiding the complication of the core mass associated with time. The "defect density = curvature" relation Eq. (112) defines the problem entirely.

2. The defect density is assumed to be coincident with the *disclination* density. As we explained this will impose the topological quantization in the form of curvature fluxoids quantized in units of the Franck vector.

3. The spatial manifold is filled with a single crystal that is perfectly periodic away from the curvature fluxoids cores. The space group of this crystal corresponds with the data specifying the nature of this matter. This is obviously an idealization – perfect single crystals exceeding even a weight of 1 kg are extremely rare in nature.

4. The curvature fluxoids will in analogy with the Abrikosov lattices formed by magnetic fluxoids form the most regular, symmetric lattices compatible with the circumstances including the condition that the fluxoids should be as far apart as possible. This is yet again an idealization. The regularity of the Abrikosov lattice of the superconductor is due to the fact that the density of fluxoids is such that the vortex lines are sufficiently close together for the mutual screened repulsive interaction to be still substantial. This is not the case for the curvature fluxoids: the range of their interactions shrinks to the lattice constant as related to the curvature confinement while the inter-fluxoid distances are of a "cosmological" dimension.

We will furthermore limit ourselves in the examples we discuss nearly entirely to the simple *homogeneous* closed Riemannian manifolds having the topology of $S_2$ and $S_3$ "balls" in 2- and 3 dimensions, respectively. It is surely possible to address hyperbolic ("open") manifolds, as well as more complicated topologies and geometries but we leave this to further study. This section is devoted to develop a first intuition on basis of wedge fluxoids. In the next section we will generalize it a bit further by focussing in on the role of the twist dislocations in 3D.

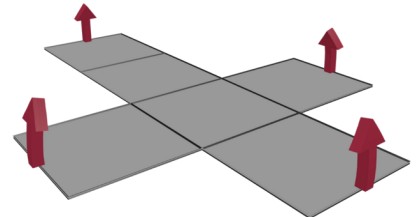

(a)Unfolded net of the cube, where the arrows indicate how to fold.

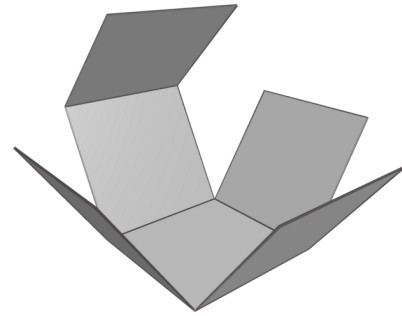

(b)Just before the cube is finished folding. All curvature is absorbed in the corners of the cube.

FIG. 1. How the net of a cube folds into a cube.

### A. Crystal gravity universes in two dimensions and the Platonic solids.

Let us first focus on the two dimensional case. The benefit is that the geometry is particularly easy to visualize by employing the embedding of the 2D manifold in 3D space – it is just about the surface of objects having the topology of a ball. We focus on the simplest of all curved manifolds: the maximally symmetric ball $S_2$, the simply connected compact manifold characterized by a constant Gaussian curvature $G(r) = 1/L^2$ where $L$ is the radius of the ball.

We can no longer afford to be cavalier with regard to the symmetry of the crystal. The space groups have been classified, and in two dimensions there are a total of 17 "wallpaper" groups. Let us consider first the simplest example: the square ("tetragonal") lattice, the simplest of all lattices with wallpaper group p4m. This is characterized by a four-fold rotational symmetry and accordingly the Franck vector of the elementary disclination is $\pi/2$. We may resolve this into a core that for positive crystal curvature corresponds with a triangle in the square lattice.

How to absorb the curvature of the ball by employing such curvature fluxoids? We learned that disclinations are constructed by cutting out Volterra wedges. This is in turn coincident with a procedure called "nets" in the discrete geometry literature, being a method to systematically classify polyhedra [76]. This is illustrated for the cubic case in Fig. 1(a). Take a flat sheet of paper

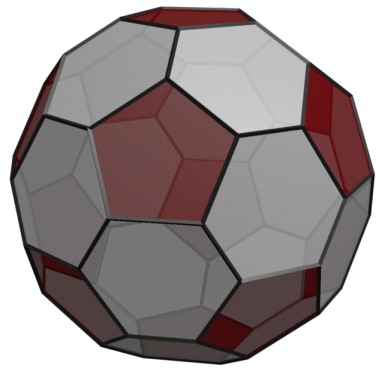

FIG. 2. A "soccer ball" made from hexagons and pentagons. 12 pentagons are needed to absorb the curvature of the sphere.

and cut it as indicated: the pieces that are removed coincide with the $\pi/4$ Volterra wedges associated with the disclinations. Fold it along the dashed lines under a right corner and glue the faces together: the outcome is that the ball has turned into a cube!

What happened with the (intrinsic) Gaussian curvature? The faces of the cube are obviously flat but the edges are characterized by one principal radius of curvature along the edge that is infinite: the edges also represent flat space. Another way of viewing it is that at the edges there is according to the "internal observer" of the crystal geometry no interruption of the square lattice: the crystal geometry is flat and thereby the background geometry as well. Henceforth, all the curvature is concentrated at the 8 vertices that obviously coincide with conical singularity type fluxoids characterized by an opening angle $\pi/2$ [77]

We conclude that the ball is "topologized" into a cube. It reveals a key insight. In this elementary example we find that by unleashing the topologization rules on the Riemannian ball having the tetragonal space group as input the curvature fluxoids form a "type II lattice" that is coincident with the birth place of (discrete) geometry: the cube, a platonic solid!

In fact, upon zooming in to resolve the microscopic lattice it actually corresponds with a *truncated cube* where the vertices are actually triangles: one of the Archimedean solids. This makes sense; Archimedean solids [78] are defined as convex uniform polyhedra composed of regular polygons (determined by the unit cells of the crystal ) meeting in identical vertices (the elementary disclinations, the triangle) excluding the 5 platonic solids (including the primitive cube).

Let us consider another example to get used to the idea. This one should have a particular appeal to physicists. Depart from the honeycomb lattice, overly familiar given its physical realization in the form of graphene. This is governed by the hexagonal wallpaper group P6mm characterized by a six fold rotation axis.

The elementary disclinations with positive Franck scalar are now 5 rings, characterized by an opening angle of $\pi/3$ (Fig. 7).

As we needed 8 disclinations to cover the solid angle of $4\pi$ in the case of the cube, now 12 $\pi/3$ disclinations are required. One can save the effort to construct a net because this was accomplished long ago by an anonymous shoemaker, see Fig. (2). The outcome is an icosahedron: this is recognized by the layman as a football, and by chemists as the 'buckyball' molecule formed from carbon. On the macroscopic scale this appears as the convex regular icosahedron, one of the platonic solids, characterized by 20 equilateral triangle faces, 30 edges and 12 vertices. Upon zooming in to the microscopic scale one finds that the 6 rings decorating the surface are replaced by a regular array of 12 5 rings, truncating the vertices: the Archimedean "truncated icosahedron".

The reader may already have discerned that we are dealing with the birth place of topology in the context of geometry: the Gauss-Bonnet theorem [79] in the incarnation of Euler's polyhedral formula [27, 80],

$$F - E + V = \chi \tag{135}$$

applying to compact manifolds characterized by the topological Euler characteristic $\chi$ ("number of handles"). $F$, $E$ and $V$ refer to the number of faces, edges and vertices of the polyhedron. The particular tessellation is determined by the wallpaper group including the disclinatons. For example, consider the icosehadron associated with the hexagonal crystal. The Euler characteristic of the ball is $\chi = 2$ and the number of pentagons required to absorb the curvature is easy to count, given that we tesselate the manifold by hexagons and pentagons. Write the number of faces as the number of hexagons $H$ and pentagons $P$ such that $F = H + P$. One edge is shared by two faces implying $E = (6H + 5P)/2$, while each vertex is shared by three faces such that $V = (6H + 5P)/3$. It follows from Eq. (135) that $P = 12$.

In this context, Gauss-Bonnet implies that the number of required disclinations is a topological quantity. All that matters that the manifold has genus zero (closed manifold with no handles). One can deform the manifold topologically. The form of the "type II lattice" will then chance but the number of curvature fluxoids will be the same. For instance, deform the sphere in an ellipsoid and for the tetragonal crystal the cube will turn into a rectangular prism.

Once again, departing from the idealized rules associated with Higgsing gravity we find that Riemannian geometry – it works in the same way in 3D as we will see in a moment – turns into the classic art of discrete geometry. This is in the form of the regular polyhedra that fascinated the classic greeks, enriched by elementary principles of algebraic topology. As input topological data (the Euler characteristic) and the group theoretical wallpaper data are required and this suffices to specify the shape of the polyhedron.

The question arises whether this can be completely classified in 2D. We suspect that the machinery to accomplish this is available in the rather intimidating mathematical literature (for physicists) dealing with 2D tilings. The prime candidate is in the form of John Conway's topological "orbifold" machinery [81, 82] given its remarkable capacity to classify completely the wallpaper groups in a flat background while it is inherently topological, departing from the Euler characteristic of the manifold.

## B.  The polytopes in three dimensions.

The generalization of non-Euclidean geometry to manifolds beyond two dimensions employing the powerful weaponry of differential geometry is the great accomplishment of Riemann. There is much more structure in higher dimensions and this is reflected in the crystal gravity topology in higher dimensions as we will illustrate in the remainder. We suspect that the latter represents a considerable challenge even for contemporary mathematics – this may be the occasion where crystal gravity is inspirational for advances at the frontiers of pure mathematics.

Different from the single scalar curvature invariant in 2D, there are a total of six invariants in 3D. The Weyl curvature is vanishing in 3D and therefore these are all associated with the Ricci tensor, or equivalently the Einstein tensor. We already found out that these precisely match the six components of the disclination density. The off-diagonal components of the latter tensor enumerate the twist disclinations which will be the subject of the next section. Although the picture is incomplete, by only considering wedge disclinations it is particularly easy to get a basic intuition regarding the essence of this affair.

In addition, we will only consider the maximally symmetric closed manifold $S_3$: the ubiquitous three dimensional ball characterized by a constant scalar curvature as e.g. the spatially closed universe of FLRW cosmology [30, 31]. Considering only wedge disclinations one anticipates a 3D extension of the wisdoms we identified in 2D. This is indeed the case and the topologized universes are identified as the generalization of the polyhedra to three dimensions: the *polytopes*.

Polytopes [83] were considered for the first time in the 19-th century by the mathematician Schläfli. As the discrete two dimensional geometry of a polyhedron is mapped on the surface of a three dimensional body embedded in a three dimensional space, one can as well map discrete geometry in three dimensions on the surface of a four dimensional body in 4D space. But the difficulty is that is much harder to use the "visual" methods of the greeks. One better relies on abstract algebraic counting methods as started by Schäfli, turning into a mathematical tradition that further developed the subject up to Coxeter who revitalized this field in the second half of the 20-th century [84, 85].

Let us consider an elementary example illustrating the principle. We depart from the primitive cubic spacegroup associated with a cubic lattice with a single "atom" in the unit cell. As we already noticed in this case we can rely on Cartesian coordinates to label the disclination densities: $\theta_{zz}$ refers to a disclination line propagating along the $z$ direction while the $\pi/2$ Volterra wedge is removed from the $xy$ plane. The $\theta_{xx}$ and $\theta_{yy}$ components are clearly equivalent. As in 2D we are seeking a maximally regular "type II" lattice of wedge curvature fluxoids absorbing the curvature of the ball. We learned in the previous section that the "point" 2D wedge fluxoids are "pulled out" in lines in 3D. One anticipates accordingly that the faces, edges and vertices of the 2D polyhedra are "pulled out" in "cells" (3D volumes, $C$), faces ($F$) and edges ($E$) in 3D. The edges are lines that can actually meet in 3D forming vertices ($V$). As the faces of the polyhedra represent flat parts in the 2D geometry, these are lifted up to the 3D "cells" that are also flat. The 3D edges are surely the curvature fluxoids carrying the 3D curvature: we learned in the previous section that these are lines in 3D. One anticipates that these have to propagate in the three orthogonal directions forming a cubic type II lattice themselves, given the homogeneous curvature on $S^3$ departing from a cubic crystal.

Polytopes are the most regular objects that can be constructed from these building blocks: the incarnation of the Platonic solids in 4D. We are interested in the regular convex 4-polytope and these are tabulated [86, 87] and we can look up which qualify. Remarkably, only 6 exists and one qualifies: the generalization of the cube to 4 dimensions, called the *tesseract* [88].

Schäfli found out that the Euler polyhedral formula Eq. (135) generalizes to the 3D surfaces as,

$$-C + F - E + V = \chi \tag{136}$$

where the Euler characteristic of the $S^3$ ball $\chi = 0$ – notice that this is the discrete geometry version of the Poincare conjecture for the continuum that was proven only very recently. One finds that the tesseract is characterized by a number of cells $C = 8$, faces $F = 24$ , edges $E = 32$ and vertices $V = 16$. This makes sense. The discretized curvature is counted by the edges. These do correspond with the vertices on the cube, that have to be pulled out in lines that have to occur in the three orthogonal directions in 3D. The cube has 8 vertices, which therefore have to turn into 32 edges on the 4 dimensional version of the cube.

It is entertaining to find out how to image polytopes. As in 2D (Fig. 1(a)) we can also employ in 3D the cutting and gluing "nets" procedure, paying the price that the visual system gets increasingly strained. Instead of the squares, one departs from cubes that will form the cells of the tesseract. Depart from a flat space arrangement of cubes having the same structure as the 2D version: the "Dali cross" constructed in Fig. 3(a). Instead of

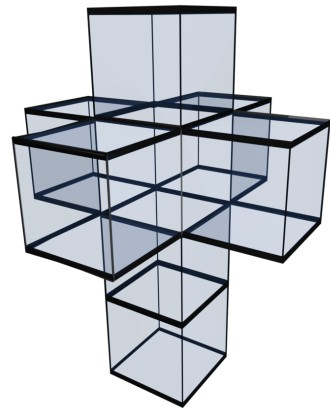

(a)The Dali cross is the 3D net of a tesseract

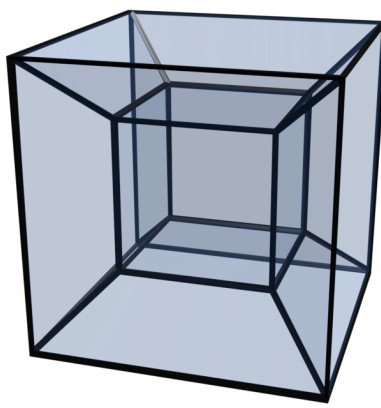

(b)Projection of a completed tesseract. Here, curvature is also present in the lines where the cubes intersect.

FIG. 3. How the net of a cube folds into a cube.

glueing the edges of the squares one has now to glue the faces of the cubes and it is easy to see that these glueing surfaces do not carry intrinsic curvature for the same reason that the edges of the cube represent flat geometry. But the vertices in 2D turn into the edges in 3D and these correspond with "cosmic strings" with opening angle $\pi/2$ in the plane perpendicular to the propagation direction: we have identified the wedge curvature fluxoids. As we already stated, a total of 32 of such fluxoids suffices to absorb all the curvature of the ball. As in 2D, upon zooming in to the microscopic scale we will find out that these edges are truncated into a row of triangles: the tesseract is actually truncated.

Last but not least, the construction shows that the fluxoids propagating in the $x, y$ and $z$ directions *have* to cross each other at the 16 vertices. In order to absorb the curvature of the ball in a "cubic way" it is obvious that fluxoids propagating in the three orthogonal directions are required, but that they have to cross 16 times is a less obvious necessity.

Another informative way to image the tesseract is in the form of the *Schlegel diagram* [89] . As a 2D projection, it can be employed to give an impression of the field of view (in perspective) of a traveller traversing the tesseract universe, see ref. [90]. Such an observer would just encounter a small number (32) of straight lines orientated in the three orthogonal directions that he/she can be detect by the same interference signals signalling cosmic strings. Alternatively, flying by the string his/her orbit (geodesic) would acquire a sweep anchored in the plane perpendicular to the propagation direction of the string (see next section). Notice that the gripping weirdness of such tesseract movies has earned it a place in popular culture, a recent highlight being the tesseract found at Gargantua's singularity in the movie "Interstellar", undoubtedly child of co-producer Kip Thorne's imagination.

The tesseract is an elementary example illustrating the principle. In general one should depart from the space group of the crystal. This adds quite some complexity since there are 230 space groups in 3D. For simple crystals where one can ignore the structure inside the unit cell one only needs the information regarding the directions of the lattice axis: the point group symmetries associated with the Bravais lattice. Both the propagation direction and the Franck vector are oriented along the lattice vectors. Considering e.g. the hexagonal $(D_{6h})$ crystal structure. The "base plane" is associated with the "honeycomb" $\pi/3$ Franck vectors while the vertical direction is governed by a two-fold rotation axis and the associated $\pi$ disclinations. One infers that this should imply a quasi-regular polytope characterized by a curvature fluxoid lattice that will look quite different in different directions, reflecting that the curvature fluxoids have a quite different topological charge pending the inequivalent lattice directions.

The classification of 4D polytopes is a much more recent affair in mathematics than the 3D polyhedra. Started with Schaefli in the 19-th century their study continues until the present day, involving famous contemporary mathematicians such as Coxeter and John Horton Conway. Does this available machinery suffices to exhaustively classify all maximally regular "coverings" of $S_3$ in terms of wedge curvature fluxoids departing from the 230 space groups? Although by itself quite a challenge, it is actually in a critical way stll oversimplified as we will now find out.

## IX. CURVATURE IN THREE DIMENSIONS AND THE TWIST DISCLINATIONS.

The notion that the type II curvature fluxoid lattices are associated in 3D with polytopes is an essential one. By only considering the wedge fluxoids as we did in the previous section we could just get away with the "visual" nets and Schlegel diagrams. This is an appealing way for

the visual system to recognize how the polyhedra generalize to polytopes.

However, given the non-Abelian group of rotations in 3D the curvature fluxoids as rooted in the rotational structure acquire non-Abelian traits in their topology. This is encapsulated by the existence of the *twist* disclinations. We encountered the general distinction between edge and screw dislocation. Departing for simplicity from a cubic lattice with its Cartesian preferred frame, one can orientate the Burgers vector in a direction orthogonal to the propagation direction or either in a parallel direction. This refers to the off-diagonal- (edge) and diagonal (screw) components of the dislocation density tensor, respectively. The basic rules are similar for the disclination densities: both the Franck vector and the propagation direction orientate along the (cubic) lattice directions. When the Franck vector points in the same direction as the propagation direction (diagonal entries disclination density) one recognizes the "cosmic string" like wedge disclination highlighted in the previous section. But one may as well orientate the Franck vector in the two lattice directions orthogonal to the propagation direction, see Fig. (8): we call these the "twist" disclinations, identified with the off diagonal entries of the disclination density tensor.

Although recognized in the elasticity literature, these have had all along a rather marginal existence given that free disclinations cannot exist in solids. These become physical in liquid crystals but it turns out that the nematic-type crystals that are realized in the laboratory (doing the hard work in liquid crystal display technology) are of a special "uniaxial" kind. This is the usual affair with rod-like molecules and this turns out to be associated with an exceptional *Abelian* $D_{\infty h}$ 3D point group. One can identify such order with any 3D point group, but this implies high rank tensor order parameter theories that are barely explored [64, 65]: the topological sector is as of yet completely uncharted (see also ref. [91]). This general theme of non-Abelian topology is related to the notions behind topological quantum computation [92, 93].

We do not have the ambition here to make a serious headway in this intriguing subject. Instead, in Section IX A we will present an explicit construction of the twist curvature fluxoids where we are much helped by the availability of the background metrics in the literature [94]. Subsequently, we will construct the holonomies associated with the wedge- and twist fluxoids as an illustration of the non-Abelian topology: this shows that the geodesics describing e.g. the trajectories of test particles are different pending in which order the various fluxoids are encountered (Section IX B).

The study of homogeneous geometries in 3 dimensions is an active field of mathematical research. A relatively recent highlight is the Thurston classification pertaining to closed manifolds, demonstrating it to be a much richer affair than in 2D. This in turn has bearing on the "polytope" fluxoid lattice question, enriched by the non-Abelian topology of the curvature fluxoids. To give a first impression we consider in Section IX C the simple case of a Kantowski-Sachs "cylinder" geometry to subsequently reconsider the ball $S^3$ of the previous section. Due to the anisotropy in the background geometry, we find in the first case a simple linear array of exclusively wedge fluxoids. On the maximally symmetric $S^3$ we do find however that the "tesseract" fluxoid lattice is actually characterized by a six fold degeneracy reflecting the non-Abelian nature of this affair.

## A. Constructing the twist fluxoids.

In the previous section we discussed the way that *wedge* disclinations merge with a cosmic string metric into "wedge curvature fluxoids". We found out that these are the building blocks to construct the "gravitational Abrikosov lattice" departing from a homogeneous- and isotropic "ball" background, becoming polytopes such as the tesseract dealing with a cubic crystal. But we learned in section VI that there are in total six independent components of the defect/disclination density in 3 space dimensions. Let us again take the cubic crystal as minimal example so that we can use Cartesian coordinates to label the disclination density. For the tesseract we mobilized an equal number of wedge fluxoids propagating in the $x, y, z$ directions. These correspond with the *diagonal* components of the disclination densities $\theta_{xx}, \theta_{yy}, \theta_{zz}$ that appear on the same footing in absorbing the background curvature. But there are in addition the three *off-diagonal* components $\theta_{ij} \neq 0, i \neq j$ as independent curvature parameters.

How to construct the curvature fluxoids associated with such an "off-diagonal" curvature? Let us first recall the meaning of the off-diagonal components of the disclination density tensor. In the cubic crystal the disclination line propagation direction and Franck vector are both pointing along the "cartesian" lattice directions $x, y, z$. Those of interest are characterized by a Franck vector being orthogonal to the propagation direction of the disclination line (see Fig. (8): the twist disclinations.

Recall the definition of the disclination density Eq. (118) and choose a disclination propagating in the z-direction to find,

$$\theta_{zk}^{twist}(\vec{x}) = \Omega_k \delta_z(L, \vec{x}). \qquad (137)$$

where the Franck vector $\vec{\Omega}$ is either oriented in the $k = x$ or $y$ direction, corresponding with a rotation in the $yz$ or $xz$ plane. It is immediately clear that a generalization of the conical singularity type background geometry of the wedge fluxoid is required that can be married with the twist disclination.

It is a fortunate circumstance that such "twist" generalizations of the cosmic string metric were explicitly constructed some time ago by Puntigam and Soleng

("PG") [94]. These authors were aware of crystal geometry, albeit obviously not of crystal gravity, and they asked the question of how to construct the equivalent geometry in the Riemannian manifold. PG mobilized a rather sophisticated differential geometry machinery to get at the results: Moebius type matrix representation differential geometry, soldering together Cartesian metrics to translate the Volterra constructions for the twist sector into explicit metrics. As for the wedge disclination

one removes a wedge quantized by the Franck vector, to glue the surfaces back again but now in a direction involving a direction parallel to the propagation direction of the disclination line. This is a bit of a challenging operation for the visual system (again, Fig. 8)

In this way PG derived metrics that precisely satisfy what we need. Consider a defect propagating in the z-direction and the metrics associated with the pair of twist disclinations Eq. 137 turn out to be,

$$ds^2 = -dt^2 + dx^2 + dy^2 + dz^2 + 2\frac{\alpha}{r^2}(zdy - ydz)(xdy - ydx) + (\frac{\alpha}{r^2})^2(y^2 + z^2)(xdy - ydx)^2$$

$$ds^2 = -dt^2 + dx^2 + dy^2 + dz^2 + 2\frac{\alpha}{r^2}(zdx - xdz)(xdy - ydx) + (\frac{\alpha}{r^2})^2(x^2 + z^2)(xdy - ydx)^2 \tag{138}$$

in Cartesian coordinates, where $r^2 = x^2 + y^2 + z^2$. These describe a string-like geometry propagating in the spatial $z$ direction while as for the wedge variety these correspond with topological curvature defects where the space away from the core is locally flat. The associated curvature is parametrized in terms of a single parameter $\alpha$ having the same status as the deficit angle in the wedge cosmic string geometry.

Computing the Einstein tensors associated with these metrics one finds,

$$G_{xk} = 2\pi\alpha\hat{n}_k\delta_z(L, \vec{x}). \tag{139}$$

where $\hat{\vec{n}}$ is a unit vector. For the first- and second metric in Eq. (138) this vector is oriented in the $x-$ and $y$ direction, respectively. We see that by associating $|\Omega| = 2\pi\alpha$ we fulfil the confinement condition $G_{ij} = \theta_{ij}$: the opening angle gets quantized in units of the Franck vector, determined by the point group symmetries. As for the wedge fluxoids these require a Planck mass to stabilize their cores, rendering them to be irrelevant for the physical universe.

## B. Non-Abelian topology: the holonomies of the curvature fluxoids.

Again, these metrics are locally flat everywhere except at the core where the quantized curvature is localized. To illustrate the topological nature of the ensuing twist curvature fluxoids let us focus in on the holonomies. Consider a closed loop around such a fluxoid, to observe how a vector is parallel-transported around such a loop. Orient this loop in a plane perpendicular to the propagation direction of the fluxoid, i.e. the $xy$ plane for the metric Eq. (138). The change in a vector $R$ being parallel-transported around such a loop can be written as

$$R'^\mu = \mathcal{G}^\mu{}_\nu R^\nu \tag{140}$$

where $\mathcal{G}$, the *Frank matrix*, is the path ordered exponential of the integral of the *Lorentz connection* $^{(L)}\Gamma^\mu{}_\nu$ when transported around the closed loop $S$:

$$G^\mu{}_\nu = \mathcal{P}\exp\left(\oint_S {}^{(L)}\Gamma^\mu{}_\nu\right). \tag{141}$$

Keeping an eye on the time direction as well with the string in the rest frame, let us first consider outcome for the now familiar wedge dislocation with the metric Eq. (130). The Lorentz connection becomes,

$$^{(L)}\Gamma^\mu{}_\nu = \begin{pmatrix} 0 & 0 & 0 & 0 \\ 0 & 0 & -1 & 0 \\ 0 & 1 & 0 & 0 \\ 0 & 0 & 0 & 0 \end{pmatrix} \frac{\alpha}{2\pi(x^2 + y^2)}(xdy - ydx). \tag{142}$$

and the Franck matrix describes a rotation by the angle $\alpha$ in the $xy$ plane,

$$\mathcal{G}^\mu{}_\nu = \begin{pmatrix} 1 & 0 & 0 & 0 \\ 0 & \cos\alpha & -\sin\alpha & 0 \\ 0 & \sin\alpha & \cos\alpha & 0 \\ 0 & 0 & 0 & 1 \end{pmatrix}. \tag{143}$$

This implies that the trajectory of a particle determined by its geodesic approaching the wedge fluxoid in the plane perpendicular to its direction will stay in this plane, although it will change direction in encircling the fluxoid by an amount set by the deficit angle. This lies at the origin of the lensing effects making it possible to observe cosmic strings.

Let us now see what happens in the presence of a twist fluxoid. For the first metric in Eq. (138) the Lorentz connection takes the form

$$^{(L)}\Gamma^\mu{}_\nu = \begin{pmatrix} 0 & 0 & 0 & 0 \\ 0 & 0 & 0 & 0 \\ 0 & 0 & 0 & -1 \\ 0 & 0 & 1 & 0 \end{pmatrix} \frac{\alpha}{2\pi(x^2 + y^2)}(ydz - zdy). \tag{144}$$

and the Franck matrix describes now a *rotation* by an angle $\alpha$ in the $y - z$ plane:

$$\mathcal{G}^{\mu}_{\ \nu} = \begin{pmatrix} 1 & 0 & 0 & 0 \\ 0 & 1 & 0 & 0 \\ 0 & 0 & \cos\alpha & -\sin\alpha \\ 0 & 0 & \sin\alpha & \cos\alpha \end{pmatrix}. \qquad (145)$$

Similarly, for the second metric in Eq. (138) one finds a rotation in the $x - z$ plane. In other words, where the wedge disclination causes a deficit angle around an axis parallel to the disclination line, the twist disclinations induce a similar deficit angle around one of the axis perpendicular to the disclination line as illustrated in Fig. 8. This implies that a particle approaching a twist fluxoid in the xy plane, will upon encircling the fluxoid be swept out of this plane by an amount set by $\alpha$. Pending whether one is dealing with an "$xz$" or "$yz$" (or an arbitrary combination of the two) the trajectory will be also altered with regard to its approach in the $xy$ plane.

These holonomies signal the *non-Abelian* nature of the topology in this geometrical setting, reflecting the non-Abelian nature of the rotation group in 3D. The outcomes are pending the order of the different operations. Imagine a space containing two straight curvature fluxoids lines in parallel orientation. Consider a probe particle on a geodesic trajectory approaching these fluxoids in the plane orthogonal to the propagation direction. The first fluxoid this particle encounters is of the wedge kind and accordingly the particle will chance its direction, staying however in the plane of approach. Subsequently, it encounters a twist dislocation: we learn from its holonomy that the particle will be slung out of the plane of approach. But now consider the same situation except that the particle first meets the twist- and then the wedge fluxoid. One infers immediately that the trajectory of the particle leaving this "system" will be very different compared to the first case.

### C.  Anisotropic curvature: the cases of Kantowski-Sachs and the 3D ball.

We have established that the nature of curvature *fluxoids* in 3D is a quite rich affair: there are a total of six distinguishable of such fluxoids that can be in turn distinguished in terms of their wedge- or twist character. There is a lot more going on compared to superconductors characterized by a single form of fluxoid characterized by a scalar charge (flux quantum). The latter is of course due to the fact that there is only one kind of spatial gauge curvature in electrodynamics – the magnetic field. In a type II superconductor this magnetic field gets just chopped up in an array of quantized fluxoids.

Geometry in 3D including the relations to topology is a famously interesting and challenging affair, forming a central subject of study in pure mathematics until the present day. It was already understood in the 19-th century that in 2D three geometries are fundamental: the sphere $S^2$, the euclidean plane with no curvature $E^2$ and the hyperbolic plane $H^2$. But how to generalize this to the much richer three dimensional case? A crucial progress is due to Thurston who arrived at a classification of homogeneous closed 3D geometries in the late 1970's [95], identifying 8 different classes. This played a key role via his "geometrization conjecture" to the famous achievement by Perelman proving the Poincare conjecture. This boils down to the notion that regardless the initial conditions, during the evolution ("Ricci flow") the geometry will homogenize into one of Thurston's "model geometries" (see, e.g. [96]) with its topological ramifications (e.g., finding closed "sphere" like manifolds).

This Thurston classification also clarifies the much older Bianchi classification of anisotropic homogeneous 3D manifolds [97], and the "exceptional" Kantowski-Sachs case [98] which are more familiar to physicists. These can all be identified within Thurston's fundamental classification – in Ref's [99, 100] this is explained for physicists. A crucial additional in Thurston's scheme is the connection to topology: his classes are the ones that can be realized on *closed* manifolds. Apparently the classification of 3D geometries on open, hyperbolic manifolds is still an open problem. We will ignore such manifolds here all together, leaving it to a future effort.

Once again, the question is regarding the nature of the "Abrikosov lattice" of curvature fluxoids. The essence is that instead of having one homogeneous form of spatial gauge curvature as in electromagnetism – the magnetic field – there are now the eight different kinds of curvatures according to Thurston as if there are now eight different kinds of magnetic field. The curvature fluxoids themselves occur in six different varieties with their tensor properties controlled by the space groups of the solid, distinguishable through their wedge- or twist nature illustrating the non-abelian nature of the topology as rooted in the 3D rotational/point group symmetries. Is it possible to classify the possible forms of "gravitational Abrikosov lattices"? As we argued in the previous section this may reveal hitherto unrecognized connections between discrete geometry – the polytopes – and differential/algebraic geometry and topology.

This may be the best way to formulate the mathematical challenge. Our ambition is here very limited: let us just consider the simplest cases to trigger the curiosity of the reader. Before we revisit $S^3$, the 3D ball that we already looked at the previous section let us first consider the Kantowski-Sachs case which appears as the simplest possible option.

In the previous section we were a bit cavalier in the construction of the tesseract. We should inspect the Einstein tensor of the background geometry to find out how to topologize the curvature using the fundamental topological rule: the components of the Einstein tensor have to be in one-to-one relation with the components of the disclination density, $G_{ij} = \theta_{ij}$. Let us see how this works

in Kantowski-Sachs space or Thurston type 4 [100].

The space-time version has a long history in cosmology but we are here only interested in the 3D spatial manifold having as metric,

$$dl^2 = a^2 dx^2 + b^2 \left( d\theta^2 + sin^2\theta d\phi^2 \right) \qquad (146)$$

in terms of cylinder coordinates with $a^2, b^2$ positive constants. This is just a 3D cylinder, a simply connected space with the topology $R \times S^2$ where $R$ is the straight line and $S^2$ the sphere. We assume that the crystal has cubic symmetry and therefore we need its Einstein tensor in Cartesian coordinates $\mu, \nu = x, y, z$,

$$G_{\mu\nu} = \begin{pmatrix} -a^2/b^2 & 0 & 0 \\ 0 & 0 & 0 \\ 0 & 0 & 0 \end{pmatrix} \qquad (147)$$

Looking at the spatial section, this has the topology of a 3D cylinder: $R \times S^2$, where the axis of the cylinder is in the $x$ direction. For every $x$ an $S^2$ is realized in the $yz$ plane with its scalar curvature parametrized by $-a^2/b^2$. It is immediately clear how to construct the polytope: the intuitive procedure we used in the previous section applies immediately. For a specific $x$ we are just dealing with the 2D sphere turning into the cube where the corners correspond with the 2D wedge fluxoids. These are now pulled out in the x-direction turning into a cubic array of straight 3D wedge fluxoid lines propagating in the x-direction. It is also immediately obvious that this absorbs the curvature embodied by the only non-zero "wedge" component of the Einstein tensor $G_{xx}$.

This is of course coincident with what we would expect from the "net" procedure in the previous section when we were only aware of the wedge fluxoids. But let us now revisit the 3D ball. By itself there was nothing wrong with the net/Schlegel diagram construction showing the tesseract formed entirely from wedge fluxoids. However, in this maximally symmetric space we better be aware that there is room as well for twist fluxoids. Let us first inspect the Einstein tensor of $S^3$. Given a radius $R$ it is a textbook wisdom [30] that the Einstein tensor is proportional to the metric, in 3D hyperspherical coordinates $r, \theta, \phi$

$$dl^2 = dr^2 + R^2 \sin^2(r/R)(d\theta^2 + \sin^2\theta d\phi^2) \qquad (148)$$

But since the fluxoids are quantized according to the cubic symmetry of the crystal we need the corresponding Einstein tensor again in the awkward Cartesian coordinates, $\mu, \nu = x, y, z$ while $R^2 = x^2 + y^2 + z^2$

$$G_{\mu\nu} = \frac{1}{R^2-1} \begin{pmatrix} 1-y^2-z^2 & xy & xz \\ xy & 1-x^2-z^2 & yz \\ xz & yz & 1-x^2-y^2 \end{pmatrix} \qquad (149)$$

This shows that in this cubic frame there is an equal amount of "wedge- and twist sourcing" in the background, not surprisingly since the isometry of the geometry is maximally isotropic. The implication is that we may as well employ the twist disclinations to absorb the curvature.

It is a fundamental rule that because of the cubic symmetry of the crystal and the maximal isotropy of $S^3$ a tesseract has to form with the 32 edges corresponding with the fluxoid lines orientated in the $x, y, z$ directions. In the previous section we assumed that the Franck vectors would point in the same directions as their propagation direction forming wedge fluxoids characterized by $\theta_{ii}$ absorbing the $G_{ii}$ curvature in the background. But we have now the freedom to orientate the Franck vectors in the two directions orthogonal to their propagation direction since we learned that the fluxoids also exist in the twist form.

Although the disclination density are symmetric, their propagation direction and the Franck vectors are of course distinguishable given a realization of the type II "tesseract" lattice. Let us therefore use capital symbols $X, Y, Z$ to indicate the propagation direction and $x, y, z$ for the orientation of the Franck vector. The crystal symmetry imposes that the disclination lines should always correspond with the edges of the tesseract, being equally orientated along the $X, Y, Z$ directions. We discovered in the previous section that the curvature can be absorbed entirely by wedge disclinations, corresponding with $Xx, Yy$ and $Zz$ components of the disclination currents in this "asymmetric" notation. However, let us keep a wedge disclination in the X-direction absorbing the "$x$" (Franck vector) curvature but instead of employing the wedge disclinations in the $y, z$ directions we may as well absorb the curvature in the $z$ direction by using a $Y$ twist disclination and the other way around: we can label this as a $Xx, Yz, Zy$ Tesseract. We can in the same way identify $Xz, Yy, Zx$ and $Xy, Yz, Zz$ "wedge and twist mixtures". Finally there are two pure twist tesseracts: $Xy, Yz, Zx$ and $Xz, Yx, Zx$. The conclusion is that the Tesseract is actually sixfold degenerate! This is clearly rooted in the non-Abelian nature of the rotational properties of the cubic lattice together with the maximal isometry of $S^3$.

The take home message is that the relations between the polytopes, crystal space groups and Riemannian geometry are further enriched by these interesting intricacies emerging from the non-Abelian rotational symmetry of the crystal point group that in turn communicate directly with the nature of the anisotropic curvature classified by Thurston for compact 3D manifolds. We just scratched the surface in this first encounter with the "gravitational Abrikosov lattices", signalling that there is a profound mathematical landscape to be explored. We hope that this will stimulate others to have a closer look.

## X.   DISCUSSION AND OUTLOOK.

Arriving at the end of this first exploration of crystal gravity, the question arises: what it is good for? Although we percieved this adventure as a splendid form of physical (and, potentially, mathematical-) recreation on basis of what we have found out so far we are not convinced that any of it will be of grave consequence.

The recreation aspect is perhaps most obvious in the later "mathematical" chapters dealing with the gravitational Abrikosov lattice in two- and three dimensional spaces with Euclidean signature. A common thread in the development has been in the ease to understand matters intuitively, with the pattern recognition capacity of the human visual system being of great help in directly recognizing what is going on. The ball turning into a cube by the "higgsing of the geometry" by a square lattice crystal in two dimensions (Section VIIIA) is case in point. It is directly obvious that the intrinsic curvature of the sphere is absorbed in the corners of the cube. The entertaining part is that this is precisely coincident with the Higgsing of the curved geometry where the corners have the status of "curvature fluxoids", deep inside governed by the same principles responsible for the Abrikosov flux lattice in superconductors. Isn't it so that the "cube" is by the simplest way to explain to students the general workings of fluxoids in general?

There is surely still the exercise to be completed of classifying all "gravitational Abrikosov lattices" in 2D, departing from the isometry and topology of the background manifold and the space group of the crystal. But this is literally an exercise: the mathematical counting device to accomplish has been available in the 19-th century in the form of Euler's polyhedral formula Eq. (135).

Geometry in three Euclidean dimensions is however a different affair: it is still a frontier of contemporary mathematics. In a recent era there has been substantial progress in the classification of polytopes like the tesseract while the study of 3D curved manifolds has been on the foreground with Perelman's proof of the Poincaré conjecture resting on Thurston's classification of closed manifolds. In addition one has to cope with the richness of the 3D space groups. Last but not least, one has to deal with the non-Abelian nature of the curvature fluxoids as we discussed in the previous section. These are just observations indicating that there is a lot going on. However, in the 19-th century Schäfli already wrote down the 3D generalization of Euler's polyhedral formula, Eq. (136). This suggests that by just departing from tessellations determined by the space group one can keep track of how the intricacies of Thurnston's fundamental geometries, the "non-Abelian" degeneracies and so forth of the "polytope type II lattice".

Whether any of it will ever be of grave consequence to physics is unclear to us. To stand any chance one has to assert the presence of a solid of cosmological dimensions – the arguments in section IV insisting that the gravitational penetration depth beyond which the crystal gravity effects become manifest has to be expressed in units of light years. One can contemplate that dark matter is elastic, an idea that has repeatedly been forwarded in the cosmology community. When the "tesseracts" would have a physical existence there would have been potential for grand consequence. It could then have offered a surprising mechanism explaining the cosmological flatness problem. Spatial curvature would have been expelled everywhere except for a handful of cosmic string like curvature fluxoids. The probability for any of these to be observable from the earth would become near infinitesimal.

But the Lorentzian time-axis interferes. As we discussed in Section VII, the gravitating core of the fluxoids would require Planck scale energy density to be stabilized, a requirement which cannot possibly be combined with the rather mundane scales governing any form of "cosmological solid" as implied by e.g. the bounds on the graviton mass. At first sight it appears as a paradox: the non-linear nature of geometrical curvature and crystal curvature implies the mutual confinement. But the same non-linearity rooted in the semi-direct relation between translations and rotations offers a simple but boring solution. The topological current associated with crystal curvature – the defect density – reflects this non-linearity in the form of "lego brick topology". The dislocations with their topological Burgers vector quanta play the role of the bricks that together with a rather general building regulation ("Burgers vector polarization") may represent rotational "topology" – only the most sturdy constructions (disclinations) re-establish topological quantization. In the solid dark matter universe we predict by lack of alternatives the maximally structureless outcome – the dislocation gas.

On a side, there is yet another form of matter that breaks space-time symmetry that may be looked at more closely: liquid crystals. These are substances that maintain translations but break the rotational invariance. These can be viewed as descendants of crystals: upon proliferating dislocations translational symmetry is restored but as long as these occur such that the Burgers vectors are locally antiparallel the rotational symmetry breaking is maintained. Disclinations are the natural topological defects while the "torque rigidity" is now deconfined and these disclinations relate to the background curvature by the same "Coulomb force" rule as for e.g. magnetic fluxoids. But also in this context one runs into the Planck scale core problem associated with the curvature fluxoid cores. This amounts to a conundrum: typical liquid crystals are formed from rod-like molecules and it is far from clear how to identify the "gas of equal Burgers vectors dislocations" dealing with such microscopy.

Let us reconsider the main limitation of this study: we have been entirely focussed on *stationary* geometries. The core-business of GR is of course revolving around *dynamical* evolutions. The construction of a dynamical theory of crystal gravity is a considerable challenge. One has to combine the relativistic rigor of the old "relastic-

ity" constructions with the demands of symmetry breaking including the demise of Lorentz invariance. Surprises cannot be excluded but we are not optimistic. After all, the main effects pertain to spatial curvature and we found out that this is eventually captured by the boring dislocation gas. In other regards, we expect that there is not much of a difference between solid matter as compared to the usual liquids in dealing with black holes, cosmological evolutions and so forth.

What remains are a number of rather practical results that we presented in the early Sections IV and V. A first issue relates to the question whether dark matter could be an elastic substance. We have the impression that this can be easily excluded or confirmed using available astronomical surveys. It is not clear to us whether it is fully realized by cosmologists that the distinguishing characteristic of an elastic manifold is in its shear rigidity. Visible matter has the capacity to exert stress on dark matter (and vice versa) through gravity. The distinguishing characteristic of *elastic* dark matter should be that it will exert a restoring force in response to a shear stress build up by visible matter through the gravitational force. As highlighted repeatedly, shear stress is associated with *quadrupoles*: one therefore expects that the *spin 2* components in the large scale distributions of visible matter would be suppressed. It is up to the astronomers to further explore this venue when the need arises.

Yet another practical matter is our observation in Section VE that gravitational waves have the capacity to be absorbed by mobile dislocations that occur in the bulk of a solid. Obviously, this may become of relevance only in extreme conditions. For instance, the pinning energy of dislocations in Weber style gravitational wave detectors will exceed by many orders of magnitude the shear stress caused by a passing gravitational wave. Yet again, this may be of interest dealing with a rocky planet that is littered by grain boundaries being in proximity of a merging black hole binary.

Finally, let us turn to the initial motivation for this work: can an "elasticity-gravity" holographic duality be constructed in analogy with the greatly successful fluid-gravity duality? With the latter it is demonstrated that the Navier-Stokes equations describing the hydrodynamical fluid in the boundary can be directly related to the gradient expansion near-horizon gravitational dynamics in the deep interior of the holographic bulk. Can a similar procedure be formulated, relating the elasticity governing the macroscopic properties of the boundary crystal (having a similar status as the Navier-Stokes equations of the fluid) to the deep interior gravitational physics in the bulk?

The point of departure is the matter of principle that the translational symmetry breaking is "relevant towards the IR": the inhomogeneity is increasing along the holographic radial directions moving from the boundary to the deep interior: the crystal lattice is largest at the black brane horizon. The difference with the boundary is that this bulk crystal lives in a dynamical GR background where Newton's constant is of "order 1": the gravitational penetration depth $\lambda_G$ becomes microscopic. This is crystal gravity territory.

One is in first instance interested in linear response – the probe limit – and the required machinery is found in section IV. In particular, the stress-formalism with its "stress graviton" tensor gauge fields should be particularly convenient, encoding the degrees of freedom of the crystal in the same mathematical language as the gravitons themselves. But there is yet another complication: the extra dimension of the holographic bulk in the form of the radial direction.

Along the radial direction the symmetry of the crystal is frozen and the lattice constants etcetera are the same in the deep interior as near the boundary. The amplitude of the modulation is just increasing towards the "deep infrared". Considering the overall symmetry of this bulk, the radial direction is similar to the time axis. Translations and rotations are broken only in the space directions shared by bulk and boundary. The Lorentzian time direction stays homogeneous but in the holographic bulk there is in addition the radial direction that is also homogeneous. The radial direction is like an additional time axis, albeit now with an Euclidean signature. It is of course also the key dimension associated with the Anti-de-Sitter curvature and the holographic dictionary. This amounts to a considerable complication even on the probe level: one has now to accommodate next to the time axis also an "anisotropic" Euclidean radial direction, introducing more of the kind of difficulties one encounters with the time axis (Section IVG). It remains to be seen whether this suffices to "pull" the near horizon dynamics along the radial axis to the boundary which is the central wheel in fluid-gravity duality.

## Appendix A: Conventions

We employ the "mostly plus" metric signature, the Minkowski metric tensor is $\eta_{\mu\nu} = \mathrm{diag}(-1, 1, 1, 1)$.

The partition function, action and Lagrangian are related as $Z = \int \mathcal{D}\{\text{fields}\} \exp(\mathrm{i}S/\hbar)$, $S = \int \mathrm{d}t\mathrm{d}^3x \, \mathcal{L}$. We use ordinary SI units, so that for instance the Maxwell Lagrangian is:

$$\mathcal{L}_{\text{Maxw}} = -\frac{1}{4\mu_0} F_{\mu\nu} F^{\mu\nu} + J_\mu A^\mu$$

$$= \frac{1}{2\mu_0}(\frac{1}{c}^2 E^2 - B^2) - V\rho + J_m A_m. \qquad (A1)$$

Many calculations are carried out in imaginary time $\tau = \mathrm{i}t$. For the Matsubara frequency $\omega_n$, this leads to the analytic continuation:

$$-\mathrm{i}\omega_n \to \omega + \mathrm{i}\delta, \qquad \delta \ll 1. \qquad (A2)$$

In the mostly-plus metric signature, most quadratic terms in the Lagrangian are negative. By going to imaginary time, the kinetic terms are also negative. For

this reason we define the Euclidean Lagrangian as $\mathcal{L}_E = -\mathcal{L}(t = -i\tau)$. We therefore use:

$$Z = \int \mathcal{D}\{\text{fields}\} e^{-S_E/\hbar},$$

$$S_E = \int d\tau \, d^3x \, (-\mathcal{L}) \equiv \int d\tau \, d^3x \, \mathcal{L}_E. \qquad (A3)$$

## Appendix B: Helical coordinates.

Here we present the details of the helical coordinates that are used throughout this work. It is a coordinate system where the basis vector are eigenvectors under 3-rotation *in momentum space.* This mostly follows Kleinert's definitions [5].

### 1. The helical basis

For concreteness, we parametrize the 3-momentum $\mathbf{q}$ by angles $\eta, \zeta$ as

$$\begin{pmatrix} q_x & q_y & q_z \end{pmatrix} = q \begin{pmatrix} \cos\eta & \sin\eta\cos\zeta & \sin\eta\sin\zeta \end{pmatrix}, \qquad (B1)$$

where $q = |\mathbf{q}|$.

The first step is to define a cartesian coordinate system in momentum space, with basis vectors:

$$\hat{e}^L = \begin{pmatrix} \cos\eta & \sin\eta\cos\zeta & \sin\eta\sin\zeta \end{pmatrix}^T,$$
$$\hat{e}^R = \begin{pmatrix} -\sin\eta & \cos\eta\cos\zeta & \cos\eta\sin\zeta \end{pmatrix}^T,$$
$$\hat{e}^S = \begin{pmatrix} 0 & \sin\zeta & -\cos\zeta \end{pmatrix}^T. \qquad (B2)$$

Here L is longitudinal (parallel to momentum) while R and S are transverse (orthogonal to momentum and to each other). These vectors are all real $\hat{e}^E = \hat{e}^{E*}, E = L, R, S$.

The generator of rotations around axis $k$ in 3-space is

$$(S_k)_{mn} = -i\epsilon_{kmn}. \qquad (B3)$$

The *helicity matrix* $H$ is now defined as

$$H_{mn} = q_k (S_k)_{mn}. \qquad (B4)$$

Linear combinations of the vectors Eq. (B2) which are eigenvectors $\hat{e}^{(h)}$ of $H$ with eigenvalue $h = 0, +1, -1$ are

$$\hat{e}^0 = \hat{e}^L, \quad \hat{e}^{+1} = \frac{1}{\sqrt{2}}(\hat{e}^S + i\hat{e}^R), \quad \hat{e}^{-1} = -\frac{1}{\sqrt{2}}(\hat{e}^S - i\hat{e}^R). \qquad (B5)$$

Note that these eigenvectors are related as

$$\hat{e}^{+1*} = -\hat{e}^{-1}, \qquad \hat{e}^{-1*} = -\hat{e}^{+1}. \qquad (B6)$$

These vectors are orthonormal and complete

$$\sum_m \hat{e}_m^{h*} \hat{e}_m^{h'} = \delta_{hh'}, \qquad (B7)$$

$$\sum_h \hat{e}_m^h \hat{e}_n^{h*} = \delta_{mn}. \qquad (B8)$$

We can define the projectors on these helicity eigenvectors

$$P_{mn}^{(h)} = \hat{e}_m^{(h)} \hat{e}_n^{(h)*}, \qquad (B9)$$

which satisfy $(P^{(h)})^2 = P^{(h)}$ and $\sum_h P^{(h)} = 1$.

The helicity matrix $H$ is part of the familiar $SU(2)$ or $SO(3)$-structure, with raising and lowering matrices

$$H^+ = -\sqrt{2}\hat{e}_m^{+1} S_m, \qquad H^- = \sqrt{2}\hat{e}_m^{-1} S_m. \qquad (B10)$$

These satisfy the usual algebra

$$[H^+, H^-] = 2H, \quad [H, H^+] = H^+, \quad [H, H^-] = -H^-, \qquad (B11)$$

These operators act on the basis vectors as

$$H^+ \hat{e}^{+1} = 0, \qquad H^- \hat{e}^{+1} = \sqrt{2}\hat{e}^0, \qquad (B12)$$
$$H^+ \hat{e}^0 = \sqrt{2}\hat{e}^{+1}, \qquad H^- \hat{e}^0 = \sqrt{2}\hat{e}^{-1}, \qquad (B13)$$
$$H^+ \hat{e}^{-1} = \sqrt{2}\hat{e}^0, \qquad H^- \hat{e}^{-1} = 0. \qquad (B14)$$

### 2. Helicity decomposition of vector fields.

We can express a vector field in helicity components. With the projectors (B9), a natural definition of helicity components of a vector field $A_m(\mathbf{q})$ follows from

$$A_m(\mathbf{q}) = \sum_{h=0,+1,-1} P_{mn}^{(h)} A_n = \sum_h \hat{e}_m^{(h)} (\hat{e}_n^{(h)*} A_n) \quad (B15)$$

However, for real-valued fields $A(\mathbf{x})$, the Fourier components must obey $A(-\mathbf{q}) = A^*(\mathbf{q})$ under momentum inversion. We are going to define helicity components which also satisfy this condition [? ]. Under reversal of momentum $\mathbf{q} \to -\mathbf{q}$, the angles $\eta$ and $\zeta$ transforms as

$$\eta \to \pi - \eta, \qquad \zeta \to \pi + \zeta \qquad (B16)$$

implying that,

$$\cos\eta \to -\cos\eta, \qquad \sin\eta \to \sin\eta,$$
$$\cos\zeta \to -\cos\zeta, \qquad \sin\zeta \to -\sin\zeta. \qquad (B17)$$

This leads to the following transformation properties of the unit vectors,

$$
\begin{aligned}
\hat{e}^{\mathrm{L}}(q) &= -\hat{e}^{*\mathrm{L}}(-q), & \hat{e}^{0}(q) &= -\hat{e}^{0*}(-q),\\
\hat{e}^{\mathrm{R}}(q) &= \hat{e}^{\mathrm{R}*}(-q), & \hat{e}^{+1}(q) &= -\hat{e}^{+1*}(-q),\\
\hat{e}^{\mathrm{S}}(q) &= -\hat{e}^{*\mathrm{S}}(-q), & \hat{e}^{-1}(q) &= -\hat{e}^{-1*}(-q).
\end{aligned} \tag{B18}
$$

We will insert a factor of i for the components $\mathrm{L, S}, 0$ of vector fields, to ensure the vector field stays real-valued:

$$
\begin{aligned}
A_m &= \mathrm{i}\hat{e}_m^{\mathrm{L}}A^{\mathrm{L}} + \hat{e}_m^{\mathrm{R}}A^{\mathrm{R}} + \mathrm{i}\hat{e}_m^{\mathrm{S}}A^{\mathrm{S}}\\
&= \mathrm{i}\hat{e}_m^{0}A^{0} + \mathrm{i}\hat{e}_m^{+1}A^{+1} + \mathrm{i}\hat{e}_m^{-1}A^{-1}.
\end{aligned} \tag{B19}
$$

For instance this implies,

$$
\partial_m A_m(x) \rightarrow \mathrm{i}q_m A_m(q) = -q A^{\mathrm{L}}(q). \tag{B20}
$$

The momentum-space and helicity components are then defined by

$$
\begin{aligned}
A^{\mathrm{L}} &= -\mathrm{i}\hat{e}_n^{\mathrm{L}*}A_n, & A^{0} &= -\mathrm{i}\hat{e}_n^{0*}A_n,\\
A^{\mathrm{R}} &= \hat{e}_n^{\mathrm{R}*}A_n, & A^{+1} &= -\mathrm{i}\hat{e}_n^{+1*}A_n,\\
A^{\mathrm{S}} &= -\mathrm{i}\hat{e}_n^{\mathrm{S}*}A_n, & A^{-1} &= -\mathrm{i}\hat{e}_n^{-1*}A_n.
\end{aligned} \tag{B21}
$$

Because the factors of $-\mathrm{i}$ change sign under complex conjugation, while $A_n(\mathbf{q}) = A_n^*(-\mathbf{q})$ since $A_n(x)$ is real-valued, the components on the left-hand side obey this condition as well.

For the curl of a vector field $B_m = \varepsilon_{mnk}\partial_n A_k$ this implies (using $\partial_n \rightarrow \mathrm{i}q_n$):

$$
B^{0} = 0 \qquad, B^{+1} = -q A^{+1}, B^{-1} = q A^{-1}. \tag{B22}
$$

### 3. Helicity decomposition of tensor fields.

For vector fields, we have used the spin-1 operator $S_k$ in the defining representation Eq. (B3). Stress tensors are (0,2)-tensors, so we need a tensor representation. It is given by $S_{ma} = S_m \otimes 1_a + 1_m \otimes S_m$ or explicitly:

$$
(S_k)_{mn,ab} = \delta_{mn}(S_k)_{ab} + (S_k)_{mn}\delta_{ab}. \tag{B23}
$$

The helicity tensor is defined as $H = H \otimes 1 + 1 \otimes H$:

$$
H_{mn,ab} = q_k(S_k)_{mn,ab} = \hat{e}_k^{0}(S_k)_{mn,ab} \tag{B24}
$$

Also the raising and lowering operators are defined as $H_\pm = H_\pm \otimes 1 + 1 \otimes H_\pm$

The basis vectors (or rather, tensors) can be expressed by linear combinations of tensor products of the basis vectors $e_m^{(h)}$ of Eq. (B5):

$$
\hat{e}_{ma}^{(s,h)} = \sum_{h_1+h_2=h} C_{s,h}^{h_1 h_2} \hat{e}_m^{h_1} \hat{e}_a^{h_2} \tag{B25}
$$

Here $C_{s,h}^{h_1 h_2}$ are the Clebsch–Gordan coefficients with quantum numbers $s = 0, 1, 2$ and $h = -s, \dots, s$. For spin-1 $\times$ spin-1 these give (vector indices suppressed):

$$
\hat{e}^{2,2}(q) = \hat{e}^{+1}\hat{e}^{+1} \hspace{4.5em} = \hat{e}^{2,-2*}(q), \tag{B26}
$$

$$
\hat{e}^{2,1}(q) = \frac{1}{\sqrt{2}}(\hat{e}^{+1}\hat{e}^{0} + \hat{e}^{0}\hat{e}^{+1}) \hspace{1em} = -\hat{e}^{2,-1*}(q), \tag{B27}
$$

$$
\hat{e}^{2,0}(q) = \frac{1}{\sqrt{6}}(\hat{e}^{+1}\hat{e}^{-1} + 2\hat{e}^{0}\hat{e}^{0} + \hat{e}^{-1}\hat{e}^{+1}) = \hat{e}^{2,0*}(q), \tag{B28}
$$

$$
\hat{e}^{1,1}(q) = \frac{1}{\sqrt{2}}(\hat{e}^{+1}\hat{e}^{0} - \hat{e}^{0}\hat{e}^{+1}) \hspace{1em} = -\hat{e}^{1,-1*}(q), \tag{B29}
$$

$$
\hat{e}^{1,0}(q) = \frac{1}{\sqrt{2}}(\hat{e}^{+1}\hat{e}^{-1} - \hat{e}^{-1}\hat{e}^{+1}) \hspace{1em} = \hat{e}^{1,0*}(q), \tag{B30}
$$

$$
\hat{e}^{0,0}(q) = \frac{1}{\sqrt{3}}(\hat{e}^{0}\hat{e}^{0} - \hat{e}^{+1}\hat{e}^{-1} - \hat{e}^{-1}\hat{e}^{+1}) \hspace{0.3em} = \hat{e}^{0,0*}(q) \tag{B31}
$$

Or in short $\hat{e}^{s,h} = (-1)^h \hat{e}^{s,-h*}$. The basis tensors with negative $h$-value are found by changing all $+1 \leftrightarrow -1$ on the right-hand side.

These are orthonormal and complete:

$$
\sum_{ma} \hat{e}_{ma}^{s_1 h_1 *}\hat{e}_{ma}^{s_2 h_2} = \delta_{s_1,s_2}\delta_{h_1,h_2}, \tag{B32}
$$

$$
\sum_{s,h} e_{ma}^{s,h}e_{nb}^{s,h*} = \delta_{mn}\delta_{ab}. \tag{B33}
$$

Again we can define projectors

$$
P_{mn,ab}^{s,h} = \hat{e}_{ma}^{s,h}\hat{e}_{nb}^{s,h*} \tag{B34}
$$

satisfying,

$$
P_{mk,ac}^{s,h}P_{kn,cb}^{s',h'} = P_{mn,ab}^{s,h}\delta_{ss'}\delta_{hh'}, \quad \sum_{s,h} P_{mn,ab}^{s,h} = \delta_{mn}\delta_{ab}. \tag{B35}
$$

We also see that $s = 2, 0$ are symmetric tensors, while $s = 1$ is antisymmetric: $e_{ma}^{2,h} = e_{am}^{2,h}$, $e_{ma}^{0,h} = e_{am}^{0,h}$ while $e_{ma}^{1,h} = -e_{am}^{1,h}$.

Under momentum reversal $\mathbf{q} \rightarrow -\mathbf{q}$, the behaviour is inferred from that of the spin-1 vectors on the right-hand side, but since all spin-1 vectors are odd Eq. (B18), the

2-tensors are all even, leading to $\hat{e}^{(s,h)}(q) = \hat{e}^{(s,h)*}(-q)$ for all $s, h$.

A general tensor field can then be decomposed:

$$A_{ma} = \sum_{s,h} P^{s,h}_{mn,ab} A_{nb} = \sum_{s,h} \hat{e}^{s,h}_{ma} A^{s,h}, \qquad \text{(B36)}$$

$$A^{s,h} = \sum_{nb} \hat{e}^{s,h*}_{nb} A_{nb} \qquad \text{(B37)}$$

These components satisfy $A^{s,h}(-\mathbf{q}) = A^{s,h*}(\mathbf{q})$.

## Appendix C: Topological defects of crystals, a primer.

Some of the readers may not be familiar with the basics of the topology associated with crystalline order. This has actually played a key role in the history of the role of topology in physics. The *dislocation* was the first topological defect recognized as such by the dutch theorist Burgers, identifying the Burgers vector as the topological quantum number employing his Burgers circuit. As emphasized repeatedly, the pattern recognition capacities of the human visual system are quite useful in helping us to comprehend these matters. There is not better way to get quickly acquainted to these matters than by inspecting simple cartoon pictures. Much of this elementary introduction will rest on this "method". For completeness, in the last subsection we will present a summary of some of the main results of algebraic topology in this context

We will follow here the same organization as in the main text. First we will focus in on the translational defects: the dislocations in their edge- and screw varieties, as well as their unusual "fracton" kinematics (glide versus climb motions), the main actors in Section V. In the second subsection we will discuss the "proper" rotational defects at the centre of attention in Sections VIII, IX, the disclinations. In fact, these have been explored much less than dislocations in this particular context since these do not occur in normal solids for the confinement reasons discussed at length in the main text.

The third section may be of interest even to the readers who are familiar with the subject: the defect density, playing a key role in Section VII. This revolves around the "topological image" of the fact that finite translations are indistinguishable from rotations (the semi-direct condition). The fundamental condition for a crystal to "absorb" curvature is in the requirement that a finite density (infinite number) of dislocations with equal Burgers vectors should be present. In turn, there are different ways to organize such dislocations: in three space dimensions in the form of a line of singularities (disclination), or in a plane ("open" grain boundary) or even in the form of just a gas of dislocations. The soft-matter community exploring the rigid 2D curved backgrounds appears to be aware but otherwise one does not find reference to this affair in the literature.

Once again, much of it can be easily visualized by resting on a procedure pioneered by Volterra in the 19-th century: by folding, cutting and glueing pieces of solid stuff (like paper) where the only required abstraction is that one has to work with perfect crystal lattices.

### 1. Restoring translations: the dislocations.

This passage is illustrating the various motives that are at work in the Section V in the main text.

The subject of topology in physics was literally born in 1939 by the identification by Burgers of the topological nature of the *dislocation*, resting on the work by Volterra and other mathematicians. It is the topological excitation exclusively associated with *translational* symmetry breaking: as e.g. free vortices are the unique agents associated with the destruction of the superfluids, free dislocations turn a crystal actually in at least a liquid crystal if not in a true liquid. These restore the translation invariance while extra conditions are required for the rotational symmetry breaking, as we will discuss.

The edge dislocation is readily visualized. Insert an extra plane of atoms in a crystal that ends somewhere. The end of this plane forms a line in the three dimensional crystal, analogous to the vortex line. Take a plane perpendicular to this line and draw a loop of arbitrary size around the dislocation (the *Burgers circuit*) and compute

$$\oint \mathrm{d}u_a = n_a. \qquad \text{(C1)}$$

As for the vortex, the displacement field has become multivalued while the associated topological invariant is now a spatial vector $\vec{n}$: the *Burgers vector*. The translational symmetry breaking is destroyed: in geometrical language, upon measuring distances by hopping from atom to atom one finds that one more hop is needed "below" the dislocation (Fig. 4) as compared to "above" the dislocation (the multivaluedness). In the main text, we meet this Burgers loop in Eq.'s ( 81 -84) linking it via Stokes theorem to the local expression for the dislocation current.

As we emphasized in the second Section, the effects of the point group (anisotropy) on linearized elasticity are quantitative and rather inconsequential. One can safely rely on the isotropic theory instead. But this is very different for the dislocations where the point group symmetry is crucial. The Burgers vector takes values in the lattice vectors, in accordance with the discrete point group rotational symmetry of the lattice. The elementary dislocations are characterized by Burgers vectors with a length set by the lattice constants. For instance on a square lattice this is governed by the $C_4$ rotations, and the Burgers vector takes values $\vec{b} = (1,0), (0,1), (-1,0), (0,-1)$ while in e.g. a hexagonal crystal it will take 6 values. As we discussed in Section V, not only the Burger's vectors are oriented along the

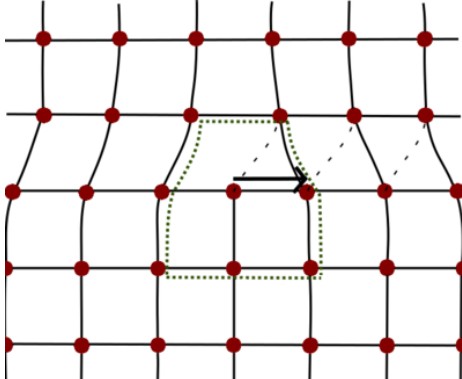

FIG. 4. Illustration of an edge dislocation in a crystal. An extra column of atoms has been inserted in the lower half plane. The Burgers loop, indicated by the dotted green line, needs one extra hop below the dislocation core to close. The resulting Burgers vector points perpendicular to the column of inserted atoms

axis of the crystal, but also the direction of propagation of the dislocation line in the lattice. The static topological source (disclination density) is therefore a symmetric rank 2 tensor in 3D that is invariant under the point group operations, with one index referring to the propagation direction of the dislocation line and the other to the direction of the Burgers vector.

The edge dislocation in 3D (figure 4) is characterized by a Burgers vector oriented perpendicular to the direction of the defect line.But it is also possible to have a Burgers vector oriented in the same direction as the defect line: this is the *screw* dislocation (see Fig. 5), encoded in the diagonal of the dislocation density tensor. This distinction that is already manifest departing from isotropic elasticity (see main text) is general, it also applies when one is paying full tribute to the space group symmetries.

In Section **IV C** we referred to the unusual constraints for the kinematics of dislocations: the "glide" versus "climb" motions, that were only recently recognized to

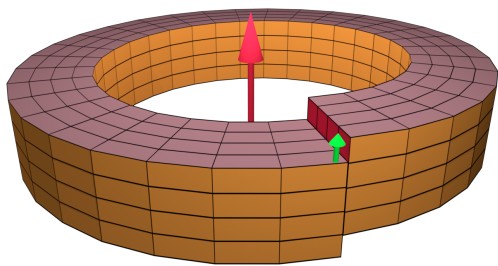

FIG. 5. Illustration of a screw dislocation. When a Burgers loop is taken around the dislocation line (large red arrow), an extra step is needed vertically to close the loop. The resulting Burgers vector (small green arrow) is therefore parallel to the dislocation line.

be examples of the general "fracton" principle. This was however understood early on because it is obvious from the simple cartoons. Dealing with an atomistic solid at temperatures well below the melting temperature the density of "loose atoms" (substitutional/interstitial defects) is exponentially suppressed and very low. Consider now the dislocation line as the termination of an extra plane of atoms inserted in the solid. In order to move in the direction perpendicular to the Burgers vector one needs loose atoms in order to extend the plane but these extra atoms are not available and this motion is impossible (see Fig. 6). This is the "climb motion" that is impeded. On the other hand, in order to move the dislocation in the direction of the Burgers vector one has to just break a bond and reattach it at a nearest-neighbour site: this is the "glide motion" that is allowed.

Resonating with the observations in the main text that the geometry is associated with torsion, it has been long known that dislocations do not fall in the gravity field of the earth. The reason is that they "do not occupy volume" and they do not carry gravitational mass – there is no issue of the kind that spoiled the disclinations, that a big gravitational mass is required to stabilize the geometrical curvature at the core.

Fig. (6) illustrates in a pictorial fashion the principle that when a static external shear stress is applied in a directional parallel to the Burgers vector, the dislocation will accelerate in the glide direction. Apply a force in opposite directions to the upper- and lower surface of the "crystal" in the figure corresponding with a shear stress (this is the effect of the graviton): by re-attaching the dislocation core in the glide direction this shear strain will relax. This is the fundamental principle behind metal working. As it turns out, in a typical metal glued together by the rather un-directional "metallic bond" a dislocation will accelerate in the field of shear force with a characteristic *inertial* mass equal to the mass of the atoms forming the metal. Dealing however with covalent- and ionic solids the glide motion of such dislocations involves a large activation energy with the effect that the dislocations are immobile. Accordingly, these solids are brittle.

### 2. The rotational defects: the disclinations.

Dislocations are the topological agents associated with the restoration of translational symmetry. But crystals also break the isotropy of space down to a discrete point group. Other topological agents are apparently required to restore the rotational invariance. The general topological entity is the defect density that we will discuss in the next subsection. The disclination is a particular realization fulfilling the general defect density requirement, having a special status because it enforces a topological quantization of the rotational structure in the form of the Franck vector topological invariant.

In fact, disclinations attracted not much attention in

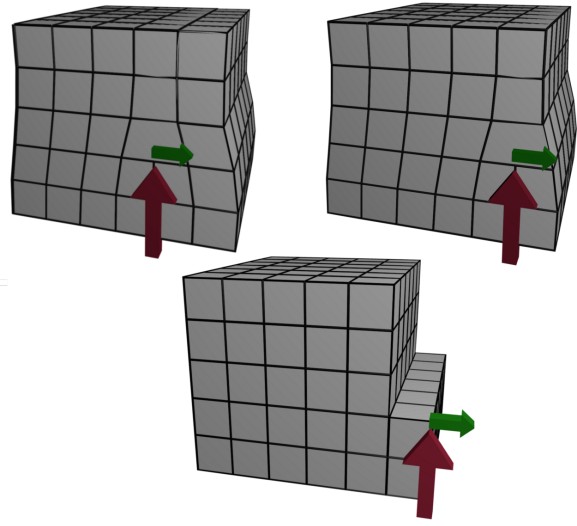

FIG. 6. Glide motion of an edge dislocation under static shear stress. The dislocation will move in the direction of its Burgers vector.

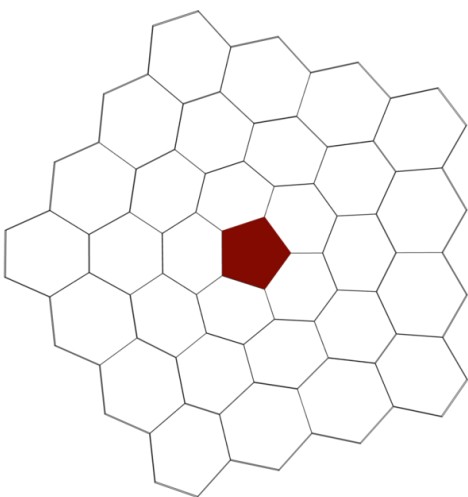

FIG. 7. Wedge disclination in a graphene-like structure. At the centre of the disclination a 5-ring has been inserted. The 6-fold symmetry is hereby becoming globally ambiguous.

the history of the physics of solids for the reason that they are never encountered in free form. As we explained at length in the main text these are *confined* in a flat space, it takes an infinite energy to create them. An exception is found in the context of two dimensional solids like graphene. This refers to the cone, alluded to repeatedly, using the third dimension to avoid the confinement.

The Volterra process is the easy-to-visualize sequence of cutting and welding of the solid, respecting topological requirements. Imagine again the paper cone, constructed by cutting out a wedge and glueing the sheet of paper. But now this sheet is formed from a perfect "chicken wire" (hexagonal) graphene lattice. Imagine taking out

a wedge but in such a way that the chemical damage is avoided as much as possible. From Fig. (7) one infers that by placing a pentagon at the tip of the cone one can actually "cut and glue" the lattice in such a way that the lattice is perfect everywhere else. One can instead add a wedge, terminating it with a 7-ring at the tip: this is the anti-disclination accommodating in fact a hyperbolic curvature which is (as usual) hard to visualize even in two dimensions.

The "Volterra" cutting- and glueing "seam" has become immaterial, but the consequence is that the defect at the tip acquires a quantized rotational value (the pentagon). This is characterized by a deficit angle that precisely matches the missing- or extra $\pi/3$ deficit angle associated with the pentagon or heptagon, respectively. This angle is the topological quantum number characterizing the disclination, called the Franck "scalar" in 2D. It is quantized in units of $2\pi/N$ where $N$ is the periodicity associate with the point group operations: for a $C_6$ axis $N = 6$.

Hence, we observe that the rotational topological defects are point-like in 2D, as the dislocations. Obviously, we also certified that these are uniquely associated with *curvature* in the crystal geometry – see the main text. But we can only visualize them easily using the third dimension: one infers the obvious difficulty drawing it in the plane given the confining strains, as in Fig. (7).

As for the dislocations, 2D is not quite representative for higher dimensions although it is a useful point of departure. We learned that the Burger's vector of the 2D edge dislocations turns into the 3D Burgers vector while the dislocation "point" turns into a 3D line characterized by a vector enumerating the propagation direction, combining in a rank 2 symmetric tensor dislocation density. This works in the same way for disclinations. The Franck vector is aligned with the rotational axis of the disclination. This axis is pointing by default in the third dimension in 2D and therefore the topological invariant is just the Franck scalar. This reflects on the one hand the Abelian nature of point groups in 2D, as well as the fact that there is only scalar curvature in this dimension (Section VI D).

In 3D this turns into a Franck *vector* taking values set by the crystal point group symmetry. In addition, the disclination will turn into a line propagating along lattice directions in 3D. Hence, the topological disclination density will have the same structure as the dislocation density: it is a rank 2 symmetric tensor that is stitched together from the vector indicating the propagation direction ("sense") of the disclination line and the Frank vector, which will both transform under the point group symmetries of the crystal.

The simplest way to construct a 3D disclination is to just "pull out" the 2D disclination in the third direction: this is the analogue of the construction of the 3D *edge* dislocation. Take a 3D "cake", cut out the wedge and weld it together again. Given the confinement this is impossible to visualize; in Fig. (8) one attempts to fool

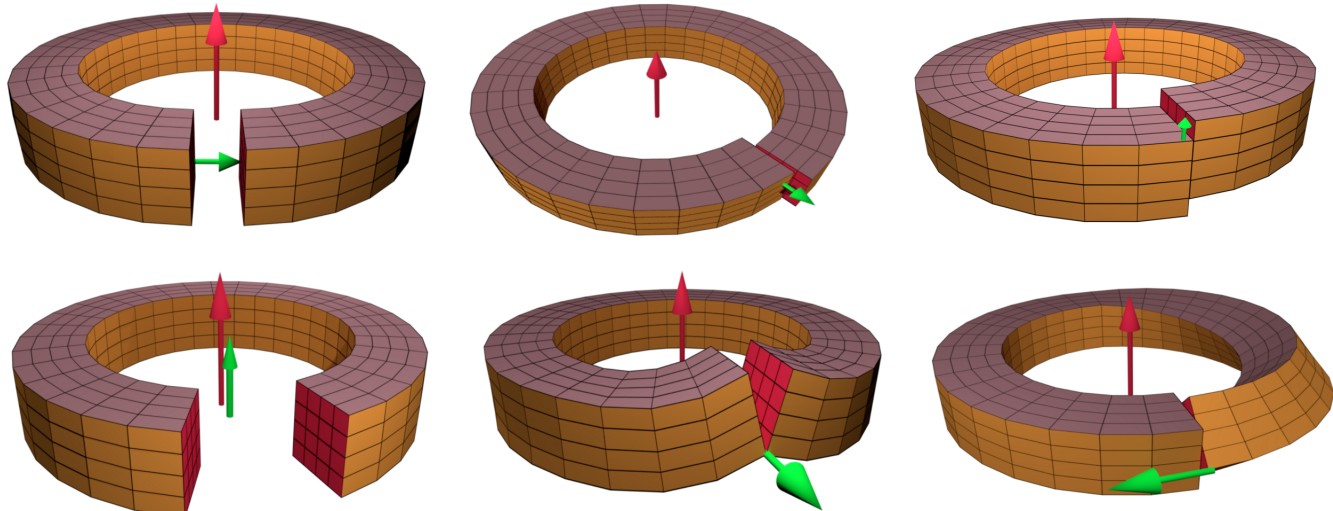

FIG. 8. An overview of the different types of Dislocations and disclinations by Volterra construction. On the top row are the dislocations: the two types of edge dislocations, where the Burgers vector (in green) is perpendicular to the dislocation line (red), and the screw dislocation, where the dislocation line and Burgers vector are parallel. On the bottom row are the disclinations: the wedge disclination, where the Frank vector (in green) is parallel to the dislocation line (in red), and the two twist disclinations where the Frank vector is perpendicular to the dislocation line. After Ref. [94].

the visual system by starting from a finite size ring: the bottom left image corresponds with this "wedge disclination", characterized by a Franck vector being parallel to the propagation direction. These are at the focus of attention in Section VIII.

As for the dislocations one can decompose the topological currents into components where the topological vectors and propagation vectors are either aligned or orthogonal – the edge and screw dislocation affair. We just constructed the case that Franck- and propagation vector are parallel – the wedge disclinations – but we are left with two transversal orientations. These are the *twist* disclinations highlighted in Section IX: their Volterra constructions are shown as lower middle- and right panel in Fig. (8), actually representing quite a challenge to decode for our visual system! As we highlight in Section IX, these are associated with curvature encoded in the off-diagonal elements of the Einstein tensor in the preferred rotational frame of the crystal.

### 3. The "semidirect" relations, rotational topology and the defect density.

As we showed in Section VI the topological quantity that is associated with rotations and curvature is the defect density. In Section VI B we discuss how this can be decomposed in a disclination density and "everything else" called the contortion. But what is the meaning of this "everything else"? The defect density is a topological quantity of an unusual kind, rooted in the semi-direct relation between translations and rotations. Given that this is not standard material we discussed it already at

length in the main text (see also Section VII) and here we will illustrate it with the Volterra cartoons.

The semidirect nature of the Poincare- as well as the space groups have the unusual consequence that the translational- and rotational topological defect structures are interrelated in a building block ("lego-brick") manner. Let us again consider the 2D wedge anti-disclination obtained by inserting a wedge of material characterized by the "Franck" opening angle. But this may as well be viewed as a stack of lines of atoms coming to an end, organized as indicated in Fig. (9). The disclination can be viewed as a bound state of an infinite number of dislocations! This is just the topological expression of the fact that an infinite number of infinitesimal translations are required for a finite rotation.

One observes however that the Burger's vectors of the stack of dislocations are everywhere pointing in the same direction – this already gives a first clue of how to "construct" rotational topological structure from the translational defects. For instance, one can contemplate how to construct a state of matter that breaks space rotations while translational invariance is restored resting on this topological language. This refers to the nematic type liquid crystals. Departing from the crystal a proliferation of dislocations is required to restore the translational invariance. But how to maintain the broken rotational symmetry of the crystal in this "dislocation condensate"? The precise answer is "Burgers neutrality": locally Burgers vectors occur in precisely anti-parallel configurations [24, 25]. Given this condition the rotational order is not affected, and the disclinations deconfine becoming the observable (in a flat space) topological defects of the nematic state.

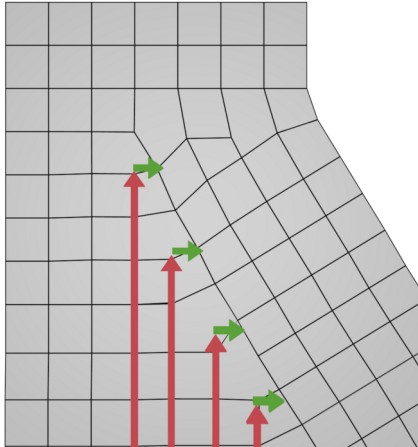

FIG. 9. A disclination can be thought of as an infinite stack of dislocations, every next one translated in the direction of their Burgers vectors.

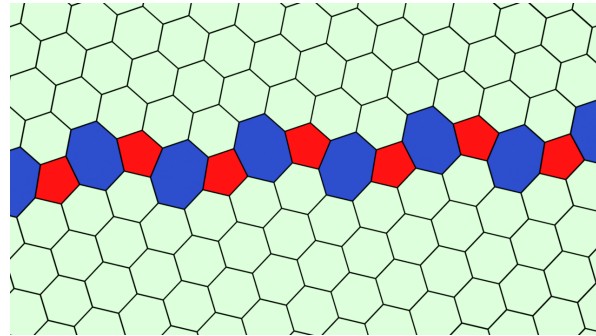

FIG. 10. Two misaligned 2D graphene structures come together in a grain boundary. Bound pairs of 5-rings (red) and 7-rings (blue) form the dislocations, which are stacked together to create a "seam" between the two structures.

But it also works the other way around. A next rule is that a *dislocation* can always be viewed as a bound *disclination-antidisclination pair* separated by one lattice constant (the magnitude of the Burger's vector). A typical example is the dislocation as realized in graphene consisting of a 5- and 7 ring glued together – the grain boundary that will come next is just a stack of such dislocations, see Fig. (10). In the top row of Fig. (8) it is illustrated how the edge- and screw dislocations can be constructed by a Volterra constructions, by taking out a "disclination wedge" directly reinserting it after a shift by a lattice constant.

The disclination is special in the regard that by selecting the proper Franck vector the stack of dislocations forming a line in 2D or a plane in 3D line becomes immaterial because the lattice can be matched precisely away from the disclination core. Obviously, the requirement for this seam to become immaterial is the origin of the topological quantization captured by the Franck vector. But we already got a sense that this is not an absolute condition when it comes to destroying the rotational symmetry breaking. Somehow, properly organized configurations of dislocations with equal Burgers vectors should suffice. This is the wisdom encoded in the formula for the defect density. But what is the meaning of "properly organized"?

Here the "grain boundary" enters. In fact, in the reality of materials science grain boundaries are the most prominent defect structures. Anything that is not a single crystal is usually littered with them. The reason for this is simple. From the melt small crystallites start to form independently, growing in size until they meet. At their boundaries their lattice directions will typically not be aligned. Temperatures are typically still close to the melting temperature and there is plenty of opportunity for the atoms at the grain boundaries to anneal in the optimal local configurations capable of accommodating the misfits in lattice orientation.

The outcome is typically that yet again a *stack* of dislocations is formed characterized by equal Burgers vector. This is illustrated in Fig. (10). The pairs of 5-7 rings are the dislocations with the microscopically resolved cores as we just discussed. It is seen that these are all oriented in the same way, and the effect is a rotation of the lattice directions upon traversing this grain boundary. This rotation is called the misfit angle and this is in turn set by the distance between the dislocations in the grain boundary (see Section VII).

The origin of the special stability of such grain boundaries lies in the interactions between the dislocations. Dislocations with equal Burgers vector repel each other through the long range strain fields in the lattice. This is the same as for superfluid vortices with the same sense of rotation. However, it is easy to see that when the Burgers vectors are orientated in an orthogonal direction relative to the plane (in 3D) in which they reside as is the case for a grain boundary these repulsive interactions disappear. One can already infer this from Fig. (10): there are obviously no long range strains on either side of the boundary.

The simplest example is a "twinned crystal" where, say, two single crystals are grown together meeting at a single grain boundary. Obviously some kind of ambiguity with regard to the rotational symmetry is introduced by the grain boundaries. This is the prime cause that "polycrystalline" (or "granular") solids are described by *isotropic* elasticity on the macroscopic scale – grain boundaries have averaged away the point group symmetry. But has this dealings with the rotational defects that represent curvature in the crystal geometry?

The answer is obviously negative. A simple classroom experiments can clarify it. Take a piece of graph paper, cut in two pieces, remove a wedge on one side and glue it back again. One sees the misfit of the "lattice directions" and with a bit of imagination one can infer that by moving the lines on the paper near the seam it can be "annealed" in a stack of dislocations. But this can all be achieved without lifting the paper in the vertical

direction: such a grain boundary represents a flat, non-curved manifold in crystal geometry! One can continue such glueing and pasting until one has a patchwork of misaligned grains and this is how granular solids look like.

Apparently, an extra condition is required for configurations of equal Burgers vector dislocations to represent the defect density. This is yet again very easy to identify with cutting and glueing. In the procedure we just described the grain boundaries form *closed* manifolds – these never end anywhere inside the solid. But let us get back to the construction of the cone, now using graph paper. To obtain the disclination one has actually to cut out precisely a $\pi/2$ wedge (assuming a square "lattice'): one finds that this lattice is perfectly uninterrupted at the gluing surface. But cut out now a smaller wedge and glue it: the glueing surface turns now in a grain boundary!

The simple moral is that one can accomodate any opening angle, paying the prize that a grain boundary emerges at the tip singularity extending to infinity. In 3D this grain boundary corresponds with a stack of dislocations, forming a surface. Instead of having to pay only the core energy of the line-like (codimension 1) disclination one can sacrifice the quantization of curvature by paying the prize associated with a "half" plane of dislocation cores, the codimension 2 grain boundary.

This reveals the principle. In summary, implied by the semi-direct relation between translations and rotations one runs into "lego-brick topology": the translational defects are like lego bricks that have to be stacked to build the rotational defects. A first requirement is that the Burgers vectors have to point in the same direction. The added requirement is however that in order for these to represent curvature the configurations should be topologically deformable in grain boundaries that terminate in the bulk of the solid. To recover the quantization of curvature the grain boundaries become immaterial, only the termination line (disclination) is still identifiable. But at the "chemical" expense associated with forming of a grain boundary seam one can accommodate any degree of curvature.

These are the simple principles that underly the discussion in Section (VII). Dealing with an effective rigid background curvature, as the hard surfaces of soft matter and the gravitating physical space-time one can identify a homogeneous curvature in crystal geometry in the limit that the curvature radius is infinite as compared to the lattice constant. One identifies a grain boundary termination line to every point in space, insisting that the opening angle deficit becomes infinitesimal. This in turn corresponds with the separation of the dislocations in the grain boundary becoming infinite, obtaining a homogeneous gas of equal Burgers vector dislocations being perfectly compatible with a homeogenous background curvature. Surely the prize is that the co-dimension of this "defect" is set by the volume of the crystal.

## 4. The algebraic topology of crystals

Having relied in the above entirely on the Volterra "cut and glue" method let us finally present a short overview of the algebraic topology that is behind this affair. By definition, a defect is a singularity in an otherwise ordered medium, quantified by an order parameter field with long-range order. A topological defect is *topological* because it is invariant under continuous deformations of the order parameter field. The mathematical concept of continuous deformation is *homeomorphism*, and two configurations that differ only by a continuous deformation are *homeomorphic* to each other. One can then classify configurations which are not homeomorphically equivalent. This leads to the *classification of stable topological defects* in terms of *homotopy groups*. The standard reference is the review by Mermin [101] (see also Ref. [63]). Let us focus here entirely on the dislocations and disclinations, the objects that are characterized in the present context by topological quantum numbers being the focus of the homotopy groups.

Crystals are instances of ordered media with spontaneously broken symmetry: the full global symmetry group $G$ of the action is not respected by the medium, which is only invariant under a subgroup $H \subset G$. The order parameter (field) is a function $\phi(x)$ on every point in real space $\mathbb{R}^d$, taking values in *order parameter space* which is in one-to-one correspondence to the *coset space* $G/H$, i.e. the equivalence classes in $G$ where elements that differ by any element in $H$ are equivalent: $\exists h \in H : g_1 = g_2 h \Rightarrow g_1 \sim g_2$. In general $G/H$ is not a group itself.

We consider directed loops in order parameter space: let $\mathcal{C}$ be any closed loop in real space $\mathbb{R}^d$, parametrized by $c \in [0, 1]$ with base point $x^\star = x(0) = x(1)$. We can follow the evolution of the order parameter field $\phi(x)$ as $x$ traverses $\mathcal{C}$. Clearly this is also a closed loop since $\phi(x(0)) = \phi(x(1)) \equiv \phi^\star$.

If we have such a loop $f$ in order parameter space with base point $\phi^\star$, we can consider continuous deformations $f'$ that keep the base point fixed; these deformations can be due to either deforming the base contour $\mathcal{C}$ or by change of the order parameter field $\phi(x)$, but for our purposes this distinction is irrelevant. Given any two loops $f_1$ and $f_2$ with the same base point $\phi^\star$, we say they are *homotopic* or *homotopically equivalent* if $f_1$ can be continuously deformed into $f_2$.

In particular, some loop may be contracted onto the single point $\phi^\star$ (this is the 'constant loop' $\phi(x(c)) = \phi^\star \ \forall c$). This corresponds to a uniform order parameter, so a state without any defects. Conversely, a defect is a singularity in the order parameter field, and a loop around the singularity cannot be contracted to a point. Away from the singularity, the order parameter field can still be deformed, and there are many configurations which correspond homotopically to the same defect.

The loops form a group. Clearly, we can consecutively

transverse one loop and then another, and this total curve constitutes a loop itself, beginning and ending at the base point $\phi^\star$. This concatenation of loops is the group product. The constant loop mentioned above is the group unit, and transversing a loop in the opposite direction is the group inverse. Since we are only interested in homotopically inequivalent loops, we look at equivalence classes of loops, which form a group with the same product. This group is called the *fundamental group* or the *first homotopy group* $\pi_1(G/H)$ of the topological space $G/H$. In this way, the possible inequivalent defects are classified by group elements of $\pi_1(G/H)$. It turns out that the fundamental groups related to different base points $\phi^\star$ are isomorphic, so in the end we can forget about the base point [101] (but see below).

To calculate the group $\pi_1(G/H)$, one uses an important result from homotopy theory [101]:

$$\pi_1(G/H) \simeq \pi_0(H). \qquad (C2)$$

The fundamental group of $G/H$ is isomorphic to the *zeroth homotopy group* of the residual symmetry group $H$, which are simply the disconnected components of $H$. In crystals, the residual symmetry is entirely discrete, and the disconnected components of a discrete group are just the group elements themselves. Therefore we have $\pi_1(G/H) \simeq H$ (for discrete groups $H$). One caveat is that Eq. (C2) only holds if $G$ is simply connected; otherwise, we must look at the universal covering group $\bar{G}$ of $G$, which is a simply connected group $\bar{G}$ with a surjective map $\bar{G} \to G$. Via the same map we can define the *lift* of $H$ to $\bar{H}$. One can show that in fact $\pi_1(G/H) \simeq \pi_1(\bar{G}/\bar{H})$, so that we can extend Eq. (C2) to:

$$\pi_1(G/H) \simeq \pi_1(\bar{G}/\bar{H}) \simeq \pi_0(\bar{H}) \simeq \bar{H}. \qquad (C3)$$

The fundamental group can be either Abelian or non-Abelian. If it is Abelian, defects are unambiguously classified by group elements of $\pi_1(G/H)$. However this is not the case if it is not Abelian. Consider a single defect with topological charge $a \in \pi_1(G/H)$. If we introduce another defect with charge $b$ at some distant point, the loop around $a$ can be continuously deformed to a loop that encircles first $b$, then $a$, and then $b$ in the opposite direction. But in a non-Abelian group $a \neq bab^{-1}$. Therefore, the topological invariant for defects with non-Abelian fundamental groups are the *conjugacy classes* instead of the group elements [101].

The fundamental group $\pi_1$ classifies topological defects of codimension 2: points in the 2-dimensional plane, or lines in 3-dimensional space. Other homotopy groups classify topological defect of different dimensionality: for instance $\pi_0(G/H)$ labels defects of codimension 1 (domain walls), while $\pi_2(G/H)$ labels defects of codimension 3 (point defects in 3-dimensional space). In our case, only $\pi_1(G/H)$ is non-trivial. This is the reason we have only looked at line defects in 3-space throughout this work: they are the only interesting ones.

### a. Dislocations.

Dislocations are translational defects, so we focus only on the translational symmetry $G = \mathbb{R}^d$. In a crystal this is broken down to the translational symmetries of the Bravais lattice, isomorphic to $H = \mathbb{Z}^d$. The order parameter space is $\mathbb{R}^d/\mathbb{Z}^d$, the possible positions of the origin of the unit cell in the background space, modulo lattice translations. Since $\mathbb{R}^d$ is simply connected, the fundamental group following from Eq. (C2) is,

$$\pi_1(\mathbb{R}^d/\mathbb{Z}^d) \simeq \pi_0(\mathbb{Z}^d) \simeq \mathbb{Z}^d. \qquad (C4)$$

The inequivalent dislocations are therefore labelled by elements of the lattice itself. Unsurprisingly, this is what we had already found above: the topological invariant is the Burgers vector, which takes values in the lattice vectors. This group is Abelian.

### b. Disclinations.

Disclinations are rotational defects. The rotational group is $O(3)$, which includes reflections. Since the residual symmetry group also contains reflections in cases of our interests, there are no reflection topological defects. We can therefore simplify the discussion by looking only at proper rotations $G = SO(3)$. The proper point group of the crystal is a discrete subgroup of $SO(3)$. The possible subgroups are: the cyclic groups of order $n$, $C_n$; the dihedral groups of order $2n$ $D_n$; the tetrahedral group $T$, the octahedral group $T$ and the icosahedral group $I$. However, due to restrictions posed by the broken translational symmetry, only $n = 1, 2, 3, 4, 6$-fold rotations are permitted, and icosahedral point group does not occur either. Note that the dihedral groups cover several possilities of reflection planes arising from non-commutative rotations in $SO(3)$.

The group $SO(3)$ is not simply connected, and its covering group is $\bar{G} = SU(2)$. The point group is lifted to $\bar{H} = \bar{P}$. So the complete classification of stable disclinations in crystals with point group $P$ is given by Eq. (C3) as:

$$\pi_1(SO(3)/P) \simeq \pi_1(SU(2)/\bar{P}) \simeq \pi_0(\bar{P}) \simeq \bar{P}. \qquad (C5)$$

However, as is well known, the lift of $SO(3)$ to $SU(2)$ is relevant to fermions, while ordinary disclinations are 'bosonic' in this respect. Furthermore, while non-Abelian discrete subgroups of $SO(3)$ such as $D_n, n > 2$ exist, defects classified by such groups do not appear as disclinations. In practice, we are almost exclusively interested in rotational disclinations associated with the $C_2$, $C_3$, $C_4$ or $C_6$ cyclic subgroups, corresponding to Frank vectors of $180°$, $130°$, $90°$ or $60°$ respectively.

### c. Dislocations and disclinations combined.

The point group $\bar{P}$ can be non-Abelian, but as mentioned above, these do not feature as disclinations in ordinary crystals. A much more profound complication arises due to the semidirect product structure between translations and rotations in the Euclidean group $E$. The notation $E = \mathbb{R}^d \rtimes O(d)$ means that rotations *act on* the translations. Let $(\mathbf{a}, R)$ be a group element of $E$ with a translation (vector) $\mathbf{a}$ and a rotation (matrix) $R$. Since $E$ is a group, we are supposed to be able to take the group product: two consecutively transformations constitute another transformation. But it is easily seen that translations and rotations do not commute. Instead, the group product is [101]:

$$(\mathbf{a}_1, R_1) \circ (\mathbf{a}_2, R_2) = (\mathbf{a}_1 + R_1 \mathbf{a}_2, R_1 R_2). \qquad (C6)$$

This is to be compared with the group product if we would have a direct product $\mathbb{R}^d \times O(d)$ and rotations and translations independent: $(\mathbf{a}_1 + \mathbf{a}_2, R_1 R_2)$. In Eq. (C6) the first translation $\mathbf{a}_2$ is rotated by the second rotation $R_1$. Similarly, the inverse of $(\mathbf{a}, R)$ is $(-R^{-1}\mathbf{a}, R^{-1})$ instead of $(-\mathbf{a}, R^{-1})$.

This group product carries over to the topological defects. A general topological defect in a crystal is denoted by $(B^j, \Omega^{kl})$, with $B^j$ the Burgers vector and $\Omega^{kl}$ the rotation in the plane spanned by $k$ and $l$ associated with Frank vector $\Omega^n = \epsilon_{nkl}\Omega^{kl}$. The product rule for these defects is the same as Eq. (C6) [101].

This has several important consequences, which have been touched upon before:

- The Burgers vector is not topologically invariant in the presence of a defect with non-zero $\Omega$. Suppose we have a disclination–anti-disclination pair that together is rotationally neutral. Splitting this pair, letting each part pass on opposite sides of a dislocation $(\mathbf{B}, 0)$, and bringing them together again corresponds to

$$(0, \Omega) \circ (\mathbf{B}, 0) \circ (0, \Omega^{-1}) = (\Omega\mathbf{B}, 0) \neq (\mathbf{B}, 0). \qquad (C7)$$

  The Burgers vector is rotated.

- The true topological invariants are conjugacy classes. A dislocation with Burgers vector $\mathbf{B}$ is equivalent to any Burgers vector $n\Omega\mathbf{B}$ with $n \in \mathbb{Z}/0$ and $\Omega$ an elementary Frank rotation via Eq. (C7). Similarly, a disclination $(\mathbf{0}, \Omega)$ is in the same conjugacy class as any $(\mathbf{B} - \Omega\mathbf{B}, \Omega)$ with $\mathbf{B}$ any allowed Burgers vector [101]. Note, however, that the rotational part (the second entry in the pair $(\cdot, \Omega)$), is always the same. Therefore, the rotational—and therefore curvature—character of any disclination is topologically invariant.

- A disclination–anti-disclination pair is not topologically neutral, but has a Burgers vector topological charge, where the Burgers vector is determined by the separation between the pair. This a way to see that disclinations are confined in crystals on purely topological grounds.

- In the same vein, a semi-infinite stack of dislocations with parallel Burgers vector can also be viewed as a disclination.

- Conversely, a finite amount of Burgers vector may correspond to a finite density of disclinations and anti-disclinations. Notice however that in this classic classification scheme the part of the defect density that is not topologically quantized (the disclinations) are just ignored. There is no mention of grain boundaries and their status in this algebraic topology setting – they are juts not part of this scheme.

### d. Issues associated with broken translation symmetry.

The breaking of translation symmetry poses a problem for the standard treatment of homotopy theory, since this rests entirely on transformations in continuous, not discrete, space. However, we can still consider smooth deformations on the background space, while order parameter space is discrete. In other words, we should consider the broken translational symmetry as a property of the medium $\phi(x)$, and then it can be treated in the usual manner. The question what would happen if space itself were discrete has to our knowledge not received much attention. Under the present thesis, that space must conform itself to the broken spatial symmetry of the medium, this question becomes quite relevant, but we leave it for future research. Meanwhile, it is known that there are some subtleties in the standard homotopy theory of topological defects if translational symmetry is broken in the medium. For instance, some defect configurations are not independent of the base point $\phi^\star$[102, 103]. On the other hand, the Frank vector seems to be a topological invariant; as far as the curvature (not torsion) properties are concerned, one can rely on standard homotopy theory.

### ACKNOWLEDGMENTS

This work has profited from many insightful discussions. In the first place of course Hagen Kleinert: his resilience in pursuing his "multivalued fields" agenda is exemplary. It has been a pleasure for us to find out how his grand scheme came particularly to fruition in crystal gravity. David Nelson and in particular also Vincenzo

Vitelli have played a key role in raising our awareness of the solids on rigid curved surfaces. Vitelli deserves special mention for explaining to us in an early state of this research the "sphere turning into a cube" ramification of the Higgsing. In a later state, we learned much from Luca Giomi regarding the mechanism of frustration in this context. On the gravity side, we have profited repeatedly from insights of Koenraad Schalm. A special thank is for Robert Jan Slager and David Rodrigues for their advice and insights as related to the mathematical tradition dealing with three dimensional topology,

and Amit Acharya for sharing his insights regarding the state of the art in material science plasticity. We are also grateful to Blaise Gouteraux and Simon Ross for their thorough proofreading of the manuscript. FB and JZ acknowledge financial support by the Netherlands Organization for Scientific Research (NWO/OCW) and AJB by the Ministry of Education, Culture, Sports, Science and Technology (MEXT)-Supported Program for the Strategic Research Foundation at Private Universities "Topological Science" (Grant No. S1511006) and by Japan Society for the Promotion of Science (JSPS) Grant-in-Aid for Early-Career Scientists (Grant No. 18K13502).

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
