# Peer review of "Crystal gravity"

_SciPost Physics_

## Round 1 · Referee Report · Anonymous (Referee 1) · 2022-1-24

Report

This is an extremely interesting piece of work that deals with a question of (well, to me) tremendous physical interest. We are very familiar with the fact that when a theory with (say) a U(1) spontaneously broken global symmetry is gauged, the putative Goldstone modes are eaten by the gauge field, and the resulting spectrum is massive, with the phenomenology associated superconductors (expulsion of magnetic flux, quantized vortices, etc.). What then happens if one considers a solid (which spontaneously breaks translational symmetry) in the real world (where — loosely speaking — translational symmetry is gauged by gravity)? Is the graviton gapped? What is a gravitational superconductor? Etc.

This work aims to answer that question in detail: for the most part it does so, explaining in detail the phenomenology of a gravitational theory coupled to a giant space-filling crystal. It does so from a rather sophisticated and effective field-theory point of view. The topic is rich and rather technical due to the proliferation of polarizations; analogies to the U(1) case help a lot here, but there is a lot going on. There are many great nuggets of wisdom that link ideas in “real-life” physics (e.g. associated with things like principles of Weber detector design) to formal considerations of symmetry breaking and effective theory, and there are quite a few surprises (e.g. that it is torsion that is expelled, and that the phonons remain massless though the graviton acquires a mass).

Overall, after a few readings I think I have been able to appreciate the key points and I believe that the paper is definitely worthy of publication; it addresses a very natural physical question, and it does so in a very creative and interesting way.

However, I did find the work rather tough going, and I would like to make a few suggestions to make the paper easier to read, and then another comment on a point that the authors did not (I believe) address. 1. The authors frequently compare the gravitational case to the U(1) superconductor; however this analogy is stretched out over several chapters and it can become difficult to maintain in memory how it works simply due to all the moving parts. I suggest that the authors create a table or dictionary relating all the key ingredients (e.g. a_{\mu} maps to h_{\mu\nu}, the torsion maps to the magnetic flux, etc.), together perhaps with a page number for each entry showing where the explanation is given. 2. The paper is not consistent about raising and lowering indices; there are also many different kinds of indices (m, a, i, j, \mu \nu) and I don’t think it is clear how they relate to each other, and which of them should be raised and lowered and which not. (I am not sure this was every fully explained in the paper). For a paper dealing with gravity in a fundamental way this is not good. I believe it’s mostly okay because gravity is often (but not always) linearized, but I think this should be made explicit, perhaps by consistently using the flat-space metric to raise and lower them. (This will also make the paper more accessible to a gravitational audience). 3. Finally, a physics question that should be addressed: the authors frequently state that it is unlikely that this will be phenomenologically relevant simply because space is not full of a giant crystal (though they do make interesting statements about dark matter etc). However could it be relevant even in principle? I am a bit confused because if I fill space with a giant crystal then once the spatial extent R of the crystal is bigger than its own Schwarzschild radius it will collapse into a black hole (unless the situation is intrinsically time-dependent like in cosmology). If it has energy density \rho then this will happen when G R^{3} \rho = R, i.e. when R = \sqrt{1/(G \rho)}, i.e. parametrically the same as the gravitational screening length (1) (I imagine the shear modulus is controlled by the same physics as the energy density). Thus there doesn’t seem to be a way to separate these scales, and there is a worry that the physics here can never be accessed, even in principle. In my eyes this does not invalidate the results, but I think it should be addressed cleanly, and I would be extremely interested in knowing if the above conclusion could be evaded. (In practice there is probably a time-dependent version of this theory that could be relevant for cosmology towards which this construction is a first step).

Overall, I am happy to recommend this for publication; however I think the authors should consider the points above to strengthen the paper.

---

## Round 1 · Referee Report · Anonymous (Referee 2) · 2022-1-25

Report

This paper is an interesting read: it is well written, comprehensive, and overall worth publishing. A significant amount of it brings together and reviews results that are not necessarily new, but fit nicely together and are meant to provide the reader with the necessary background. However, despite the fairly extensive bibliography, the authors miss several opportunities to acknowledge prior work and place their own in the appropriate context. In light of sentences such as "We address a subject that could have been analyzed a century ago", the reader might be left with the mistaken impression that this paper introduces an entirely novel viewpoint on the subject. I recommend that the authors consider the following points and comment on the prior literature when appropriate:

  1. First, the Higgsing gf gravity has been considered extensively in the high energy physics literature. See the following paper and all the references therein for a sample:

  2. Bonifacio, James, Kurt Hinterbichler, and Rachel A. Rosen. "Constraints on a gravitational Higgs mechanism." Physical Review D 100.8 (2019): 084017.

  3. The Higgsing gf gravity is also tightly related to massive gravity, a topic that has been studied very extensively. Of particular relevance to this paper are Lorentz-violating theories of massive gravity, which have been first studied systematically in:

  4. Dubovsky, Sergei L. "Phases of massive gravity." Journal of High Energy Physics 2004.10 (2004): 076.

  5. The author's skepticism that this topic could be "of relevance to the physics of our universe" is perhaps a bit premature. For instance, interesting models of inflation drive by solids have been proposed, see:

  6. Gruzinov, Andrei. "Elastic inflation." Physical Review D 70.6 (2004): 063518.

  7. Endlich, Solomon, Alberto Nicolis, and Junpu Wang. "Solid inflation." Journal of Cosmology and Astroparticle Physics 2013.10 (2013): 011.
  8. Kang, Jonghee, and Alberto Nicolis. "Platonic solids back in the sky: Icosahedral inflation." Journal of Cosmology and Astroparticle Physics 2016.03 (2016): 050.
  9. Nicolis, Alberto, and Guanhao Sun. "Scalar-tensor mixing from icosahedral inflation." Journal of Cosmology and Astroparticle Physics 2021.04 (2021): 074.

The last two paper seem also relevant for the discussion of platonic solids in Sec. VIII.

  1. Although holography is not the main focus of this paper, it is mentioned in the introduction as one of the motivations for this study. It is worth pointing out that holographic solid states have been found to be related to monopoles in AdS, see e.g.

  2. Bolognesi, Stefano, and David Tong. "Monopoles and holography." Journal of High Energy Physics 2011.1 (2011): 1-28.

  3. Esposito, Angelo, et al. "Conformal solids and holography." Journal of High Energy Physics 2017.12 (2017): 1-29.

  4. I find the discussion in the first column of p.33 about the symmetries broken by a solid confusing. The statement that "finite translations are the same as rotations" is at best very misleading, at worst plain wrong. Furthermore, the reason why solids don't feature Goldstone modes associated with the breaking of rotation (or boosts, for that matter) is well understood as is a manifestation of the inverse Higgs mechanism:

  5. Ivanov, Evgeny Alexeevich, and Victor Isaakovich Ogievetskii. "Inverse Higgs effect in nonlinear realizations." Theoretical and Mathematical Physics 25.2 (1975): 1050-1059.

  6. Low, Ian, and Aneesh V. Manohar. "Spontaneously broken spacetime symmetries and Goldstone’s theorem." Physical review letters 88.10 (2002): 101602.
  7. Nicolis, Alberto, Riccardo Penco, and Rachel A. Rosen. "Relativistic fluids, superfluids, solids, and supersolids from a coset construction." Physical Review D 89.4 (2014): 045002.

  8. The dual description of superfluids discussed in Sec. III has been studied extensively. For instance, see:

  9. F. Lund and T. Regge, Unified Approach to Strings and Vortices with Soliton Solutions, Phys. Rev. D 14 (1976) 1524

  10. A. Zee, Vortex strings and the antisymmetric gauge potential, Nucl. Phys. B 421 (1994) 111
  11. Horn, Bart, Alberto Nicolis, and Riccardo Penco. "Effective string theory for vortex lines in fluids and superfluids." Journal of High Energy Physics 2015.10 (2015): 1-58.

  12. A good portion of this paper is about defects in gravity, which was a topic discussed for instance in

  13. M.O Katanaev, I.V Volovich, Theory of defects in solids and three-dimensional gravity, Annals of Physics, Volume 216, Issue 1, 1992, Pages 1-28

  14. Bennett, D. L., et al. "The relation between the model of a crystal with defects and Plebanski's theory of gravity." International Journal of Modern Physics A 28.13 (2013): 1350044.

It would be important the the authors comment on previous work on defects in gravity and how it relates to their own work.

Finally, I should emphasize that I am listing specific papers only to provide the authors with an entry point to the relevant literature. This is emphatically not a request for specific citations, although the authors may choose to cite some of the papers above if deemed relevant. Instead, it is a broad encouragement to strengthen their connections with the already existing literature.

---

## Round 1 · Referee Report · Anonymous (Referee 3) · 2022-3-7

Report

This paper studies the coupling between (dynamical) gravity and elastic matter. This is an old subject. What is new to some extent in the context of this paper is that the coupling is done at the level of an action and, additionally, care is given to the possibility of defects in elastic mater. This is an interesting addition to this literature. However, I have various comments that I believe should be addressed or given some consideration before this paper is considered for publication.

Structural: - The paper is very long with a very long introduction and in many ways akin to a review. This long introduction does not help significantly to understand the new contributions of this paper. This would have been a better paper if it had been condensed into 20 pages with the new contributions. Can the authors attempt at making clear a distinction between what is new in this paper and what is not?

Literature: There is a wide variety of literature missing which leads me to disagree with various statements.

  • I am in disagreement with the comments in section IA and in the abstract suggesting that this is the first time that such study has led to a proper understanding of the coupling of elastic degrees of freedom and gravity. Relativistic elasticity is typically used to study neutron star crusts, a literature that seems to have been completely ignored in this paper (e.g. arXiv:2003.05449 and referees therein). In this context, the coupling is usually done at the level of the equations of motion: elastic matter is coupled to Einstein’s equations via an appropriate elastic stress tensor.

  • The authors motivate their work in IB, page 3, as “aimed at informing this community regarding the role of elasticity in this context”, referring to the holographic study of elasticity and in particular references [11,12] and progress “hindered by unfamiliarity with elasticity”. Additionally, when referring to relativistic elasticity the authors cite the work of Carter and Quintana. I think that the authors missed crucial references in the context of modern treatments of relativistic elasticity, also in relation to the holographic community. For instance, the work of Fukuma et al. (arXiv:1104.1416 and references therein) and in particular the work of Armas et al. (arXiv:1908.01175, in particular section 2 and appendix A), which provide a more rigorous geometric formulation of relativistic elasticity than any other literature the authors refer to.

  • The authors do not mention the role elasticity has played in the realm of quantum matter and describing pinned crystals and charge density waves. For instance arXiv:cond-mat/0103392 and the more modern treatments such as arXiv:1702.05104 and arXiv:2001.07357.

  • This paper is rather close in spirit to a recent paper trying to address more rigorously how matter couples to dynamical gravity. The paper arXiv:1907.04976 is a recent exploration that discusses coupling to fluid degrees of freedom, instead of elastic ones. The overlap between the two works, however, would be visible when discussing a more rigorous approach to this work in which dynamics is taken into account.

Could the authors perhaps revise some of their comments and take into account this relevant literature?

Technical details:

  • The authors claim to address mostly the stationary case but fluctuations (gravitons/phonons) are considered. Could the authors make it more clear when the stationary assumption is being used and when it is not? In relation to this, would it be possible to write the full action that is being considered somewhere, that is the elastic part with kinetic terms and the Einstein-Hilbert action?

  • The “working horse of crystal gravity” as referred to equation (10) has already been written down, in particular in section 2.2 of arXiv:1908.01175. Eq. (10) corresponds to the linearised version in the Goldstones and background metric of (2.14) of arXiv:1908.01175. Perhaps the authors could refer to this? The difference is that the authors will supplement this action with the Einstein-Hilbert action.

  • Related to the above comment, the authors could have used a more geometric language to describe the crystal. In particular \mathcal{W}{ma} introduced in equation (10) is just the strain tensor expanded linearly in the Goldstones and background metric. This strain coincides with the linear version of (2.4) of arXiv:1908.01175. It would be useful to comment on what the meaning of \mathcal{W} is, as at the moment it appears as if it was an educated guess.

  • The language employed throughout the paper is rather non-covariant, Eq. (10) being an example with multiple non-contracted indices, which is a bit at odds with the relativistic nature of Einstein general relativity. More consideration could have been given to carefully describe the crystal space and the background spacetime, and in particular mention what is the geometric structures associated with both. How are the indices m and a raised/lowered in Eq.(10)?

  • The Einstein equations coupled to elastic matter are never written down explicitly, except in very special circumstances in which the action is taken on-shell for specific configurations. This is a bit strange for any relativist. Could the authors write down the Einstein equations and equations of motion for the crystal in generality with the stress tensor for elastic matter that they are considering? This would surely make the connection with earlier literature more clear.

  • Extending this work to dynamical settings also requires a non-linear approach to elasticity. I believe that the language in this paper would have to substantially change in order to describe such situations. Could the authors comment on this and what previous works they expect to be useful in order to do so?

I would recommend the paper for publication once the authors review these comments.

---

## Round 2 · Author Response

Dear Editor,

Being engaged in editorial activities, I am well aware of the tedium associated with processing long manuscripts with a content that falls in between the various subdisciplines. But we are grateful for the work and thought you invested in finding referee’s in your network responding with reports that turned out to be quite helpful.

Eventually, this revolves around the communication over the barriers between the various sub-disciplines. Frankly, when writing the first version the precise nature of the barrier was not sharply on our own radar. In the mean-time this has clarified. While waiting for the reports we tested this affair to the primary “customer base” -- the relativists. This clarified that what we call in the paper the “topologization of the gauge curvature” as primary motive of the Higgs phase is highly unfamiliar in this community: the affair of the Abrikosov flux lattices that became the dominant theme in the study of superconductivity in the condensed matter community. This never played any role in GR, but it is our experience that upon getting it across it is received as an intriguing eye opener.

This is also reflected in the referee reports. The substance of the paper is in the sections V-IX – the first 4 sections are setting the stage -- and you surely noticed that there is nothing found in any of the reports alluding to this. The reports of referee’s 2 and 3 reflect their struggle in trying to position it in the terrain that is familiar to them but the issue is that it is just something else.

Our formulation of the introduction is to be blamed, being in this regard not explicit enough. In the revised manuscript we have reformulated these passages putting the emphasis now much more on the “topologization” motive. We have rewritten Section I.A in this spirit, and added a new section I.B concisely summarizing the state of the art of the research on “solids in gravity” and how this relates to what we are looking at. This is intended to spell out what our paper is not about and here we are much helped by the reports. The literature on “solid cosmology” (referee 2), neutron star crusts etcetera and holographic applications (referee 3) is quite extensive. We are grateful for their listing of recent literature that we employ in this section.

Sections 1C-1G are mildly edited to adjust to this modified narrative and we did not touch section 1H, the executive summary of the results in our paper.

With regard to the substance, we were delighted by the observation of referee 3 that in the mean time the “working horse” Eq. (10) has entered the holography inspired literature: arXiv:1908.01175. We knew about it already some ten years ago and the original motivation of our work was holography related, aiming at correcting wrong tensor structure in the bulk as deduced from homogeneous ploys (e.g., Q-lattices). But in the process we found out that it is just way more fun to aim at the more general topologization affair. We have put arXiv:1908.01175 now in the limelight, both in the new general introduction, as well as in the introduction of the relevant section IIA. In one regard we actually disagree with this referee: \mathcal{W}__{ma} is not at all intuitive, it captures the bare essence of the “frame fixing” behind any Higgs mechanism but in this context it is directly obvious for the human visual system (the “intuition” of the referee). The mathematical formalisms are just not adding anything, these take this as input. We point this out in the modified introductory sentences of section IIA.

It appears that referee 1 invested quite some energy in studying in detail how it all works – we are very grateful! He arrives at excellent suggestions. Indeed, to keep track of the duality-mapping relations between vortices, dislocations and disclinations a “route map” table is most useful. Following his recommendation, we have added this in a new section 1.I. As also mentioned by referee 3 our “sloppy” handling of upper-\lower indices may be confusing for the relativists. But this is actually very easy: getting to business with “crystal geometry” we are invariably dealing with manifolds “flattened” by the crystal while in the compiutations we rely on Euclidean signature: the covariant/contravariant index structure is therefore redundant. At the instance that the Lorentzian signature of the background comes into play (e.g., gravitons) we pay tribute to the index structure. We spell this out in the new section 1J.

These are the changes we have made in the revised paper. Let us now respond to the remaining issues raised by the referee’s:

Referee 1:
Again, we thank the referee for his efforts to study our paper thoroughly and we of course like his appreciation of our effort! We alluded in the above already to his first two points. With regard to his third point: we fear that the referee is cutting here a corner. The gravitational penetration depth is actually determined by the shear modulus, a material property, while the Schwarzschild radius only requires a mass density – there is no universality in this dimensional analysis. More pressing, such questions belong to the realm of “solid cosmologies”. As we emphasize, we stay away from this substance matter. Our impression is that this community is on the right track, wiring in the ‘Higgsing’ through the breaking of spatial shift symmetries. However, when we approach it from the elasticity side we do hit a brick wall in the form of the “messing up” of the (GR) tensor structure due to the bad breaking of Lorentz invariance that by back reaction should also imprint on the dynamical evolution of the background geometry. We find this confusing and it begs for a further analysis. In summary, for these reasons we prefer to ignore this issue raised by the referee in the paper.

Referee 2:
As already announced, the referee is of course right with his recommendation for references to state of the art papers in closely related subjects. We now discuss and cite the papers mentioned in his first three points. We actually ignore the last two papers mentioned under his point three since these appear to be completely unrelated to what we find – as pointed out in Section VII departing from real solids this will not happen (the “dislocation gas” affair).
With regard to his comment in point 3 that we are too sceptical: this is a misunderstanding, we refer in this context just to effects of the “topologization”, given the revised introduction it should now be clear that we having nothing to say regarding elastic inflation and so forth. Point 4: we are quite familiar with this holographic work but the only relation between these monopoles and our disclinations is merely in the word “topology”. Point 5: this is easy, in so far the Goldstone bosons are at stake it is just the same affair discussed in a different language. In the “algebraic” approaches he refers to the absence of rotational modes is eventually rooted in the fact that [L^a, P^b] = i epsilon_{abc} P^c (L and P angular and linear momentum) which is just expressing the semidirect relations between translations and rotations. Notice that the Golstobe counting is in our text used merely as a transition towards the introduction of the topological incarnation, in the form of the well understood “lego-topology” discussed at length in VI-B and Appendix C. Point 6: in condensed matter the “vortex-boson” duality reviewed in Section III has turned into a main stream affair with an associated huge literature in condensed matter physics. E.g., Matthew Fisher received not long ago a Buckly prize for introducing it in the context of the superconductor-insulator transition in the late 1980’s. The papers he mentions have no special standing in the particular context where we discuss this affair. Our quantum liquid crystal review is just very convenient for turorial purposes. Point 7: this is similar, these papers just fit in the “Kleinert” tradition that also summarizes a huge literature and they do not anything useful for our particular purposes.

Referee 3:
We do have the impression that this referee did not manage to get a grip on our material. As acknowledged by the other two referee’s our paper is long because there is much novelty to be found, while it is just beneficial for the readership to take the time to explain the unfamiliar background material. We can assure the referee that the substances we have in the offering (Sections IV-IX) are concisely written and impossible to compress significantly. Reading his report it is obvious that he/she did not even get the elementary points. His first point: as we discussed in the beginning, we fully agree with his statement that a sophisticated literature emerged dealing with neutron star crusts and we refer to that in the new section I-B. Ironically, he seems to have missed our observation in Section IV-E that neutron stars are still small compared to the gravitational penetration depth, being therefore unrelated. His second and technical details (TD) point 2 : attracting the attention to Armas et al., arXiv:1908.01175 is indeed most helpful and we discussed in the beginning this is now put in the limelight. His third and fourth point: this reveals his confusion. Dealing with pinned charge density and especially with fluids (visco-elastic or not) the topologization which is the subject of our paper is just not in existence! This is just a different chapter in the book of physics. With regard to the holography inspired tradition in this regard, it is ironic that since the work by Frenkel in the 1930’s it is generally accepted in the professional fluid community that viscoelastic behaviour is best understood in terms of a solid littered with a low density of free dislocations. Dislocations are not quite part of the canon of this holographic community and it could well be beneficial for these developments to study our material to familiarize with them in the context where these take over control, in the “full” solids that we discuss. Further technical details:
Technical detail 1: We actually spell this out at every relevant instance in the manuscript. It is simple: the time axis is at work dealing with gravitons, as usual probes around the “flattened” (by the crystal) background in Section IVD-G, and when delaing with the curvature fluxoid (Section VII). It appears that the cause of confusion is in the central played by the curvature of the spatial manifold in a co-moving frame which is unusual in GR.
Technical detail 2: fully agreed, and we now highlight this both in the introduction of section II and in the general introduction.
Technical detail 3: the way that {\cal W} is not intuitive, it is the bare essence of the Higgsing. One wants to change the frame arbitrarily but the presence of the crystal constraints it to passive diffeomorphisms and for reasons we spell out we only have to consider small metric fluctuations (locally “self-linearized”, the non linearities are all “collected” in the topological parts).
Technical detail 4: the referee seems to refer here to our “sloppy” dealings with covariant/contravariant indices, as the first referee did. We have added section I.I to the introduction explaining this.
Technical detail 5: For technical convenience much of our derivations revolve around writing down effective actions, from which the EOM’s (Einstein equations) can be derived. As we spell out at length, departing from the crystal the bad breaking of Lorentz invariance messes up the tensor structures. This is in Einstein equation language associated with the stress-energy tensor of the solid. This is announced in the introduction, spelled out in Section II.B, and given some substance in Section IV.G. We leave this for future work: it is not at all clear to us how this is dealt with in the “solid cosmology” literature.
Technical detail 6: again the referee is missing an essential point. As we spell out repeatedly the leading non-linearities are captured by the topological excitations. The very complicated “fracton” dynamics of the dislocations and disclinations will be at center stage in a dynamical setting: the on-going pursuit in materials science. On the other hand, one can consider higher gradient elasticity as a starting point. This is analyzed in detail in the Kleinert book but it turns out to only affect some numbers associated with the dynamics of the topological defects. For this reason we ignore it.

Also on behalf of my co-authors Balm and Beekman,

Jan Zaanen

---

## Round 2 · List of Changes

1. We reformulated the initial parts of the introduction, focussing the attention of the reader on the "topologization", the analogue of Abrikosov lattices in the "crystal gravity" context. For this purpose we rewrote Section I.A, added section I.B concisely reviewing the existing work on solids in GR, spelling out the relations with what we will pursue. Here we added the references to the most modern part of the literature brought to our attention by referee's 2,3. Sections 1C-1G have been subjected to mild text editing to adjust it to the new flow of arguments. We did not touch the long summary section 1H.
2. Following the suggestions of Referee 1 we added the duality overview table in section I.I and we added a very short section I.J explaining our handling of upper- and lower indices.
3. We modified the introduction of Section II, highlighting the paper arXiv:1908.01175 as brought to our attention by Referee 3.

You are currently on this page

Resubmission 2109.11325v2 on 22 April 2022

---

## Editorial Decision

publication_decision_taken:_accept